# Demystifying Structural Disparity in Graph Neural Networks: Can One Size Fit All?

**Haitao Mao**[1], **Zhikai Chen**[1], **Wei Jin**[4], **Haoyu Han**[1],
**Yao Ma**[3], **Tong Zhao**[2], **Neil Shah**[2], **Jiliang Tang**[1]
[1]Michigan State University    [2]Snap Inc    [3]Rensselaer Polytechnic Institute.    [4] Emory University
{haitaoma, chenzh85,hanhaoy1,tangjili}@msu.edu,
{tzhao,nshah}@snap.com    may13@rpi.edu,    wei.jin@emory.edu

## Abstract

Recent studies on Graph Neural Networks(GNNs) provide both empirical and theoretical evidence supporting their effectiveness in capturing structural patterns on both homophilic and certain heterophilic graphs. Notably, most real-world homophilic and heterophilic graphs are comprised of a mixture of nodes in both homophilic and heterophilic structural patterns, exhibiting a structural disparity. However, the analysis of GNN performance with respect to nodes exhibiting different structural patterns, e.g., homophilic nodes in heterophilic graphs, remains rather limited. In the present study, we provide evidence that Graph Neural Networks(GNNs) on node classification typically perform admirably on homophilic nodes within homophilic graphs and heterophilic nodes within heterophilic graphs while struggling on the opposite node set, exhibiting a performance disparity. We theoretically and empirically identify effects of GNNs on testing nodes exhibiting distinct structural patterns. We then propose a rigorous, non-i.i.d PAC-Bayesian generalization bound for GNNs, revealing reasons for the performance disparity, namely the aggregated feature distance and homophily ratio difference between training and testing nodes. Furthermore, we demonstrate the practical implications of our new findings via (1) elucidating the effectiveness of deeper GNNs; and (2) revealing an over-looked distribution shift factor on graph out-of-distribution problem and proposing a new scenario accordingly.

## 1 Introduction

Graph Neural Networks (GNNs) [1–4] are a powerful technique for tackling a wide range of graph-related tasks [5, 3, 6–10], especially node classification [2, 4, 11, 12], which requires predicting unlabeled nodes based on the graph structure, node features, and a subset of labeled nodes. The success of GNNs can be ascribed to their ability to capture structural patterns through the aggregation mechanism that effectively combines feature information from neighboring nodes [13].

GNNs have been widely recognized for their effectiveness on homophilic graphs [14, 2, 11, 4, 15–17]. In homophilic graphs, connected nodes tend to share the same label, which we refer to as *homophilic patterns*. An example of the homophilic pattern is depicted in the upper part of Figure 1, where node features and node labels are denoted by colors (i.e., blue and red) and numbers (i.e., 0 and 1), respectively. We can observe that all connected nodes exhibit homophilic patterns and share the same label 0. Recently, several studies have demonstrated that GNNs can also perform well on certain heterophilic graphs [18, 13, 19]. In heterophilic graphs, connected nodes

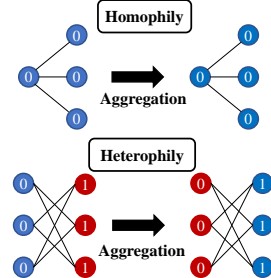

Figure 1: Examples of homophilic and heterophilic patterns. Colors/numbers indicate node features/labels.

tend to have different labels, which we refer to as *heterophilic patterns*. The example in the lower part of Figure 1 shows the heterophilic patterns. Based on this example, we intuitively illustrate how GNNs can work on such heterophilic patterns (lower right): after averaging features over all neighboring nodes, nodes with label 0 completely switch from their initial blue color to red, and vice versa; despite this feature alteration, the two classes remain easily distinguishable since nodes with the same label (number) share the same color (features).

However, existing studies on the effectiveness of GNNs [14, 18, 13, 19] only focus on either homophilic or heterophilic patterns solely and overlook the fact that real-world graphs typically exhibit a mixture of homophilic and heterophilic patterns. Recent studies [20, 21] reveal that many heterophilic graphs, e.g., `Squirrel` and `Chameleon` [22], contain over 20% homophilic nodes. Similarly, our preliminary study depicted in Figure 2 demonstrates that heterophilic nodes are consistently present in many homophilic graphs, e.g., `PubMed` [23] and `Ogbn-arxiv` [24]. Hence, real-world homophilic graphs predominantly consist of homophilic nodes as the majority structural pattern and heterophilic nodes in the minority one, while heterophilic graphs exhibit an opposite phenomenon with heterophilic nodes in the majority and homophilic ones in the minority.

To provide insights aligning the real-world scenario with structural disparity, we revisit the toy example in Figure 1, considering both homophilic and heterophilic patterns together. Specifically, for nodes labeled 0, both homophilic and heterophilic node features appear in blue before aggregation. However, after aggregation, homophilic and heterophilic nodes in label 0 exhibit different features, appearing blue and red, respectively. Such differences may lead to performance disparity between nodes in majority and minority patterns. For instance, in a homophilic graph with the majority pattern being homophilic, GNNs are more likely to learn the association between blue features and class 0 on account of more supervised signals in majority. Consequently, nodes in the majority structural pattern can perform well, while nodes in the minority structural pattern may exhibit poor performance, indicating an over-reliance on the majority structural pattern. Inspired by insights from the above toy example, we focus on answering following questions systematically in this paper: How does a GNN behave when encountering the structural disparity of homophilic and heterophilic nodes within a dataset? and Can one GNN benefit all nodes despite structural disparity?

**Present work**. Drawing inspiration from above intuitions, we investigate how GNNs exhibit different effects on nodes with structural disparity, the underlying reasons, and implications on graph applications. Our study proceeds as follows: **First**, we empirically verify the aforementioned intuition by examining the performance of testing nodes w.r.t. different homophily ratios, rather than the overall performance across all test nodes as in [13, 14, 19]. We show that GCN [2], a vanilla GNN, often underperforms MLP-based models on nodes with the minority pattern while outperforming them on the majority nodes. **Second**, we examine how aggregation, the key mechanism of GNNs, shows different effects on homophilic and heterophilic nodes. We propose an understanding of why GNNs exhibit performance disparity with a non-i.i.d PAC-Bayesian generalization bound, revealing that both feature distance and homophily ratio differences between train and test nodes are key factors leading to performance disparity. **Third**, we showcase the significance of these insights by exploring implications for (1) elucidating the effectiveness of deeper GNNs and (2) introducing a new graph out-of-distribution scenario with an over-looked distribution shift factor. Codes are available at here.

## 2 Prelimaries

**Semi-Supervised Node classification (SSNC).** Let $G = (V, E)$ be an undirected graph, where $V = \{v_1, \cdots, v_n\}$ is the set of $n$ nodes and $E \subseteq V \times V$ is the edge set. Nodes are associated with node features $\mathbf{X} \in \mathbb{R}^{n \times d}$, where $d$ is the feature dimension. The number of class is denoted as $K$. The adjacency matrix $\mathbf{A} \in \{0,1\}^{n \times n}$ represents graph connectivity where $\mathbf{A}[i,j] = 1$ indicates an edge between nodes $i$ and $j$. $\mathbf{D}$ is a degree matrix and $\mathbf{D}[i,i] = d_i$ with $d_i$ denoting degree of node $v_i$. Given a small set of labeled nodes, $V_{\text{tr}} \subseteq V$, SSNC task is to predict on unlabeled nodes $V \setminus V_{\text{tr}}$.

**Node homophily ratio** is a common metric to quantify homophilic and heterophilic patterns. It is calculated as the proportion of a node's neighbors sharing the same label as the node [25, 26, 20]. It is formally defined as $h_i = \frac{|\{u \in \mathcal{N}(v_i) : y_u = y_v\}|}{d_i}$, where $\mathcal{N}(v_i)$ denotes the neighbor node set of $v_i$ and $d_i = |\mathcal{N}(v_i)|$ is the cardinality of this set. Following [20, 27, 25], node $i$ is considered to be homophilic when more neighbor nodes share the same label as the center node with $h_i > 0.5$. We

define the graph homophily ratio $h$ as the average of node homophily ratios $h = \frac{\sum_{i \in V} h_i}{|V|}$. Moreover, this ratio can be easily extended to higher-order cases $h_i^{(k)}$ by considering $k$-order neighbors $\mathcal{N}_k(v_i)$.

**Node subgroup** refers to a subset of nodes in the graph sharing similar properties, typically homophilic and heterophilic patterns measured with node homophily ratio. Training nodes are denoted as $V_{\text{tr}}$. Test nodes $V_{\text{te}}$ can be categorized into $M$ node subgroups, $V_{\text{te}} = \bigcup_{m=1}^{M} V_m$, where nodes in the same subgroup $V_m$ share similar structural pattern.

## 3 Effectiveness of GNN on nodes with different structural properties

In this section, we explore the effectiveness of GNNs on different node subgroups exhibiting distinct structural patterns, specifically, homophilic and heterophilic patterns. It is different from previous studies [13, 14, 18, 28, 19] that primarily conduct analysis on the whole graph and demonstrate effectiveness with an overall performance gain. These studies, while useful, do not provide insights into the effectiveness of GNNs on different node subgroups, and may even obscure scenarios where GNNs fail on specific subgroups despite an overall performance gain. To accurately gauge the effectiveness of GNNs, we take a closer examination on node subgroups with distinct structural patterns. The following experiments are conducted on two common homophilic graphs, Ogbn-arxiv [24] and Pubmed [23], and two heterophilic graphs, Chameleon and Squirrel [22]. These datasets are chosen since GNNs can achieve better overall performance than MLP. Experiment details and related work on GNN disparity are in Appendix G and A, respectively.

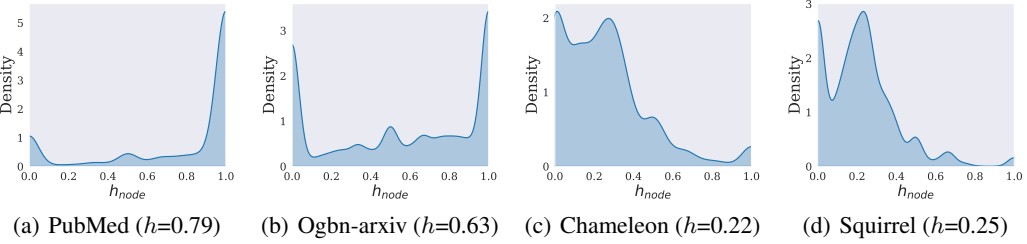

(a) PubMed ($h$=0.79)  (b) Ogbn-arxiv ($h$=0.63)  (c) Chameleon ($h$=0.22)  (d) Squirrel ($h$=0.25)

Figure 2: Node homophily ratio distributions. All graphs exhibit a mixture of homophilic and heterophilic nodes despite various graph homophily ratio $h$.

**Existence of structural pattern disparity within a graph** is to recognize real-world graphs exhibiting different node subgroups with diverse structural patterns, before investigating the GNN effectiveness on them. We demonstrate node homophily ratio distributions on the aforementioned datasets in Figure 2. We can have the following observations. **Obs.1:** All four graphs exhibit a mixture of both homophilic and heterophilic patterns, rather than a uniform structural patterns. **Obs.2:** In homophilic graphs, the majority of nodes exhibit a homophilic pattern with $h_i > 0.5$, while in heterophilic graphs, the majority of nodes exhibit the heterophilic pattern with $h_i \leq 0.5$. We define nodes in majority structural pattern as majority nodes, e.g., homophilic nodes in a homophilic graph.

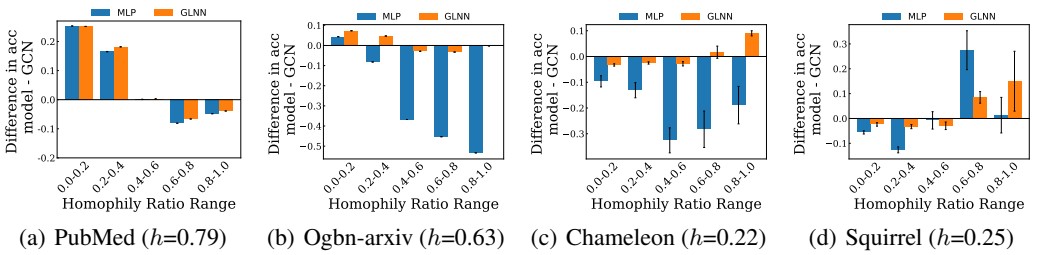

(a) PubMed ($h$=0.79)  (b) Ogbn-arxiv ($h$=0.63)  (c) Chameleon ($h$=0.22)  (d) Squirrel ($h$=0.25)

Figure 3: Performance comparison between GCN and MLP-based models. Each bar represents the accuracy gap on a specific node subgroup exhibiting a homophily ratio within the range specified on the x-axis. MLP-based models often outperform GCN on heterophilic nodes in homophilic graphs and homophilic nodes in heterophilic graphs with a positive value.

**Examining GCN performance on different structural patterns.** To examine the effectiveness of GNNs on different structural patterns, we compare the performance of GCN [2] a vanilla GNN, with two MLP-based models, vanilla MLP and Graphless Neural Network (GLNN) [29], on testing nodes with different homophily ratios. It is evident that the vanilla MLP could have a large performance gap compared to GCN (i.e., 20% in accuracy) [13, 29, 2]. Consequently, an under-trained vanilla MLP comparing with a well-trained GNN leads to an unfair comparison without rigorous conclusion. Therefore, we also include an advanced MLP model GLNN. It is trained in an advanced manner via distilling GNN predictions and exhibits performance on par with GNNs. Notably, only GCN has the ability to leverage structural information during the inference phase while both vanilla MLP and GLNN models solely rely on node features as input. This comparison ensures a fair study on the effectiveness of GNNs in capturing different structural patterns with mitigating the effects of node features. Experimental results on four datasets are presented in Figure 3. In the figure, y-axis corresponds to the accuracy differences between GCN and MLP-based models where positive indicates MLP models can outperform GCN; while x-axis represents different node subgroups with nodes in the subgroup satisfying homophily ratios in the given range, e.g., [0.0-0.2]. Based on experimental results, the following key observations can be made: **Obs.1:** In homophilic graphs, both GLNN and MLP demonstrate superior performance on the heterophilic nodes with homophily ratios in [0-0.4] while GCN outperforms them on homophilic nodes. **Obs.2:** In heterophilic graphs, MLP models often outperform on homophilic nodes yet underperform on heterophilic nodes. Notably, vanilla MLP performance on Chameleon is worse than that of GCN across different subgroups. This can be attributed to the training difficulties encountered on Chameleon, where an unexpected similarity in node features from different classes is observed [30]. Our observations indicate that despite the effectiveness of GCN suggested by [13, 19, 14], GCN exhibits limitations with performance disparity across homophilic and heterophilic graphs. It motivates investigation why GCN benefits majority nodes, e.g., homophilic nodes in homophilic graphs, while struggling with minority nodes. Moreover, additional results on more datasets and significant test results are shown in Appendix H and L.

**Organization.** In light of the above observations, we endeavor to understand the underlying causes of this phenomenon in the following sections by answering the following research questions. Section 3.1 focuses on how aggregation, the fundamental mechanism in GNNs, affects nodes with distinct structural patterns differently. Upon identifying differences, Section 3.2 further analyzes how such disparities contribute to superior performance on the majority nodes as opposed to minority nodes. Building on these observations, Section 3.3 recognizes the key factors driving performance disparities on different structural patterns with a non-i.i.d. PAC-Bayes bound. Section 3.4 empirically corroborates the validity of our theoretical analysis with real-world datasets.

## 3.1 How does aggregation affect nodes with structural disparity differently?

In this subsection, we examine how aggregation reveals different effects on nodes with structural disparity, serving as a precondition for performance disparity. Specifically, we focus on the discrepancy between nodes from the same class but with different structural patterns.

For a controlled study on graphs, we adopt the contextual stochastic block model (CSBM) with two classes. It is widely used for graph analysis, including generalization [13, 14, 31, 18, 32, 33], clustering [34], fairness [35, 36], and GNN architecture design [37–39]. Typically, nodes in CSBM model are generated into two disjoint sets $\mathcal{C}_1$ and $\mathcal{C}_2$ corresponding to two classes, $c_1$ and $c_2$, respectively. Each node with $c_i$ is associated with features $x \in \mathbb{R}^d$ sampling from $N(\boldsymbol{\mu}_i, I)$, where $\mu_i$ is the feature mean of class $c_i$ with $i \in \{1, 2\}$. The distance between feature means in different classes $\rho = \|\boldsymbol{\mu}_1 - \boldsymbol{\mu}_2\|$, indicating the classification difficulty on node features. Edges are then generated based on intra-class probability $p$ and inter-class probability $q$. For instance, nodes with class $c_1$ have probabilities $p$ and $q$ of connecting with another node in class $c_1$ and $c_2$, respectively. The CSBM model, denoted as $\mathrm{CSBM}(\boldsymbol{\mu}_1, \boldsymbol{\mu}_2, p, q)$, presumes that all nodes follow either homophilic with $p > q$ or heterophilic patterns $p < q$ exclusively. However, this assumption conflicts with real-world scenarios, where graphs often exhibit both patterns simultaneously, as shown in Figure 2. To mirror such scenarios, we propose a variant of CSBM, referred to as CSBM-Structure (CSBM-S), allowing for the simultaneous description of homophilic and heterophilic nodes.

**Definition 1** (CSBM-S$(\mu_1, \mu_2, (p^{(1)}, q^{(1)}), (p^{(2)}, q^{(2)}), \mathrm{Pr(homo)})$)**.** *The generated nodes consist of two disjoint sets $\mathcal{C}_1$ and $\mathcal{C}_2$. Each node feature $x$ is sampled from $N(\mu_i, I)$ with $i \in \{1, 2\}$. Each set $\mathcal{C}_i$ consists of two subgroups: $\mathcal{C}_i^{(1)}$ for nodes in homophilic pattern with intra-class and*

*inter-class edge probability $p^{(1)} > q^{(1)}$ and $\mathcal{C}_i^{(2)}$ for nodes in heterophilic pattern with $p^{(2)} < q^{(2)}$. $\Pr(homo)$ denotes the probability that the node is in homophilic pattern. $\mathcal{C}_i^{(j)}$ denotes node in class $i$ and subgroup $j$ with $(p^{(j)}, q^{(j)})$. We assume nodes follow the same degree distribution with $p^{(1)} + q^{(1)} = p^{(2)} + q^{(2)}$.*

Based on the neighborhood distributions, the mean aggregated features $\mathbf{F} = \mathbf{D}^{-1}\mathbf{A}\mathbf{X}$ obtained follow Gaussian distributions on both homophilic and heterophilic subgroups.

$$\mathbf{f}_i^{(j)} \sim N\left(\frac{p^{(j)}\boldsymbol{\mu}_1 + q^{(j)}\boldsymbol{\mu}_2}{p^{(j)} + q^{(j)}}, \frac{\mathbf{I}}{\sqrt{d_i}}\right), \text{for } i \in \mathcal{C}_1^{(j)}; \mathbf{f}_i^{(j)} \sim N\left(\frac{q^{(j)}\boldsymbol{\mu}_1 + p^{(j)}\boldsymbol{\mu}_2}{p^{(j)} + q^{(j)}}, \frac{\mathbf{I}}{\sqrt{d_i}}\right), \text{for } i \in \mathcal{C}_2^{(j)}$$
(1)

Where $\mathcal{C}_i^{(j)}$ is the node subgroups with structural pattern with $(p^{(j)}, q^{(j)})$ in label $i$. Our initial examination of different effects on aggregation focuses on the aggregated feature distance between homophilic and heterophilic node subgroups within class $c_1$.

**Proposition 1.** *The aggregated feature mean distance between homophilic and heterophilic node subgroups within class $c_1$ is $\left\|\frac{p^{(1)}\boldsymbol{\mu}_1 + q^{(1)}\boldsymbol{\mu}_2}{p^{(1)} + q^{(1)}} - \frac{p^{(2)}\boldsymbol{\mu}_1 + q^{(2)}\boldsymbol{\mu}_2}{p^{(2)} + q^{(2)}}\right\| > 0$, indicating the aggregated feature of homophilic and heterophilic subgroups are from different feature distributions, with a mean distance larger than 0 distance before aggregation, since original node features draw from the same distribution, regardless of different structural patterns.*

Notably, the distance between original features is regardless of the structural pattern. This proposition suggests that aggregation results in a distance gap between different patterns within the same class.

In addition to node feature differences with the same class, we further examine the discrepancy between nodes $u$ and $v$ with the same aggregated feature $\mathbf{f}_u = \mathbf{f}_v$ but different structural patterns. We examine the discrepancy with the probability difference of nodes $u$ and $v$ in class $c_1$, denoted as $|\mathbf{P}_1(y_u = c_1|\mathbf{f}_u) - \mathbf{P}_2(y_v = c_1|\mathbf{f}_v)|$. $\mathbf{P}_1$ and $\mathbf{P}_2$ are the conditional probability of $y = c_1$ given the feature $\mathbf{f}$ on structural patterns $(p^{(1)}, q^{(1)})$ and $(p^{(2)}, q^{(2)})$, respectively.

**Lemma 1.** *With assumptions (1) A balance class distribution with $\mathbf{P}(Y = 1) = \mathbf{P}(Y = 0)$ and (2) aggregated feature distribution shares the same variance $\sigma$. When nodes $u$ and $v$ have the same aggregated features $\mathbf{f}_u = \mathbf{f}_v$ but different structural patterns, $(p^{(1)}, q^{(1)})$ and $(p^{(2)}, q^{(2)})$, we have:*

$$|\mathbf{P}_1(y_u = c_1|\mathbf{f}_u) - \mathbf{P}_2(y_v = c_1|\mathbf{f}_v)| \leq \frac{\rho^2}{\sqrt{2\pi}\sigma}|h_u - h_v|$$
(2)

Notably, above assumptions are not strictly necessary but employed for elegant expression. Lemma 1 implies that nodes with a small homophily ratio difference $|h_1 - h_2|$ are likely to share the same class, and vice versa. Proof details and additional analysis on between-class effects are in Appendix D.

### 3.2 How does Aggregation Contribute to Performance Disparity?

We have established that aggregation can affect nodes with distinct structural patterns differently. However, it remains to be elucidated how such disparity contributes to performance improvement predominantly on majority nodes as opposed to minority nodes. It should be noted that, notwithstanding the influence on features, test performance is also profoundly associated with training labels. Performance degradation may occur when the classifier is inadequately trained with biased training labels.

We then conduct an empirical discriminative analysis taking both mean aggregated features and training labels into consideration. Drawing inspiration from existing literature [40–43], we describe the discriminative ability with the distance between train class prototypes [44, 45], i.e., feature mean of each class, and the corresponding test class prototype within the same class $i$. For instance, it can be denoted as $||\mu_i^{\text{tr}} - \mu_i^{\text{ma}}||$, where $\mu_i^{\text{tr}}$ and $\mu_i^{\text{ma}}$ are the prototype of class $i$ on train nodes and test majority nodes, respectively. A smaller value suggests that majority test nodes are close to train nodes within

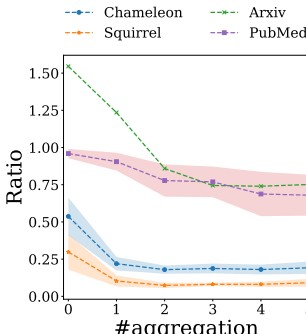

Figure 4: Illustration on discriminative ratio variation along with aggregation. x-axis denotes the number of aggregations and y-axis denotes the discriminative ratio.

the same class, thus implying superior discriminative ability. A relative discriminative ratio is then proposed to compare the discriminative ability between majority and minority nodes. It can be denoted as: $r = \sum_{i=1}^{K} \frac{||\boldsymbol{\mu}_i^{\text{tr}} - \boldsymbol{\mu}_i^{\text{ma}}||}{||\boldsymbol{\mu}_i^{\text{tr}} - \boldsymbol{\mu}_i^{\text{mi}}||}$ where $\mu_i^{\text{mi}}$ corresponds to the prototype on minority test nodes. A lower relative discriminative ratio suggests that majority nodes are easier to be predicted than minority nodes.

The relative discriminative ratios are then calculated on different hop aggregated features and original features denote as 0-hop. Experimental results are presented in Figure 4, where the discriminative ratio shows an overall decrease tendency as the number of aggregations increases across four datasets. This indicates that majority test nodes show better discriminative ability than the minority test nodes along with more aggregation. We illustrate more results on GCN in Appendix K. Furthermore, instance-level experiments other than class prototypes are in Appendix C.

### 3.3 Why does Performance Disparity Happen? Subgroup Generalization Bound for GNNs

In this subsection, we conduct a rigorous analysis elucidating primary causes for performance disparity across different node subgroups with distinct structural patterns. Drawing inspiration from the discriminative metric described in Section 3.2, we identify two key factors for satisfying test performance: (1) test node $u$ should have a close feature distance $\min_{v \in V_{\text{tr}}} \|\mathbf{f}_u - \mathbf{f}_v\|$ to training nodes $V_{\text{tr}}$, indicating that test nodes can be greatly influenced by training nodes. (2) With identifying the closest training node $v$, nodes $u$ and $v$ should be more likely to share the same class, where $\sum_{c_i \in \mathcal{C}} |\mathbf{P}(y_u = c_i|\mathbf{f}_u) - \mathbf{P}(y_v = c_i|\mathbf{f}_v)|$ is required to be small. The second factor, focusing on whether two close nodes are in the same class, is dependent on the homophily ratio difference $|h_u - h_v|$, as shown in Lemma 1. Notably, since training nodes are randomly sampled, their structural patterns are likely to be the majority one. Therefore, training nodes will show a smaller homophily ratio difference with majority test nodes sharing the same majority pattern than minority test nodes, resulting in the performance disparity in distinct structural patterns. We substantiate the above intuitions with controllable synthetic experiments in Appendix B.

To rigorously examine the role of aggregated feature distance and homophily ratio difference in performance disparity, we derive a non-i.i.d. PAC-Bayesian GNN generalization bound, based on the Subgroup Generalization bound of Deterministic Classifier [46]. We begin by stating key assumptions on graph data and GNN model to clearly delineate the scope of our theoretical analysis. All remaining assumptions, proof details, and background on PAC-Bayes analysis can be found in Appendix F. Moreover, a comprehensive introduction on the generalization ability on GNN can be found in A.

**Definition 2** (Generalized CSBM-S model). *Each node subgroup $V_m$ follows the CSBM distribution $V_m \sim CSBM(\boldsymbol{\mu}_1, \boldsymbol{\mu}_2, p^{(i)}, q^{(i)})$, where different subgroups share the same class mean but different intra-class and inter-class probabilities $p^{(i)}$ and $q^{(i)}$. Moreover, node subgroups also share the same degree distribution as $p^{(i)} + q^{(i)} = p^{(j)} + q^{(j)}$.*

Instead of CSBM-S model with one homophilic and heterophilic pattern, we take the generalized CSBM-S model assumption, allowing more structural patterns with different levels of homophily.

**Assumption 1** (GNN model). *We focus on SGC [16] with the following components: (1) a one-hop mean aggregation function $g$ with $g(X, G)$ denoting the output. (2) MLP feature transformation $f(g_i(X, G); W_1, W_2, \cdots, W_L)$, where $f$ is a ReLU-activated $L$-layer MLP with $W_1, \cdots, W_L$ as parameters for each layer. The largest width of all the hidden layers is denoted as $b$.*

Notably, despite analyzing simple GNN architecture theoretically, similar with [46, 13, 47], our theory analysis could be easily extended to the higher-order case with empirical success across different GNN architectures shown in Section 3.4.

Our main theorem is based on the PAC-Bayes analysis which typically aims to bound the generalization gap between the expected margin loss $\mathcal{L}_m^0$ on test subgroup $V_m$ for a margin 0 and the empirical margin loss $\widehat{\mathcal{L}}_{\text{tr}}^{\gamma}$ on train subgroup $V_{\text{tr}}$ for a margin $\gamma$. Those losses are generally utilized in PAC-Bayes analysis[48–51]. More details are found in Appendix F. The formulation is shown as follows:

**Theorem 1** (Subgroup Generalization Bound for GNNs). *Let $\tilde{h}$ be any classifier in the classifier family $\mathcal{H}$ with parameters $\{\widetilde{W}_l\}_{l=1}^{L}$. for any $0 < m \leq M$, $\gamma \geq 0$, and large enough number of the training nodes $N_{tr} = |V_{tr}|$, there exist $0 < \alpha < \frac{1}{4}$ with probability at least $1 - \delta$ over the sample of*

$y^{tr} := \{y_i\}_{i \in V_{tr}}$, *we have:*

$$\mathcal{L}_m^0(\tilde{h}) \leq \widehat{\mathcal{L}}_{tr}^\gamma(\tilde{h}) + O\left(\underbrace{\frac{K\rho}{\sqrt{2\pi}\sigma}(\epsilon_m + |h_{tr} - h_m| \cdot \rho)}_{(a)} + \underbrace{\frac{b\sum_{l=1}^L \|\widetilde{W}_l\|_F^2}{(\gamma/8)^{2/L} N_{tr}^\alpha}(\epsilon_m)^{2/L}}_{(b)} + \mathbf{R}\right) \quad (3)$$

The bound is related to three terms: **(a)** describes both large homophily ratio difference $|h_{\text{tr}} - h_m|$ and large aggregated feature distance $\epsilon = \max_{j \in bV_m} \min_{i \in V_{\text{tr}}} \|g_i(X, G) - g_j(X, G)\|_2$ between test node subgroup $V_m$ and training nodes $V_{\text{tr}}$ lead to large generalization error. $\rho = \|\boldsymbol{\mu}_1 - \boldsymbol{\mu}_2\|$ denotes the original feature separability, independent of structure. $K$ is the number of classes. **(b)** further strengthens the effect of nodes with the aggregated feature distance $\epsilon$, leads to a large generalization error. **(c)** $\mathbf{R}$ is a term independent with aggregated feature distance and homophily ratio difference, depicted as $\frac{1}{N_{\text{tr}}^{1-2\alpha}} + \frac{1}{N_{\text{tr}}^{2\alpha}} \ln \frac{LC(2B_m)^{1/L}}{\gamma^{1/L}\delta}$, where $B_m = \max_{i \in V_{\text{tr}} \cup V_m} \|g_i(X, G)\|_2$ is the maximum feature norm. $\mathbf{R}$ vanishes as training size $N_0$ grows. Proof details are in Appendix F

Our theory suggests that both homophily ratio difference and aggregated feature distance to training nodes are key factors contributing to the performance disparity. Typically, nodes with large homophily ratio difference and aggregated feature distance to training nodes lead to performance degradation.

## 3.4 Performance Disparity Across Node Subgroups on Real-World Datasets

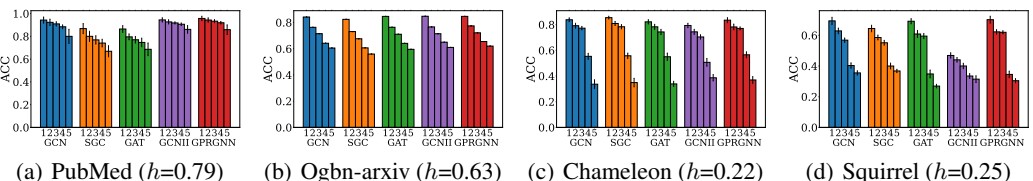

(a) PubMed ($h$=0.79)    (b) Ogbn-arxiv ($h$=0.63)    (c) Chameleon ($h$=0.22)    (d) Squirrel ($h$=0.25)

Figure 5: Test accuracy disparity across node subgroups by **aggregated-feature distance and homophily ratio difference** to training nodes. Each figure corresponds to a dataset, and each bar cluster corresponds to a GNN model. A clear performance decrease tendency can be found from subgroups 1 to 5 with increasing differences to training nodes.

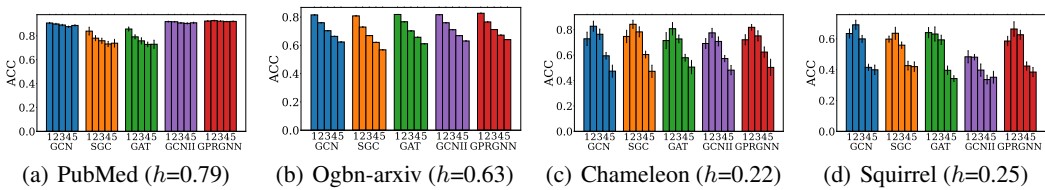

(a) PubMed ($h$=0.79)    (b) Ogbn-arxiv ($h$=0.63)    (c) Chameleon ($h$=0.22)    (d) Squirrel ($h$=0.25)

Figure 6: Test accuracy disparity across node subgroups by **aggregated-feature distance** to train nodes. Each figure corresponds to a dataset, and each bar cluster corresponds to a GNN model. A clear performance decrease tendency can be found from subgroups 1 to 5 with increasing differences to training nodes.

To empirically examine the effects of theoretical analysis, we compare the performance on different node subgroups divided with both homophily ratio difference and aggregated feature distance to training nodes with popular GNN models including GCN [2], SGC [16], GAT [11], GCNII [52], and GPRGNN [53]. Typically, test nodes are partitioned into subgroups based on their disparity scores to the training set in terms of both 2-hop homophily ratio $h_i^{(2)}$ and 2-hop aggregated features $\mathbf{F}^{(2)}$ obtained by $\mathbf{F}^{(2)} = (\tilde{\mathbf{D}}^{-1}\tilde{\mathbf{A}})^2 \mathbf{X}$, where $\tilde{\mathbf{A}} = \mathbf{A} + \mathbf{I}$ and $\tilde{\mathbf{D}} = \mathbf{D} + \mathbf{I}$. For a test node $i$, we measure the node disparity by (1) selecting the closest training node $v = \arg\min_{v \in V_0} \|\mathbf{F}_u^{(2)} - \mathbf{F}_v^{(2)}\|$ (2) then calculating the disparity score $s_u = \|\mathbf{F}_u^{(2)} - \mathbf{F}_v^{(2)}\|_2 + |h_u^{(2)} - h_v^{(2)}|$, where the first and the second terms correspond to the aggregated-feature distance and homophily ratio differences,

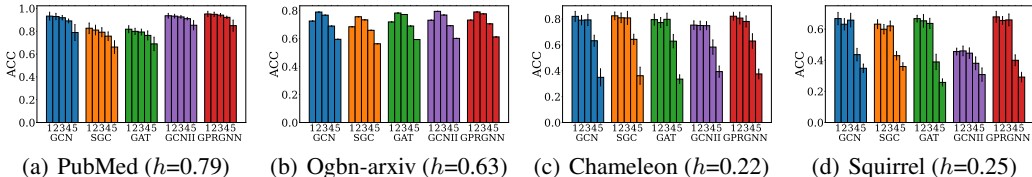

(a) PubMed ($h$=0.79)    (b) Ogbn-arxiv ($h$=0.63)    (c) Chameleon ($h$=0.22)    (d) Squirrel ($h$=0.25)

Figure 7: Test accuracy disparity across node subgroups by **homophily ratio difference** to train nodes. Each figure corresponds to a dataset, and each bar cluster corresponds to a GNN model. A clear performance decrease tendency can be found from subgroups 1 to 5 with increasing differences to training nodes.

respectively. We then sort test nodes in terms of the disparity score and divide them into 5 equal-binned subgroups accordingly. Performance on different node subgroups is presented in Figure 5 with the following observations. **Obs.1:** We note a clear test accuracy degradation with respect to the increasing differences in aggregated features and homophily ratios. Furthermore, we investigate on the individual effect of aggregated feature distance and homophily ratio difference in Figure 6 and 7, respectively. An overall trend of performance decline with increasing disparity score is evident though some exceptions are present. **Obs.2:** When only considering the aggregated feature distance, there is no clear trend among groups 1, 2, and 3 on GCN, SGC, and GAT on heterophilic datasets. **Obs.3:** When only considering the homophily ratio difference, there is no clear trend among groups 1, 2, and 3 across four datasets. These observations underscore the importance of both aggregated-feature distance and homophily ratio differences in shaping GNN performance disparity. Combining these factors together provides a more comprehensive and accurate understanding of the reason for GNN performance disparity. For a more comprehensive analysis, we further substantiate our finding involving higher-order information and a wider array of datasets in Appendix J.1.

**Summary** In this section, we study GNN performance disparity on nodes with distinct structural patterns and uncover its underlying causes. We primarily investigate the impact of aggregation, the key component in GNNs, on nodes with different structural patterns in Sections 3.1 and 3.2. We observe that aggregation effects vary across nodes with different structural patterns, notably enhancing the discriminative ability on majority nodes. These observed performance disparities inspire us to identify crucial factors contributing to GNN performance disparities across nodes with a non-i.i.d PAC-Bayes bound in Section 3.3. The theoretical analysis indicates that test nodes with larger aggregated feature distances and homophily ratio differences with training nodes experience performance degradation. We substantiate our findings on real-world datasets in Section 3.4.

## 4 Implications of graph structural disparity

In this section, we illustrate the significance of our findings on structural disparity via (1) elucidating the effectiveness of existing deeper GNNs (2) unveiling an over-looked aspect of distribution shift on graph out-of-distribution (OOD) problem, and introducing a new OOD scenario accordingly. Experimental details and discussions on more implications are in Appendix G and O, respectively.

### 4.1 Elucidate the effectiveness of Deeper GNNs

Deeper GNNs [53, 52, 15, 54–57] enable each node to capture more complex higher-order graph structure than vanilla GCN, via reducing the over-smoothing problem [58–60, 55, 61] Deeper GNNs empirically exhibit overall performance improvement, as demonstrated in Appendix G.2. Nonetheless, which structural patterns deeper GNNs can exceed and the reason for its effectiveness remain unclear. To investigate this problem, we compare vanilla GCN with different deeper GNNs, including GPRGNN[53], APPNP[15], and GCNII[52], on node subgroups with varying homophily ratios, adhering the same setting with Figure 3. Experimental results are shown in Figure 8. We can observe that deeper GNNs primarily surpass GCN on minority node subgroups with slight performance trade-offs on the majority node subgroups. We conclude that the effectiveness of deeper GNNs majorly contributes to improved discriminative ability on minority nodes. Additional results on more datasets and significant test are in Appendix I and M.

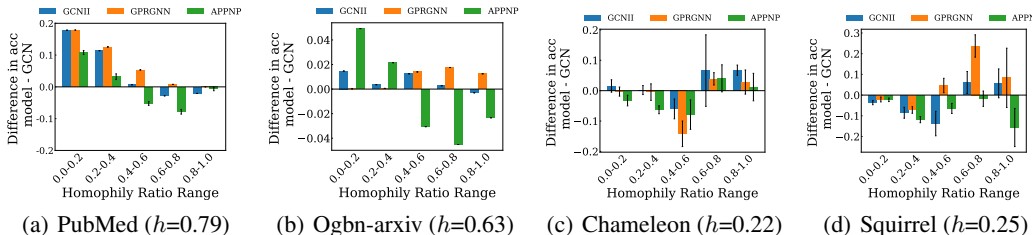

| (a) PubMed ($h$=0.79) | (b) Ogbn-arxiv ($h$=0.63) | (c) Chameleon ($h$=0.22) | (d) Squirrel ($h$=0.25) |

Figure 8: Performance comparison between GCN and deeper GNNs. Each bar represents the accuracy gap on a specific node subgroup exhibiting homophily ratio within range specified on x-axis.

Having identified where deeper GNNs excel, reasons why effectiveness primarily appears in the minority node group remain elusive. Since the superiority of deeper GNNs stems from capturing higher-order information, we further investigate how higher-order homophily ratio differences vary on the minority nodes, denoted as, $|h_u^{(k)} - h_v^{(k)}|$, where node $u$ is the test node, node $v$ is the closest train node to test node $u$. We concentrate on analyzing these minority nodes $V_{\text{mi}}$ in terms of default one-hop homophily ratio $h_u$ and examine how $\sum_{u \in V_{\text{mi}}} |h_u^{(k)} - h_v^{(k)}|$ varies with different $k$ orders. Experimental results are shown in Figure 9, where a decreasing trend of homophily ratio difference is observed along with more neighborhood hops. The smaller homophily ratio difference leads to smaller generalization errors with better performance. This observation is consistent with [20], where heterophilic nodes in heterophilic graphs exhibit large higher-order homophily ratios, implicitly leading to a smaller homophily ratio difference.

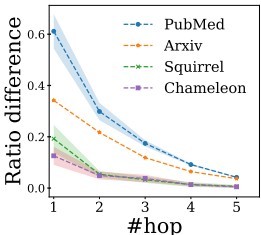

Figure 9: Multiple hop homophily ratio differences between training and minority test nodes.

## 4.2 A new graph out-of-distribution scenario

The graph out-of-distribution (OOD) problem refers to the underperformance of GNN due to distribution shifts on graphs. Many Graph OOD scenarios [62–65, 24, 66, 46, 67], e.g., biased training labels, time shift, and popularity shift, have been extensively studied. These OOD scenarios can be typically categorized into covariate shift with $\mathbf{P}^{\text{train}}(\mathbf{X}) \neq \mathbf{P}^{\text{test}}(\mathbf{X})$ and concept shift [68–70] with $\mathbf{P}^{\text{train}}(\mathbf{Y}|\mathbf{X}) \neq \mathbf{P}^{\text{test}}(\mathbf{Y}|\mathbf{X})$. $\mathbf{P}^{\text{train}}(\cdot)$ and $\mathbf{P}^{\text{test}}(\cdot)$ denote train and test distributions, respectively. Existing graph concept shift scenarios [62, 66] introduce different environment variables $e$ resulting in $\mathbf{P}(\mathbf{Y}|\mathbf{X}, e_{\text{train}}) \neq \mathbf{P}(\mathbf{Y}|\mathbf{X}, e_{\text{test}})$ leading to spurious correlations. To address existing concept shifts, algorithms [62, 71] have been developed to capture the environment-invariant relationship $\mathbf{P}(\mathbf{Y}|\mathbf{X})$. Nonetheless, existing concept shift settings overlook the scenario where there is not a unique environment-invariant relationship $\mathbf{P}(\mathbf{Y}|\mathbf{X})$. For instance, $\mathbf{P}(\mathbf{Y}|\mathbf{X}_{\text{homo}})$ and $\mathbf{P}(\mathbf{Y}|\mathbf{X}_{\text{hete}})$ can be different, indicated in Section 3.1. $\mathbf{X}_{\text{homo}}$ and $\mathbf{X}_{\text{hete}}$ correspond to features of nodes in homophilic and heterophilic patterns. Notably, homophilic and heterophilic patterns are crucial task knowledge that cannot be recognized as the irrelevant environmental variable. Consequently, we find that homophily ratio difference between train and test sets could be an important factor leading to an overlook concept shift, namely, graph structural shift. Notably, structural patterns cannot be considered as environment variables given their integral role in node classification task. The practical implications of this concept shift are substantiated by the following scenarios: **(1)** graph structural shift frequently occurs in most graphs, with a performance degradation in minority nodes, as depicted in Figure 3. **(2)** graph structural shift hides secretly in existing graph OOD scenarios. For instance, the FaceBook-100 dataset [62] reveals a substantial homophily ratio difference between train and test sets, averaging 0.36. This discrepancy could be the primary cause of OOD performance deterioration since the exist OOD algorithms [62, 72] that neglect such a concept shift can only attain a minimal average performance gain of 0.12%. **(3)** graph structural shift is a recurrent phenomenon in numerous real-world applications where new nodes in graphs may exhibit distinct structural patterns. For example, in a recommendation system, existing users with rich data can receive well-personalized recommendations in the exploitation stage (homophily), while new users with less data may receive diverse recommendations during the exploration stage (heterophily).

Given the prevalence and importance of the graph structural shift, we propose a new graph OOD scenario emphasizing this concept shift. Specifically, we introduce a new data split on existing datasets, namely Cora, CiteSeer, PubMed, Ogbn-Arxiv, Chameleon, and Squirrel, where majority nodes are selected for train and validation, and minority ones for test. This data split strategy highlights the

Table 1: Performance (Accuracy) on the proposed OOD split.

|            | Pubmed     | Ogbn-Arxiv  | Squirrel    | Chameleon   |
|------------|------------|-------------|-------------|-------------|
| GCN(i.i.d) | 89.18±0.15 | 72.99±0.14  | 58.09±0.71  | 75.09±0.79  |
| GCN        | 51.04±0.16 | 34.94±0.07  | 32.13±4.93  | 43.35±3.47  |
| MLP        | 68.38±0.43 | 33.17±0.37  | 24.57±0.77  | 34.78±4.97  |
| GLNN       | 67.51±0.25 | 35.89±0.14  | 31.51±0.70  | 47.01±1.09  |
| GCNII      | 67.76±0.36 | 36.81±0.14  | 37.15±1.39  | 41.25±2.03  |
| GPRGNN     | 57.24±0.18 | 34.95±0.43  | 42.43±7.71  | 35.27±7.67  |
| SRGNN      | 57.91±0.10 | 40.37±1.65  | 37.62±1.74  | 42.09±0.43  |
| EERM       | 65.37±1.35 | 34.23±0.46  | 40.93±0.57  | 45.84±1.05  |
| EERM(II)   | 67.59±0.91 | 40.28±0.84  | 44.31±0.40  | 48.59±0.78  |

homophily ratio difference and the corresponding concept shift. To better illustrate the challenges posed by our new scenario, we conduct experiments on models including GCN, MLP, GLNN, GPRGNN, and GCNII. We also include graph OOD algorithms, SRGNN [63] and EERM [62], with GCN encoders. EERM(II) is a variant of EERM with a GCNII encoder For a fair comparison, we show GCN performance on an i.i.d. random split, GCN(i.i.d.), sharing the same node sizes for train, validation, and test. Results are shown in Table 1 while additional ones are in Appendix G.4. Following observations can be made: **Obs.1:** The performance degradation can be found by comparing OOD setting with i.i.d. one across four datasets, confirming OOD issue existence. **Obs.2:** MLP-based models and deeper GNNs generally outperform vanilla GCN, demonstrating the superiority on minority nodes. **Obs.3:** Graph OOD algorithms with GCN encoders struggle to yield good performance across datasets, indicating a unique challenge over other Graph OOD scenarios. This primarily stems from the difficulty in learning both accurate relationships on homophilic and heterophilic nodes with distinct $\mathbf{P}(\mathbf{Y}|\mathbf{X})$. Nonetheless, it can be alleviated by selecting a deeper GNN encoder, as the homophily ratio difference may vanish in higher-order structure information, with reduced concept shift. **Obs.4:** EERM(II), EERM with GCNII, outperforms the one with GCN. Observations suggest that GNN architecture plays an indispensable role in addressing graph OOD issues, highlighting the new direction.

## 5 Conclusion & Discussion

In conclusion, this work provides crucial insights into GNN performance meeting structural disparity, common in real-world scenarios. We recognize that aggregation exhibits different effects on nodes with structural disparity, leading to better performance on majority nodes than those in minority. The understanding also serves as a stepping stone for multiple graph applications.

Our exploration majorly focuses on common datasets with clear majority structural patterns while real-world scenarios, offering more complicated datasets, posing new challenges Additional experiments are conducted on Actor, IGB-tiny, Twitch-gamer, and Amazon-ratings. Dataset details and experiment results are in Appendix G.3 and Appendices H-K, respectively. Despite our understanding is empirically effective on most datasets, further research and more sophisticated analysis are still necessary. Discussions on the limitation and broader impact are in Appendix N and O, respectively.

## 6 Acknowledgement

We want to thank Xitong Zhang, He Lyu, Wenzhuo Tang at Michigan State University, Yuanqi Du at Cornell University, Haonan Wang at the National University of Singapore, and Jianan Zhao at Mila for their constructive comments on this paper.

This research is supported by the National Science Foundation (NSF) under grant numbers CNS 2246050, IIS1845081, IIS2212032, IIS2212144, IIS2153326, IIS2212145, IOS2107215, DUE 2234015, DRL 2025244 and IOS2035472, the Army Research Office (ARO) under grant number W911NF-21-1-0198, the Home Depot, Cisco Systems Inc, Amazon Faculty Award, Johnson&Johnson, JP Morgan Faculty Award and SNAP.

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

# Appendix

## Table of Contents

# A   Related work

**Graph Neural Networks (GNNs)** have emerged as a powerful technique in Deep Learning, specifically designed for graph-structured data. They address the limitations of traditional neural networks in dealing with irregular data structures. GNNs learn node representations by aggregating neighborhood and transforming features recursively. The node representation can then be successfully utilized to a wide range of graph-related downstream tasks [2, 5, 3, 6–8, 73, 74].

The aggregation mechanism in GNNs is often viewed as feature smoothing [61, 75, 76]. This perspective leads some recent studies [26, 53, 20, 77] claiming that GNN models are overly reliant on homophilic patterns and unsuited to capturing heterophilic patterns To accommodate heterophilic graphs, recent works propose carefully designed GNN architectures, e.g., CPGNN [78], GGNN [79], GPRGNN [53], GCNII [52] GBK-GNN [27], ACM-GNN [19], Bernnet [80].

More recent analyses on GNNs [18, 13, 19] indicate that even GCN [2], a vanilla GNN, can deliver strong performance on certain heterophilic graphs. According to these findings, new metrics and understandings [81, 82, 21, 19, 13, 28] have been proposed to further expose the remaining weaknesses of GNNs.

There is a concurrent work [28] investigates when Graph Neural Networks help with node classification with a comparison between GNN and MLP. Nonetheless, our work is distinct from [28] in two primary ways:

- [28] is majorly grounded on the CSBM-H model focusing on different feature variances for different classes, from a feature perspective. In contrast, our work examines the homophilic and heterophilic patterns from a structural perspective instead of sharing the same node homophily ratio across the graph.

- [28] proposes a new metric to identify whether GNN can outperform graph-agnostic MLP on particular datasets. Nonetheless, it still focuses on all the nodes in the whole graph together. In contrast, our work reveals the scenario where GNNs show performance degradation on certain node subgroup across most homophilic and heterophilic graph datasets. Typically, we focus on a node subgroup perspective, rather than all nodes in the whole graph. Our paper highlights the drawback of GNNs in a more general case.

**Fairness on Graph** Although recent years have seen a satisfying performance from Graph Neural Networks (GNNs), risks can be found. GNNs may unintentionally learn and perpetuate biases present in the training data, potentially resulting in unfair outcomes for certain populations. Such risks raise concerns about biases and discriminatory behaviors when GNNs are used in human-centric applications.

Fairness issues on graphs can be roughly categorized into attribute bias and structural bias, based on the source of the bias. Several works [35, 83–85] focus on fairness issues originating from sensitive node attributes, e.g., gender or underrepresented ethnic groups. Other literature [63, 46, 86–88] focuses on addressing fairness issues arising from different structural information, e.g., degree, geodesic distance to the training node, and Personal Pagerank score. Our work aligns closely with structural bias, showing that a larger homophily ratio difference between training and test nodes may lead to performance degradation. To the best of our knowledge, we are the first to propose this performance disparity induced by the homophily ratio difference.

**Generalization ability analysis on Graph Neural Network** The generalization ability analysis on Graph Neural Networks aims to develop theoretical understandings of GNNs with a focus on the uniqueness of the graph structure. Recent research progress reveals the generalization ability [89, 47, 90–92] of GNN across different tasks and settings. Our work typically studies the generalization ability on the transductive node classification task indicating relationship with [93, 14, 18, 13]. Typically, [93] employs the transductive uniform stability [94] to understand why deeper GNNs generalize better than the vanilla GCN. Several studies [14, 18, 13] investigate the generalization of GCN under the CSBM model assumption, while they focus on either homophilic or heterophilic patterns rather than consider both patterns together. [46] provides the first non-i.i.d. PAC-Bayes generalization bound on GNNs, serving as the basis of our theory 1. However, there are key differences between our work and [46] detailed as follows:

- Assumption difference. [46] assumes the existence of $c$-Lipschitz continuous functions on the conditional probability of $y_i = k$ given the aggregated feature $g_i(X, G)$, denoted as $\mathbf{P}(y_i = k|g_i(X, G))$. The assumption suggests that nodes with a closer distance are likely to belong to the same class. However, this assumption may not align with the real-world scenario where nodes with closer distances may belong to different classes when they exhibit a large homophily ratio difference, as shown in Lemma 1. In contrast, we utilize a different assumption considering the influence of the structural pattern on the conditional probability, enabling analyses on the scenario that nodes with a closer distance but from different classes. This scenario happens frequently when the graph is a mixture of homophilic and heterophilic patterns. An intuitive illustration can be found in Figure 1.

- Application scope difference. [46] is primarily focused on the homophilic graphs and does not easily extend to the heterophilic ones. Empirical evidence can be found in Section 3.4 and Appendix J. In contrast, our theory can be established on most datasets except for the Actor dataset which structural information hardly helps.

**Disclaimer:** Structural disparity is different from distribution shift between train and test data. The structural disparity is that homophilic and heterophilic patterns exist simultaneously in a single node set. Such disparity consistently exists in all graphs, as shown in Figure 2. If randomly sample train and test data, the distribution shift between train and test will not happen. Both train and test sets will have homophilic and heterophilic patterns, indicating structural disparity happens on both the train and test sets.

# B  Investigation on the effectiveness of homophily ratio difference with targeted synthetic edge addition algorithm.

In Section 3.4, we intuitively show that a larger homophily ratio difference could be a key reason for performance disparity. We demonstrate the effectiveness of intuition with both theoretical analysis and experiments conducted on real-world datasets. To further verify the influence of the homophily ratio difference, we conduct more controllable synthetic experiments adopted from [13], adding different amounts of synthetic edges on real-world datasets, to further evaluate how the performance of GCN, a vanilla GNN, changes on varied homophily ratio differences. Typically, we are to manually make heterophilic nodes more homophilic and make homophilic nodes more heterophilic with the targeted homophilic edge addition and the targeted heterophilic edge addition algorithms, respectively. Notably, despite synthetic edges added, we utilize the real-world dataset serves as the entrance to keep the analysis more approach to the real-world scenario. The following subsections are organized as follows. We first introduce the targeted heterophilic and homophilic edge addition algorithms and how it leads to larger homophily ratio difference in Appendix B.1. Detailed experiment analysis can be found in Appendix B.2. Further experiment details can be found in the Appendix B.3.

## B.1  Targeted heterophilic & homophilic edge addition algorithms

Both targeted heterophilic and homophilic edge addition algorithms commence with standard, real-world benchmark graphs, modifying the structure by adding synthetic edges to manipulate the homophily ratio on either train nodes or test nodes. For example, we can add synthetic heterophilous edges on the targeted test nodes in a homophily graph. Consequently, a homophily ratio difference between the training and testing nodes can be observed.

We first introduce the homophilic edge addition algorithm, as shown in Algorithm 1. Specially, we introduce a total of $K$ edges on either training or targeted test nodes on heterophilic dataset, denoted as $\mathcal{V}_{\text{targeted}}$. For each edge addition, a node $u$ is uniformly sampled from the targeted node set, $\mathcal{V}_{\text{targeted}}$, and obtains its corresponding label, $y_u$. Another targeted node $v$ sharing the same label, $y_v$, is selected, and a homophilic edge is added between them, resulting in a homophily ratio decrease. Consequently, we can observe a homophily ratio decrease on both target nodes $u$ and $v$ with the new homophily edge added. Notably, we only add synthetic edges among targeted nodes, ensuring the homophily ratio unchanged on the other nodes.

The heterophilic edge addition algorithm, as shown in the Algorithm 2, is analogous to the homophilic one. The only difference is the selection of a newly added heterophilic edge instead of the homophilic one. While homophilic edges can be readily added by connecting nodes within the same label, adding

cross-label heterophilic edges poses a new challenge that the target node $u$ should be connected to which label, other than the target label $y_u$. Typically, we follow the principle [13] that nodes with the same label should share similar neighborhood label distributions. This principle aligns with the CSBM assumption, discussed in Section 3. Specifically, given a real-world graph $G$, we first define a discrete neighborhood target label distribution $\mathcal{D}_c$ for each label $c \in \mathcal{C}$. Examples of $\mathcal{D}_c$ can be found in Appendix B.3. Specifically, we will first randomly select a label $c$ based on distribution $\mathcal{D}_c$. Another node $v$ in the target set with label $c$ is then selected, and a heterophilic edge is added between them. This process enables us to add heterophilic edges on target nodes following similar neighborhood label distributions.

---

**Algorithm 1:** Targeted Homophilic Edge Addition

---

**input** : $\mathcal{G} = \{\mathcal{V}, \mathcal{E}\}, \mathcal{V}_{targeted} \in \mathcal{V}, K$, and $\{\mathcal{V}_c\}_{c=0}^{|\mathcal{C}|-1}$
**output** : $\mathcal{G}' = \{\mathcal{V}, \mathcal{E}'\}$

1  Initialize $\mathcal{E}' = \mathcal{E}, k = 1$ ;
2  **while** $1 \le k \le K$ **do**
3      **do**
4          Sample node $i \sim \mathsf{Uniform}(\mathcal{V}_{targeted})$;
5          Obtain the label, $y_i$ of node $i$;
6          Sample node $j \sim \mathsf{Uniform}(\mathcal{V}_{y_i} \cap \mathcal{V}_{targeted})$;
7      **while** $(i, j) \in \mathcal{E}'$;
8      Update edge set $\mathcal{E}' = \mathcal{E}' \cup \{(i,j)\}$;
9      $k \leftarrow k+1$;
10 **return** $\mathcal{G}' = \{\mathcal{V}, \mathcal{E}'\}$

---

---

**Algorithm 2:** Targeted Heterophilous Edge Addition

---

**input** : $\mathcal{G} = \{\mathcal{V}, \mathcal{E}\}, \mathcal{V}_{targeted} \in \mathcal{V}, K, \{\mathcal{D}_c\}_{c=0}^{|\mathcal{C}|-1}$ and $\{\mathcal{V}_c\}_{c=0}^{|\mathcal{C}|-1}$
**output** : $\mathcal{G}' = \{\mathcal{V}, \mathcal{E}'\}$

1  Initialize $\mathcal{E}' = \mathcal{E}, k = 1$ ;
2  **while** $1 \le k \le K$ **do**
3      Sample node $i \sim \mathsf{Uniform}(\mathcal{V}_{targeted})$;
4      Obtain the label, $y_i$ of node $i$;
5      Sample a number $r \sim \mathsf{Uniform}(0,1)$;
6      Sample a label $c \sim \mathcal{D}_{y_i}$;
7      Sample node $j \sim \mathsf{Uniform}(\mathcal{V}_c \cap \mathcal{V}_{targeted})$;
8      Update edge set $\mathcal{E}' = \mathcal{E}' \cup \{(i,j)\}$;
9      $k \leftarrow k+1$;
10 **return** $\mathcal{G}' = \{\mathcal{V}, \mathcal{E}'\}$

---

### B.2  Detailed experiment results

In this section, we conduct experiments on the homophilic graph, Cora, employing the targeted homophilic edge addition algorithm, and the heterophilic graph, Squirrel, employing the targeted heterophilic edge addition algorithm. For each dataset, we apply the synthetic edge addition algorithm to training nodes and a subset of test nodes, respectively, resulting in homophily ratio difference between training and test sets. The synthetic edges are added gradually until reaching the predefined maximum budget $K_{\max}$, resulting in multiple synthetic graphs generated. More experimental details on the synthetic graphs can be found in Appendix B.3. The GCN, a vanilla GNN model, is trained from the sketch on each synthetic graph. The experimental results are illustrated in Figure 10, where the $x$-axis represents the edge permutation ratio calculated as $\frac{K}{K_{\max}}$, where $K$ is the number of added edges on the $i$-th synthetic graph. The $y$-axis represents the test performance when adding synthetic edge on training nodes. When adding synthetic edges on the targeted test subset, The $y$-axis represents the performance on the targeted test node subset. A clear degradation in test performance is observed with more and more synthetic edges added to test nodes and training nodes exclusively. The above observations clearly demonstrate that the GCN test performance will degrade with a larger homophilic ratio difference between train and test nodes.

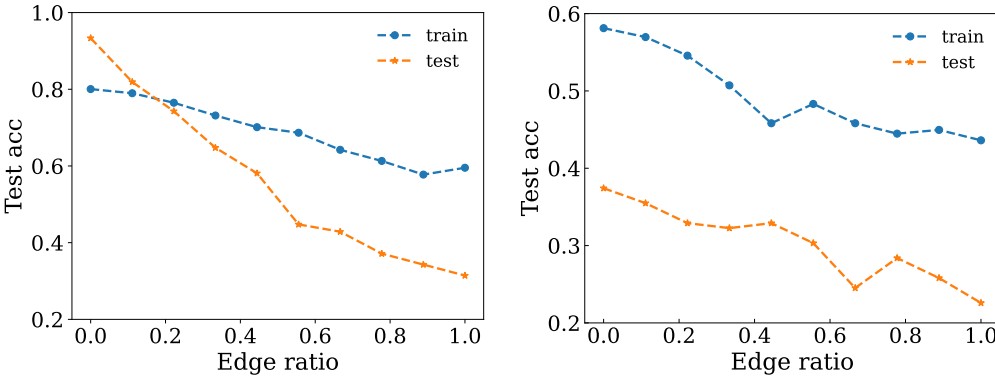

(a) Synthetic graphs generated from Cora with the targeted heterophilic edge algorithm

(b) Synthetic graphs generated from Squirrel with the targeted homophilious edge algorithm

Figure 10: The performance of GCN on synthetic graphs with various edge permutation ratios. The $y$-axis represents the test performance on all nodes when adding synthetic edge on training nodes. When adding synthetic edges on the targeted test subset, The $y$-axis represents the performance on the targeted test node subset. Test performance generally decrease with more synthetic edges added with a large homophily ratio difference.

## B.3 Details on the generated graph

In this subsection, we elaborate on the details of the synthetic graphs generated in Appendix B.2. Specifically, we provide more details regarding the distributions $\mathcal{D}_c, c \in \mathcal{C}$ employed in the Cora and Squirrel datasets. We adopted circulant matrix-like designs for simplicity and straightforward implementation. We also show details on the number of added edges, and the corresponding graph homophily ratio on the Cora and Squirrel datasets.

**Cora** We utilize the targeted heterophilic edge addition algorithm on the Cora dataset. The Cora dataset has seven labels that we represent as $0, 1, 2, 3, 4, 5, 6$, and the neighborhood distributions $\mathcal{D}_c, c \in \mathcal{C}$ for adding heterophilic synthetic edges are presented as follows.

$$\mathcal{D}_0 : \mathsf{Categorical}([0, 0.5, 0, 0, 0, 0, 0.5]),$$
$$\mathcal{D}_1 : \mathsf{Categorical}([0.5, 0, 0.5, 0, 0, 0, 0]),$$
$$\mathcal{D}_2 : \mathsf{Categorical}([0, 0.5, 0, 0.5, 0, 0, 0]),$$
$$\mathcal{D}_3 : \mathsf{Categorical}([0, 0, 0.5, 0, 0.5, 0, 0]),$$
$$\mathcal{D}_4 : \mathsf{Categorical}([0, 0, 0, 0.5, 0, 0.5, 0]),$$
$$\mathcal{D}_5 : \mathsf{Categorical}([0, 0, 0, 0, 0.5, 0, 0.5]),$$
$$\mathcal{D}_6 : \mathsf{Categorical}([0.5, 0, 0, 0, 0, 0.5, 0]).$$

The number of added edges $K$ and the corresponding homophily ratio $h$ on the targeted test nodes and the train nodes are presented in Table 2 and 3, respectively.

Table 2: $K$ and $h$ values for graphs with synthetic edges added targeted test nodes on Cora

| $K$ | 100 | 200 | 300 | 400 | 500 | 600 | 700 | 800 | 900 |
|---|---|---|---|---|---|---|---|---|---|
| $h$ | 0.645 | 0.483 | 0.382 | 0.321 | 0.279 | 0.249 | 0.223 | 0.206 | 0.194 |

Table 3: $K$ and $h$ values for graphs with synthetic edges added targeted training nodes on Cora

| $K$ | 200 | 400 | 600 | 800 | 1000 | 1200 | 1400 | 1600 | 1800 |
|---|---|---|---|---|---|---|---|---|---|
| $h$ | 0.620 | 0.507 | 0.424 | 0.365 | 0.324 | 0.292 | 0.264 | 0.243 | 0.224 |

**Squirrel** We utilize the targeted homophilic edge addition algorithm on the Squirrel dataset. We first sample the label from discrete uniform distributions. Then, we randomly select two nodes with the same label and do not have an edge between them. The number of added edges $K$ and the corresponding homophily ratio $h$ on the targeted test nodes and train nodes are presented in Table 4 and 5, respectively.

Table 4: $K$ and $h$ values for graphs with synthetic edges added targeted test nodes on Squirrel

| $K$ | 100 | 200 | 300 | 400 | 500 | 600 | 700 | 800 | 900 |
|---|---|---|---|---|---|---|---|---|---|
| $h$ | 0.237 | 0.345 | 0.412 | 0.467 | 0.505 | 0.536 | 0.562 | 0.585 | 0.606 |

Table 5: $K$ and $h$ values for graphs with synthetic edges added targeted train nodes on Squirrel

| $K$ | 1500 | 3000 | 4500 | 6000 | 7500 | 9000 | 10500 | 12000 | 13500 |
|---|---|---|---|---|---|---|---|---|---|
| $h$ | 0.261 | 0.299 | 0.330 | 0.356 | 0.379 | 0.399 | 0.417 | 0.434 | 0.448 |

## C  Instance-level discriminative analysis

In section 3.2, we conduct a discriminative analysis considering the distance between the feature means from different classes from a global perspective. In this section, we further investigate the discriminability from an instance-level perspective. Specifically, we focus on the feature distance between different nodes feature rather than the feature mean.

To qualify the discriminative difficulty on the feature of each individual test node, we propose the two metrics, local agreement ratio, and local accuracy difference. The local agreement ratio aims to measure the local clustering property in a KNN manner. It can be calculated with the following steps: (1) Given a particular node, find the top-$k$ feature-close train nodes. We set $k = 9$ in our experiment, and L2 distance is utilized as the feature distance metric. (2) Given the top-$k$ closest train nodes, we examine the number of nodes on each label. If there exists a particular label $c$ with the number of nodes over $\frac{k}{2}$, it indicates that over half of neighborhood nodes reach an agreement on the class $c$. (3) The local agreement ratio can then be calculated as the proportion of nodes reaching an agreement in the test set, denoted as:

$$r = \frac{\sum_{v \in V_{\text{te}}} \mathbb{1}\left[\exists c \in \mathcal{C}, |\mathcal{M}_v^c| > \frac{k}{2}\right]}{|V_{\text{te}}|} \tag{4}$$

where $r$ is the local agreement ratio, $V_{\text{te}}$ and $\mathcal{C}$ are the test node set and class set, respectively. $\mathcal{M}_v$ is the node set including top-$k$ nearest training of the node $v$. $|\mathcal{M}_v^c|$ is the number of nearest training nodes in class $c$. A larger agreement ratio on the test set indicates the data are more clustered with a close distance between train and test nodes. Nonetheless, a higher agreement ratio does not naturally lead to a better discriminative ability. Despite the top-$k$ feature-close nodes reaching an agreement on a particular class, the test node may have a different class from the agreement. It indicates that the center node misaligns with the top-$k$ feature-close nodes.

The local accuracy is then proposed to identify whether the category of the center node aligns with the agreement from top-$k$ feature-close nodes. Concretely speaking, the local agreement accuracy is the proportion of agreement nodes in the test set. It can be calculated as:

$$\text{Acc}_{\text{local}} = \frac{\sum_{v \in V_{\text{agree}}} \mathbb{1}\left[c_v = c_{\mathcal{N}_v}\right]}{|V_{\text{agree}}|} \tag{5}$$

where $V_{\text{agree}}$ is the test node set reaching top-$k$ feature-close nodes agreement. $c_v$ is the category of node $v$. $c_{\mathcal{M}_v}$ is the agreement category from feature-close nodes. A higher local agreement accuracy indicates that most agreement nodes are aligned with the same category.

Similar to the relative discriminative ratio proposed in Section 3.2, we illustrate the local agreement accuracy improvement on the majority nodes over minority nodes. Experiments are conducted on four homophilic datasets, Cora, CiteSeer, PubMed and Ogbn-arxiv, and two heterophilic datasets, Chameleon and Squirrel. The experimental setting details can be found in Section G. Experiment

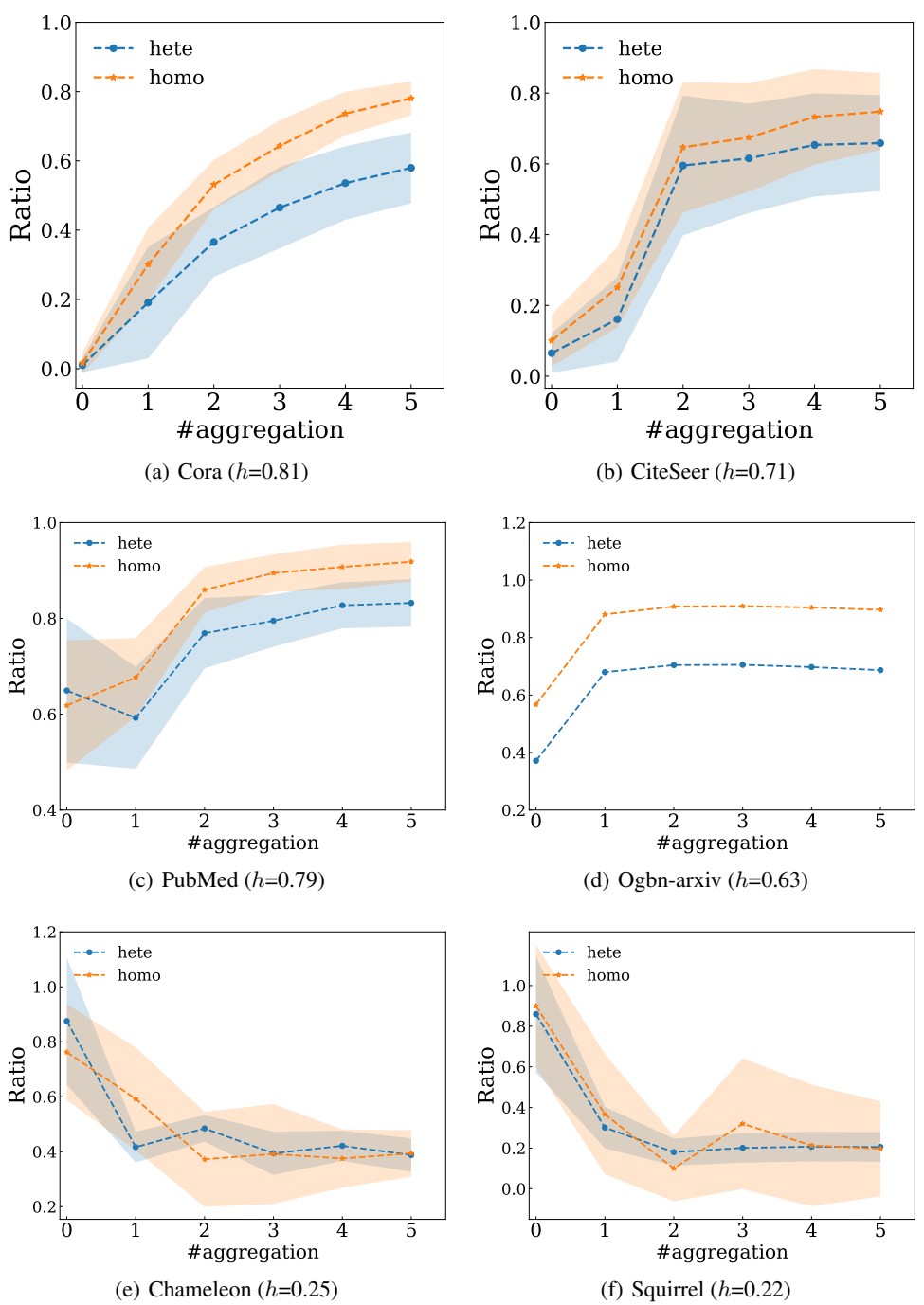

Figure 11: Illustration of the change on local agreement ratio along with the aggregation. The x-axis represents the number of aggregations and the y-axis represents the local agreement ratio. For homophilic graphs, the local agree ratio generally increases along with more aggregations, indicating better cluster effects. For heterophilic graphs, the local agreement ratio shows an opposite phenomenon, which decreases consistently.

results on local agreement ratio and local agreement accuracy difference are illustrated in Figure 11 and 12, respectively.

The observations can be found as follows: (1) For homophilic graphs, the local agree ratio generally increases along with more aggregations, indicating better cluster effects. Meanwhile, the relative accuracy improvement on the majority nodes also increases, further indicating the disparity effects on different nodes group with more improvement on the majority nodes. (2) For heterophilic graphs, the local agreement ratio shows an opposite phenomenon, which decreases consistently. Meanwhile, the relative accuracy improvement on the majority nodes only increases on the first two hops and decrease and decline from the third hop. The potential reason is that, despite a general global trend, the heterophilic patterns on each individual node may still be quite complicated, with a local pattern shift disparity. We leave the discussion on more complex local structure patterns as the future work.

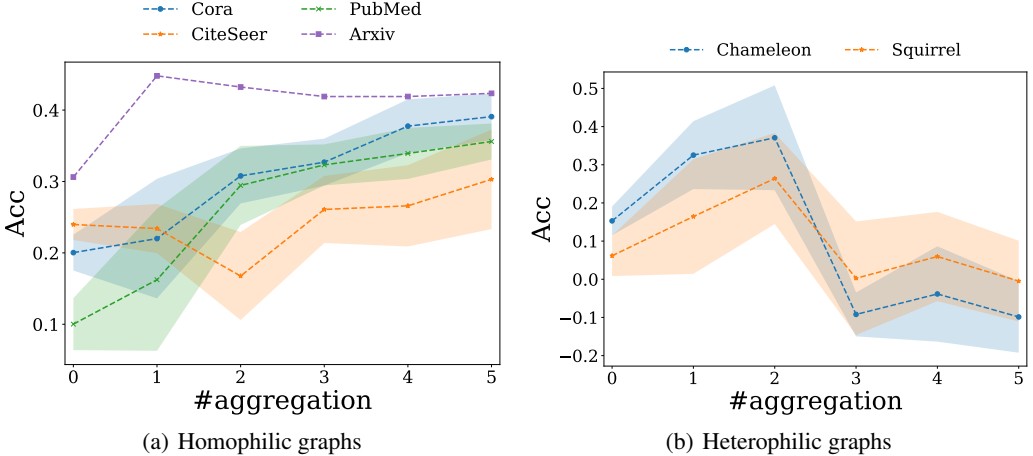

(a) Homophilic graphs                    (b) Heterophilic graphs

Figure 12: Illustration of the change on local agreement accuracy difference between the majority and minority patterns along with the aggregation. The x-axis represents the number of aggregations and the y-axis represents the majority local agreement accuracy minus the minority local agreement accuracy. For homophilic graphs, the relative accuracy improvement on the majority nodes also increases, further, while the relative accuracy improvement on the majority nodes only increases on the first two hops and decrease and decline from the third hop.

# D Effects of aggregation on nodes in different classes with structural disparity

In Section 3.1, we examine the behavior difference between nodes from the same class but with different structural patterns. In this section, we further provide a more complicated analysis focusing on the between-class patterns, i.e., linear separability. Notably, linear separability is a good indicator of feature differences in different classes, where features with better linear separability can be easier to distinguish with a suitable linear classifier.

## D.1 Linear separability analysis based on CSBM model

To ease the analysis, we utilize the CSBM-S model as the data assumption showing as follows:

**Definition 1** (CSBM-S$(\mu_1, \mu_2, (p^{(1)}, q^{(1)}), (p^{(2)}, q^{(2)}), \Pr(\text{homo}))$). *The generated nodes consist of two disjoint sets $\mathcal{C}_1$ and $\mathcal{C}_2$. each node feature $x$ is sampled from $N(\mu_i, I)$ with $i \in \{1, 2\}$. Each set $\mathcal{C}_i$ consists of two subgroups: $\mathcal{C}_i^{(1)}$ for nodes in homophilic pattern with intra-class and inter-class edge probability $p^{(1)} > q^{(1)}$ and $\mathcal{C}_i^{(2)}$ for nodes in heterophilic pattern with $p^{(2)} > q^{(2)}$. $\Pr(\text{homo})$ denotes the probability that the node is in the homophilic pattern. $\mathcal{C}_i^{(j)}$ denotes node in class $i$ and subgroup $j$ with $(p^{(j)}, q^{(j)})$. We assume nodes follow the same degree distribution with $p^{(1)} + q^{(1)} = p^{(2)} + q^{(2)}$.*

The original node features follow the Gaussian distribution:

$$\mathbf{x}_i \sim N(\boldsymbol{\mu}_1, \mathbf{I}) \text{ for } i \in \mathcal{C}_1; \text{ and } \mathbf{x}_i \sim N(\boldsymbol{\mu}_2, \mathbf{I}) \text{ for } i \in \mathcal{C}_2 \tag{6}$$

Where $\mathcal{C}_i$ corresponds to the node set corresponding to class $i$.

Based on the neighborhood distributions, the mean aggregated features $\mathbf{F} = \mathbf{D}^{-1}\mathbf{A}\mathbf{X}$ obtained follow Gaussian distributions on both homophilic and heterophilic subgroups.

$$\mathbf{f}_i^{(j)} \sim N\left(\frac{p^{(j)}\boldsymbol{\mu}_1 + q^{(j)}\boldsymbol{\mu}_2}{p^{(j)} + q^{(j)}}, \frac{\mathbf{I}}{\sqrt{d_i}}\right), \text{ for } i \in \mathcal{C}_1^{(j)}; \mathbf{f}_i^{(j)} \sim N\left(\frac{q^{(j)}\boldsymbol{\mu}_1 + p^{(j)}\boldsymbol{\mu}_2}{p^{(j)} + q^{(j)}}, \frac{\mathbf{I}}{\sqrt{d_i}}\right), \text{ for } i \in \mathcal{C}_2^{(j)}$$
(7)

Where $\mathcal{C}_i^{(j)}$ is the node subgroups with structural pattern with $(p^{(j)}, q^{(j)})$ in label $i$. To more accurately assess the effectiveness of the aggregation mechanism, we execute the largest-margin linear classifiers on nodes and evaluate their performance, to illustrate the linear separability. Notably, linear separability depends on the distance between the mean features of different classes as well as the standard deviations within each class. Typically, we focus on examining the linear separability on (1) nodes from different classes with the same structural patterns (2) nodes from different classes with different structural patterns, i.e., homophilic and heterophilic patterns.

We first examine the linear separability for nodes from different classes within the same structural pattern. We can summarize the proposition as follows:

**Lemma 2** (Linear separability on nodes with the same structural patterns ). *Considering mean aggregated features are from the same structural pattern* $\mathbf{f}_i^{(j)}$, *for* $i \in \{1, 2\}$. *For any node* $i$, *the largest-margin linear classifier on* $\mathbf{f}_i^{(j)}$ *will have a lower probability to misclassify than* $\mathbf{x}_i$, *when* $d_i > \frac{(p^{(1)}+q^{(1)})^2}{(p^{(1)}-q^{(1)})^2}$

The detailed proof can be found in Appendix D.3. The proposition suggests that aggregated features $\mathbf{f}$ have better linear separability than the original feature $x$ when the node $i$ satisfies $d_i > (p^{(1)} + q^{(1)})^2/(p^{(1)}-q^{(1)})^2$. For instance, when $p^{(1)} = 0.9$ and $q^{(1)} = 0.1$, the aggregated features are easier show better linear separability with $d_i > 1.75$, which are commonly met in real-world scenarios. This indicates that aggregation is likely to contribute to improved feature separability within the same node subgroup sharing a similar structural property.

We then examine the linear separability for nodes from different classes within different structural patterns, e.g., $h_1^{(1)}$ and $h_2^{(2)}$. Before diving deep into the rigorous analysis, we first illustrate a special example to show how the structure can lead to worse linear separability. When $p^{(1)} = q^{(2)}$ and $q^{(1)} = p^{(1)}$, we can identify that the $h_1^{(1)}$ and $h_2^{(2)}$ are exactly generated from the same distribution. Such distribution can hardly be linearly separable. We then conduct a more rigorous analysis showing a similar observation with the same pattern one. We can summarize the proposition as follows:

**Lemma 3** (Linear separability on nodes with different structural patterns). *Consider features are from different structural patterns, where* $\mathbf{f}_i^{(1)}$ *for* $i \in \mathcal{C}_1$ *and* $\mathbf{f}_i^{(2)}$ *for* $i \in \mathcal{C}_2$. *For any node* $i$, *the largest-margin linear classifier will have a lower probability to misclassify* $\mathbf{f}_i^{(1)}$ *for* $i \in \mathcal{C}_1$ *and* $\mathbf{f}_i^{(2)}$ *for* $i \in \mathcal{C}_2$ *than* $\mathbf{x}_i$ *when* $d_i > \frac{(p^{(1)}+q^{(1)})^2}{(p^{(1)}-q^{(2)})^2}$

we can find that aggregated feature show better separability when $d_i > (p^{(1)} + q^{(1)})^2/|p^{(1)} - q^{(2)}|^2$. The detailed proof can be found in Appendix D.4. For instance, when $p^{(1)} = 0.9, q^{(1)} = 0.1$ and $p^{(2)} = 0.2, q^{(2)} = 0.8$, the aggregated features can only show better linear separability with $d_i > 100$, which are hardly met in real-world scenarios. Notice that the only difference with the one in the same pattern one is that the denominator is $|p^{(1)} - q^{(2)}|^2$, smaller than the $|p^{(1)} - q^{(1)}|^2$ since $q^{(1)} > q^{(2)}$. It indicates that only nodes with higher degrees. e.g., $d_i = 100$, can achieve improved linear separability.

Based on our analysis, we can draw the conclusion that when nodes are in the same pattern, aggregation can show improved linear separability on $(p^{(1)} + q^{(1)})^2/|p^{(1)} - q^{(1)}|^2 < d_i < (p^{(1)} + q^{(1)})^2/|p^{(1)} - q^{(2)}|^2$ while the ones in the different patterns cannot. It indicates that aggregation can help when nodes are in the same pattern, however, show limitation when nodes are from different patterns. Notice that, such linear separability can be evidence for different behavior on nodes with different patterns. Nonetheless, it cannot directly indicate the performance disparity. Feature separability is conducted based on the ideal classifier with the largest margin. Better feature

separability does not necessarily result in better performance if the classifier is not well-trained with biased training data.

## D.2 Linear separability experiment on synthetic CSBM dataset

In this section, we aim to empirically examine the validity of the theoretical analysis results in Section D.1. We generate graph with the CSBM model with the following detailed settings. The class mean distance $\rho = \|\mu_1 - \mu_2\|$ is set to $0.1$. The feature dimension $d$ and the number of nodes $n$ are set to 50, and 500, respectively. $p$ and $q$ correspond to probabilities of the intra-class and inter-class probability, respectively. The mean aggregation is defined as $\mathbf{F} = \mathbf{D}^{-1}\mathbf{A}\mathbf{X}$. It is the key to generating graphs with different structural properties. To generate the homophilic node subgroup with $p > q$, we utilize the following settings with $(p = 0.01, q = 0.005)$, $(p = 0.01, q = 0.003)$, and $(p = 0.01, q = 0.001)$. To generate the heterphilic node subgroup with $p < q$, we utilize the following settings with $(p = 0.001, q = 0.005)$, $(p = 0.001, q = 0.003)$, and $(p = 0.001, q = 0.002)$.

Experiments are conducted to examine how a linear model, logistic regression, fits effectively on the aggregated features. Higher performance on logistic regression indicates better linear separability of features. It is important to note that we focus on the fitting ability rather than the generalization performance. Consequently, we do not evaluate with a test set; instead, we provide all labels for training and assess the performance on all the nodes.

Experiments on the logistic regression are conducted on the homophilic nodes solely, heterophilic nodes solely, and a mixture of homophilic and heterophilic nodes. Typically, we aim to show the linear separability both within the same structural pattern and between homophilic and heterophilic patterns. Experimental results are presented in Table 6. The following observations can be made: (1) When considering nodes following the same structural pattern, either homophilic pattern or heterophilic pattern, better performance can be observed when the probability difference $|p - q|$ between intra-class and inter-class probability is larger. (2) Comparing the performance considering both structural patterns with the one on a single structural pattern, we reveal a noticeable performance degradation on the one with mixture patterns. This suggests that the nodes with different structural patterns results in decreased linear separability. (3) Comparing the performance among different mixing structural patterns, it is evident that larger homophily ratio differences, e.g., (p=0.01, q=0.005) compared to (p=0.001, q=0.005), yield poorer performance than those with smaller homophily ratio differences. The above observations further supports the validity of Lemma 2 and 3, Theorem 1 .

Table 6: The performance of logistic regression algorithm on homophilic nodes, heterophilic nodes, and a mixture of homophilic and heterophilic nodes. The results on the first row and first column correpond to the performance on homophilic nodes and heterophilic nodes, solely.

| Hete\Homo | - | p=0.01, q=0.005 | p=0.01, q=0.003 | p=0.01, q=0.001 |
|---|---|---|---|---|
| - | - | 74.68±3.19 | 82.71±1.86 | 92.08±1.13 |
| p=0.001, q=0.005 | 79.64±2.11 | 60.84±0.64 | 62.08±0.59 | 81.38±1.02 |
| p=0.001, q=0.003 | 70.08±1.71 | 59.72±2.01 | 61.58±1.08 | 76.60±0.98 |
| p=0.001, q=0.002 | 62.08±3.04 | 65.92±1.95 | 69.42±1.03 | 74.16±1.09 |

## D.3 Proof details of linear seperability within the same pattern

In this section, we provide detailed proof of the improved linear separability on aggregated features where nodes from different classes follow the same structural pattern. The following theoretical results indicate that it is more difficult to have improved linear separability when nodes follow different structural patterns. **Notably, the following proof is derived based on [13]. For completeness, we show the previous proof as follows, the difference is that we consider a more complicated CSBM-S model with both homophilic and heterophilic patterns rather than the simple CSBM model only describing either homophilic or heterophilic pattern.**

**Lemma 2** (Linear separability on nodes with the same structural patterns ). *Considering mean aggregated features are from the same structural pattern $\mathbf{f}_i^{(j)}$, for $i \in \{1, 2\}$. For any node $i$, the largest-margin linear classifier on $\mathbf{f}_i^{(j)}$ will have a lower probability to misclassify than $\mathbf{x}_i$, when $d_i > \frac{(p^{(1)} + q^{(1)})^2}{(p^{(1)} - q^{(1)})^2}$*

Notably, we only show the case $p^{(1)} > q^{(1)}$ in Appendix D.1 for simplicity. It can be easily extend to $p^{(2)} < q^{(2)}$. Proof details can be found as follows.

Consider the vanilla mean aggregation as $\mathbf{F} = \mathbf{D}^{-1}\mathbf{A}\mathbf{X}$, the aggregated features with the same structure patterns follow Gaussian distributions:

$$\mathbf{f}_i^{(j)} \sim N\left(\frac{p^{(j)}\boldsymbol{\mu}_1 + q^{(j)}\boldsymbol{\mu}_2}{p^{(j)} + q^{(j)}}, \frac{\mathbf{I}}{\sqrt{d_i}}\right), \text{ for } i \in \mathcal{C}_1^{(j)}; \mathbf{f}_i^{(j)} \sim N\left(\frac{q^{(j)}\boldsymbol{\mu}_1 + p^{(j)}\boldsymbol{\mu}_2}{p^{(j)} + q^{(j)}}, \frac{\mathbf{I}}{\sqrt{d_i}}\right), \text{ for } i \in \mathcal{C}_2^{(j)} \tag{8}$$

Then we denote the expectation of the original node features $\mathbf{x}$ in two classes as $\mathbb{E}_{c_1}[\mathbf{x}_i]$ and $\mathbb{E}_{c_2}[\mathbf{x}_i]$. Similarly, we denote the expectation of the aggregated features as $\mathbb{E}_{c_1}[\mathbf{f}_i]$ and $\mathbb{E}_{c_2}[\mathbf{f}_i]$.

**Proposition 2.** $(\mathbb{E}_{c_1}[\mathbf{x}_i], \mathbb{E}_{c_2}[\mathbf{x}_i])$ *and* $(\mathbb{E}_{c_1}[\mathbf{f}_i], \mathbb{E}_{c_2}[\mathbf{f}_i])$ *share the same middle point.* $\mathbb{E}_{c_1}[\mathbf{x}_i] - \mathbb{E}_{c_2}[\mathbf{x}_i]$ *and* $\mathbb{E}_{c_1}[\mathbf{f}_i] - \mathbb{E}_{c_2}[\mathbf{f}_i]$ *share the same direction.*

The proposition can be calculated through direct calculations. If we consider the feature distributions of two classes, we can observe that both $\mathbf{x}$ and $\mathbf{f}$ exhibit a systematic relationship. As a consequence, we can determine the optimal linear classifier for both $\mathbf{x}_i$ and $\mathbf{f}_i$. We then define decision boundary as the hyperplane that is orthogonal to the direction $\mathbf{w} = \frac{\boldsymbol{\mu}_1 - \boldsymbol{\mu}_2}{||\boldsymbol{\mu}_1 - \boldsymbol{\mu}_2||}$ and passes through the middle point $\mathbf{m} = (\boldsymbol{\mu}_1 + \boldsymbol{\mu}_2)/2$ as:

$$\mathcal{P} = \left\{\mathbf{x} \mid \mathbf{w}^\top \mathbf{x} - \mathbf{w}^\top (\boldsymbol{\mu}_1 + \boldsymbol{\mu}_2)/2\right\} \tag{9}$$

We then show the proof details on when $\mathbf{f}$ has a lower mis-classified probability than $\mathbf{x}$ with the same decision boundary, indicating better linear separability.

**Proof.** We provide the proof only for nodes belonging to class $c_0$, as the case for nodes from class $c_1$ is symmetric and the proof follows the same logic. For a node $i$ in $\mathcal{C}_1$, We have the following:

$$\begin{aligned}\mathbb{P}(\mathbf{x}_i \text{ is mis-classified }) &= \mathbb{P}(\mathbf{w}^\top \mathbf{x}_i + \mathbf{b} \leq 0) \text{ for } i \in \mathcal{C}_1 \\ \mathbb{P}(\mathbf{f}_i \text{ is mis-classified }) &= \mathbb{P}(\mathbf{w}^\top \mathbf{f}_i + \mathbf{b} \leq 0) \text{ for } i \in \mathcal{C}_1,\end{aligned} \tag{10}$$

we can then scale the decision boundary without changing the original meaning. Then we can have $\mathbb{P}(\mathbf{w}^\top \mathbf{f}_i + \mathbf{b} \leq 0) = \mathbb{P}(\sqrt{d_i}\mathbf{w}^\top \mathbf{f}_i + \sqrt{d_i}\mathbf{b} \leq 0)$ We denote the scaled version of $\mathbf{f}_i$ as $\mathbf{f}_i' = \sqrt{d_i}\mathbf{f}_i$. Then, $\mathbf{f}_i'$ follows:

$$\mathbf{f}_i' \sim N\left(\frac{\sqrt{d_i}(p^{(j)}\boldsymbol{\mu}_1 + q^{(j)}\boldsymbol{\mu}_2)}{p^{(j)} + q^{(j)}}, \mathbf{I}\right), \text{ for } i \in \mathcal{C}_1 \tag{11}$$

Given the scale in Eq. equation 11, the decision boundary for $\mathbf{f}_i'$ is consequently shifted to $\mathbf{w}^\top \mathbf{f}' + \sqrt{d_i}\mathbf{b} = 0$. Now, considering that $\mathbf{x}_i$ and $\mathbf{f}_i'$ have the same variance, we can compare the mis-classification probabilities by merely comparing the distance from their expected values to their corresponding decision boundaries. Specifically, the distances can be described as follows:

$$\begin{aligned}\text{dis}_{\mathbf{x}_i} &= \frac{\|\boldsymbol{\mu}_1 - \boldsymbol{\mu}_2\|_2}{2} \\ \text{dis}_{\mathbf{f}_i'} &= \frac{\sqrt{d_i}|p^{(j)} - q^{(j)}|}{p^{(j)} + q^{(j)}} \cdot \frac{\|\boldsymbol{\mu}_1 - \boldsymbol{\mu}_2\|_2}{2}.\end{aligned} \tag{12}$$

The larger the distance from the expectation to the decision boundary indicates a smaller the mis-classification probability. Therefore, when $\text{dis}_{\mathbf{f}_i'} > \text{dis}_{\mathbf{x}_i}$, $\mathbf{f}_i'$ has a lower probability of being misclassified than $\mathbf{x}_i$. By comparing the two distances, we conclude that when $d_i > \frac{(p^{(j)}+q^{(j)})^2}{(p^{(j)}-q^{(j)})^2}$, the mean aggregated feature $\mathbf{f}_i'$ exhibits a lower probability of misclassification compared to the original node feature $\mathbf{x}_i$.

$$\mathbb{P}(\mathbf{f}_i \text{ is mis-classified }) < \mathbb{P}(\mathbf{x}_i \text{ is mis-classified }) \text{ if } d_i > \left(\frac{p^{(j)} + q^{(j)}}{p^{(j)} - q^{(j)}}\right)^2 \tag{13}$$

which completes the proof.

## D.4 Proof details of linear separability between different structural patterns

In this section, we provide detailed proof of the improved linear separability on aggregated features where nodes from different classes follow different structural patterns. The following theoretical results indicate that it is more difficult to have improved linear separability when nodes follow different structural patterns. **Notably, the following proof is derived based on [13]. For completeness, we show the previous proof as follows, the difference is that we consider a more complicated CSBM-S model with both homophilic and heterophilic patterns rather than the simple CSBM model only describing either homophilic or heterophilic pattern**

**Lemma 3** (Linear separability on nodes with different structural patterns). *Consider features are from different structural patterns, where $\mathbf{f}_i^{(1)}$ for $i \in \mathcal{C}_1$ and $\mathbf{f}_i^{(2)}$ for $i \in \mathcal{C}_2$. For any node $i$, the largest-margin linear classifier will have a lower probability to misclassify $\mathbf{f}_i^{(1)}$ for $i \in \mathcal{C}_1$ and $\mathbf{f}_i^{(2)}$ for $i \in \mathcal{C}_2$ than $\mathbf{x}_i$ when $d_i > \frac{(p^{(1)}+q^{(1)})^2}{(p^{(1)}-q^{(2)})^2}$*

In the following discussion, we only focus on the scenario where nodes in class $c_1$ are in a homophilic pattern with $p^{(1)} > q^{(1)}$, and nodes in class $c_2$ are in a heterophilic pattern with $p^{(2)} < q^{(2)}$. The other scenario is symmetric and the proof follows the same logic.

Consider the vanilla mean aggregation as $\mathbf{F} = \mathbf{D}^{-1}\mathbf{A}\mathbf{X}$, the aggregated features with different structure patterns follow Gaussian distributions:

$$\mathbf{f}_i^{(1)} \sim N\left(\frac{p^{(1)}\boldsymbol{\mu}_1 + q^{(1)}\boldsymbol{\mu}_2}{p^{(1)} + q^{(1)}}, \frac{\mathbf{I}}{\sqrt{d_i}}\right), \text{ for } i \in \mathcal{C}_1; \mathbf{f}_i^{(2)} \sim N\left(\frac{q^{(2)}\boldsymbol{\mu}_1 + p^{(2)}\boldsymbol{\mu}_2}{p^{(2)} + q^{(2)}}, \frac{\mathbf{I}}{\sqrt{d_i}}\right), \text{ for } i \in \mathcal{C}_2^{(2)}$$
(14)

Similar to the notation in Proposition 2, we denote the expectation of the original node features $\mathbf{x}$ in two classes as $\mathbb{E}_{c_1}[\mathbf{x}_i]$ and $\mathbb{E}_{c_2}[\mathbf{x}_i]$. Similarly, we denote the expectation of the aggregated features as $\mathbb{E}_{c_1}[\mathbf{f}_i]$ and $\mathbb{E}_{c_2}[\mathbf{f}_i]$.

**Proposition 3.** $(\mathbb{E}_{c_1}[\mathbf{x}_i], \mathbb{E}_{c_2}[\mathbf{x}_i])$ *and* $(\mathbb{E}_{c_1}[\mathbf{f}_i], \mathbb{E}_{c_2}[\mathbf{f}_i])$ *share the same middle point. When satisfying* $p^{(1)} > q^{(2)}$, $\mathbb{E}_{c_1}[\mathbf{x}_i] - \mathbb{E}_{c_2}[\mathbf{x}_i]$ *and* $\mathbb{E}_{c_1}[\mathbf{f}_i] - \mathbb{E}_{c_2}[\mathbf{f}_i]$ *share the opposite direction. Specifically, the middle point* $\mathbf{m}$ *and the shared direction* $\mathbf{w}$ *are as follows:* $\mathbf{m} = (\boldsymbol{\mu}_1 + \boldsymbol{\mu}_2)/2$, *and* $\mathbf{w} = (\boldsymbol{\mu}_1 - \boldsymbol{\mu}_2)/\|\boldsymbol{\mu}_1 - \boldsymbol{\mu}_2\|_2$. *When satisfying* $q^{(2)} > p^{(1)}$, $\mathbb{E}_{c_1}[\mathbf{x}_i] - \mathbb{E}_{c_2}[\mathbf{x}_i]$ *and* $\mathbb{E}_{c_1}[\mathbf{h}_i] - \mathbb{E}_{c_2}[\mathbf{h}_i]$ *are in same directions. It indicates that the largest-margin linear model still shares the same discriminative boundary which flips the prediction on class* $c_1$ *and* $c_2$.

Notably, with the assumption $p^{(1)} + q^{(1)} = p^{(2)} + q^{(2)}$, the linear classifier can be generally to two cases according to whether the decision boundary direction flips: (1) The decision boundary direction unchanged with $q^{(2)} > p^{(1)} > p^{(2)} > q^{(1)}$ (2) The decision boundary direction flipped with $p^{(1)} > q^{(2)} > q^{(1)} > p^{(2)}$. Notably, the two cases are symmetric with the same conclusion and proof logic. We will focus on the unchanged case with $q^{(2)} > p^{(1)} > p^{(2)} > q^{(1)}$ .

The proposition can be calculated through direct calculations. If we consider the feature distributions of two classes, we can observe that both $\mathbf{x}$ and $\mathbf{f}$ exhibit a systematic relationship. As a consequence, we can determine the optimal linear classifier for both $\mathbf{x}_i$ and $\mathbf{f}_i$. We then define decision boundary as the hyperplane that is orthogonal to the direction $\mathbf{w} = \frac{\boldsymbol{\mu}_1 - \boldsymbol{\mu}_2}{\|\boldsymbol{\mu}_1 - \boldsymbol{\mu}_2\|}$ and passes through the middle point $\mathbf{m} = (\boldsymbol{\mu}_1 + \boldsymbol{\mu}_2)/2$ as:

$$\mathcal{P} = \left\{\mathbf{x} \mid \mathbf{w}^\top\mathbf{x} - \mathbf{w}^\top(\boldsymbol{\mu}_1 + \boldsymbol{\mu}_2)/2\right\}$$
(15)

We then show the proof details on when $\mathbf{f}$ has a lower mis-classified probability than $\mathbf{x}$ with the same decision boundary, indicating better linear separability.

**Proof.** We provide the proof only for nodes belonging to class $c_0$, as the case for nodes from class $c_1$ is symmetric and the proof follows the same logic. For a node $i$ in $\mathcal{C}_1$, We have the following:

$$\begin{aligned}\mathbb{P}\left(\mathbf{x}_i \text{ is mis-classified }\right) &= \mathbb{P}\left(\mathbf{w}^\top\mathbf{x}_i + \mathbf{b} \le 0\right) \text{ for } i \in \mathcal{C}_1 \\ \mathbb{P}\left(\mathbf{f}_i \text{ is mis-classified }\right) &= \mathbb{P}\left(\mathbf{w}^\top\mathbf{f}_i + \mathbf{b} \le 0\right) \text{ for } i \in \mathcal{C}_1,\end{aligned}$$
(16)

we can then scale the decision boundary without changing the original meaning. Then we can have $\mathbb{P}\left(\mathbf{w}^\top\mathbf{f}_i + \mathbf{b} \le 0\right) = \mathbb{P}\left(\sqrt{d_i}\mathbf{w}^\top\mathbf{f}_i + \sqrt{d_i}\mathbf{b} \le 0\right)$ We denote the scaled version of $\mathbf{f}_i$ as $\mathbf{f}_i' = \sqrt{d_i}\mathbf{f}_i$. Then, $\mathbf{f}_i'$ follows:

$$\mathbf{f}_i' \sim N\left(\frac{\sqrt{d_i}\left(p^{(1)}\boldsymbol{\mu}_1 + q^{(1)}\boldsymbol{\mu}_2\right)}{p^{(1)} + q^{(1)}}, \mathbf{I}\right), \text{ for } i \in \mathcal{C}_1; \mathbf{f}_i' \sim N\left(\frac{\sqrt{d_i}\left(p^{(2)}\boldsymbol{\mu}_1 + q^{(2)}\boldsymbol{\mu}_2\right)}{p^{(2)} + q^{(2)}}, \mathbf{I}\right), \text{ for } i \in \mathcal{C}_2$$

$$(17)$$

Given the scale in Eq. equation 17, the decision boundary for $\mathbf{f}_i'$ is consequently shifted to $\mathbf{w}^\top \mathbf{f}' + \sqrt{d_i}\mathbf{b} = 0$. Now, considering that $\mathbf{x}_i$ and $\mathbf{f}_i'$ have the same variance, we can compare the misclassification probabilities by merely comparing the distance from their expected values to their corresponding decision boundaries. Specifically, the distances can be described as follows:

$$\text{dis}_{\mathbf{x}_i} = \frac{\|\boldsymbol{\mu}_1 - \boldsymbol{\mu}_2\|_2}{2}$$

$$\text{dis}_{\mathbf{f}_i'} = \frac{\sqrt{d_i}|p^{(2)} - q^{(1)}|}{p^{(1)} + q^{(1)}} \cdot \frac{\|\boldsymbol{\mu}_1 - \boldsymbol{\mu}_2\|_2}{2} = \frac{\sqrt{d_i}|p^{(1)} - q^{(2)}|}{p^{(1)} + q^{(1)}} \cdot \frac{\|\boldsymbol{\mu}_1 - \boldsymbol{\mu}_2\|_2}{2}$$

$$(18)$$

The larger the distance from the expectation to the decision boundary indicates a smaller the misclassification probability. Notably, $q^{(1)} - p^{(2)} = p^{(1)} - q^{(2)}$ with the assumptin $p^{(1)} + p^{(2)} = q^{(1)} + q^{(2)}$ Therefore, when $\text{dis}_{\mathbf{f}_i'} > \text{dis}_{\mathbf{x}_i}$, $\mathbf{f}_i'$ has a lower probability of being misclassified than $\mathbf{x}_i$. By comparing the two distances, we conclude that when $d_i > \frac{\left(p^{(1)} + q^{(1)}\right)^2}{\left(p^{(1)} - q^{(2)}\right)^2}$, the mean aggregated feature $\mathbf{f}_i'$ exhibits a lower probability of misclassification compared to the original node feature $\mathbf{x}_i$.

$$\mathbb{P}\left(\mathbf{f}_i \text{ is mis-classified }\right) < \mathbb{P}\left(\mathbf{x}_i \text{ is mis-classified }\right) \text{ if } d_i > \left(\frac{p^{(1)} + q^{(1)}}{p^{(1)} - q^{(2)}}\right)^2 \tag{19}$$

which completes the proof.

**A comparison between linear separability within classes and between classes**   Notice that the only difference with the one in the same pattern one is that the denominator is $|p^{(1)} - q^{(2)}|^2$, smaller than the $|p^{(1)} - q^{(1)}|^2$ since $q^{(1)} > q^{(2)}$. It indicates that only nodes with higher degrees. For instance, when $p^{(1)} = 0.9, q^{(1)} = 0.1$ and $p^{(2)} = 0.2, q^{(2)} = 0.8$, the aggregated features can only show better linear separability with $d_i > 100$ when nodes are from different structural patterns. The above condition can be hardly met in real-world scenarios. Based on our analysis, we can draw the conclusion that when nodes are in the same pattern, aggregation can show improved linear separability on $(p^{(1)} + q^{(1)})^2/|p^{(1)} - q^{(1)}|^2 < d_i < (p^{(1)} + q^{(1)})^2/|p^{(1)} - q^{(2)}|^2$ while the ones in the different patterns cannot. It indicates that aggregation can help more when nodes are in the same pattern, however, show limitation when nodes are from different patterns.

# E   Proof details of the conditional probability difference for nodes with the same feature but different structural patterns

In section 3.1, we examine the discrepancy between nodes 1 and 2 with the same aggregated feature $\mathbf{f}_1 = \mathbf{f}_2$ but different structural patterns. Typically, we examine the discrepancy with the probability difference of nodes 1 and 2 in class $c_1$, denoted as $|p_1(y_u = c_1|\mathbf{f}_u) - p_2(y_v = c_1|\mathbf{f}_v)|$. $p_1$ and $p_2$ are the conditional probabilities of node labels in $c$ given the feature $\mathbf{f}$ for nodes on homophilic and heterophilic structural patterns, respectively. The lemma is shown as follows:

**Lemma 1.** *With assumptions (1) A balance class distribution with $\mathbf{P}(\mathbf{Y} = 1) = \mathbf{P}(\mathbf{Y} = 0)$ and (2) aggregated feature distribution shares the same variance $\sigma\mathbf{I}$. When nodes $u$ and $v$ have the same aggregated features $\mathbf{f}_u = \mathbf{f}_v$ but different structural patterns, $(p^{(1)}, q^{(1)})$ and $(p^{(2)}, q^{(2)})$, we can have:*

$$|\mathbf{P}_1(y_u = c_1|\mathbf{f}_u) - \mathbf{P}_2(y_v = c_1|\mathbf{f}_v)| \leq \frac{\rho^2}{\sqrt{2\pi}\sigma}|h_u - h_v| \tag{20}$$

$\rho = \|\boldsymbol{\mu}_1 - \boldsymbol{\mu}_2\|$ is the original feature separability, independent with structure. $\mathbf{P}_1$ and $\mathbf{P}_2$ are the conditional probability of $y = c_1$ given the feature $\mathbf{f}$ on structural patterns $(p^{(1)}, q^{(1)})$ and $(p^{(2)}, q^{(2)})$,

respectively. Lemma 1 implies that nodes with a small homophily ratio difference $|h_1 - h_2|$ are likely to share the same class, and vice versa.

We first remind the assumptions of the CSBM-S model. There are two assumptions on the models: (1) Nodes from different components share the same feature distribution. In other word, node features within the same class are sampled from the same Gaussian distribution, regardless of different structural patterns. (2) Nodes from different components share similar degree distribution with $p^{(1)} + q^{(1)} = p^{(2)} + q^{(2)}$. Notably, our conclusions are still valid without the above assumptions. Those assumptions are not strictly necessary but employed for the elegant expression.

Based on the neighborhood distributions, the mean aggregated features $\mathbf{F} = \mathbf{D}^{-1}\mathbf{A}\mathbf{X}$ obtained follow Gaussian distributions on both homophilic and heterophilic node subgroups.

$$\mathbf{f}_i^{(j)} \sim N\left(\frac{p^{(j)}\boldsymbol{\mu}_1 + q^{(j)}\boldsymbol{\mu}_2}{p^{(j)} + q^{(j)}}, \sigma\mathbf{I}\right), \text{for } i \in \mathcal{C}_1^{(j)}; \mathbf{f}_i^{(j)} \sim N\left(\frac{q^{(j)}\boldsymbol{\mu}_1 + p^{(j)}\boldsymbol{\mu}_2}{p^{(j)} + q^{(j)}}, \sigma\mathbf{I}\right), \text{for } i \in \mathcal{C}_2^{(j)}$$
(21)

Where $\mathcal{C}_i^{(j)}$ is the node subgroups with structural pattern with $(p^{(j)}, q^{(j)})$ in label $i$. Notably, there are typically homophilic pattern with $p^{(1)} > q^{(1)}$ and heterophilic pattern with $p^{(2)} < q^{(2)}$. To simply the assumption for an elegant expression, we denote the aggregated feature mean, $\frac{p^{(j)}\boldsymbol{\mu}_1 + q^{(j)}\boldsymbol{\mu}_2}{p^{(j)} + q^{(j)}}$ as $\boldsymbol{\mu}'_{ij}$, where $i$ and $j$ correspond to the class and the structural pattern, respectively. For instance, $\boldsymbol{\mu}'_{11}$ represents the mean of nodes in class $c_1$ with the homophilic pattern. We then show the proof details as follows:

*Proof.* The conditional probability of class $c_1$ given the aggregated feature $f$, $\mathbf{P}_1(\mathbf{y} = c_1|\mathbf{f}_1)$ can be derived with the Bayes theorem:

$$
\begin{aligned}
\mathbf{P}_1(\mathbf{y}_u = c_1|\mathbf{f}_u) &= \frac{\mathbf{P}_1(\mathbf{f}_u|\mathbf{y}_u = c_1)\mathbf{P}(\mathbf{y}_u = c_1)}{\mathbf{P}_1(\mathbf{f}_u|\mathbf{y}_u = c_1)\mathbf{P}(\mathbf{y}_u = c_1) + \mathbf{P}_1(\mathbf{f}_u|\mathbf{y}_u = c_2)\mathbf{P}(\mathbf{y}_u = c_2)} \\
&\overset{(a)}{=} \frac{\mathbf{P}_1(\mathbf{f}_u|\mathbf{y}_u = c_1)}{\mathbf{P}_1(\mathbf{f}_u|\mathbf{y}_u = c_1) + \mathbf{P}_1(\mathbf{f}_u|\mathbf{y}_u = c_2)} \\
&= \frac{\exp\left(\frac{(\mathbf{f}_u - \boldsymbol{\mu}'_{11})^2}{\sigma^2}\right)}{\exp\left(\frac{(\mathbf{f}_u - \boldsymbol{\mu}'_{11})^2}{\sigma^2}\right) + \exp\left(\frac{(\mathbf{f}_u - \boldsymbol{\mu}'_{12})^2}{\sigma^2}\right)}
\end{aligned}
$$
(22)

where (a) we utilize the assumption $\mathbf{P}(\mathbf{y} = c_1) = \mathbf{P}(\mathbf{y} = c_2)$. Notably, such assumption aims to simplify for an elegant expression and better understanding, which is not necessary in our proof.

Hence, we have:

$$
\begin{aligned}
&|\mathbf{P}_1(\mathbf{y}_u = c_1|\mathbf{f}_u) - \mathbf{P}_2(\mathbf{y}_v = c_1|\mathbf{f}_v)| \\
&= \left|\frac{\exp\left(-\frac{||\mathbf{f}_u - \boldsymbol{\mu}'_{11}||^2}{\sigma^2}\right)}{\exp\left(-\frac{||\mathbf{f}_u - \boldsymbol{\mu}'_{11}||^2}{\sigma^2}\right) + \exp\left(-\frac{||\mathbf{f}_u - \boldsymbol{\mu}'_{21}||^2}{\sigma^2}\right)} - \frac{\exp\left(-\frac{||\mathbf{f}_v - \boldsymbol{\mu}'_{12}||^2}{\sigma^2}\right)}{\exp\left(-\frac{||\mathbf{f}_v - \boldsymbol{\mu}'_{12}||^2}{\sigma^2}\right) + \exp\left(-\frac{||\mathbf{f}_v - \boldsymbol{\mu}'_{22}||^2}{\sigma^2}\right)}\right| \\
&= \frac{\left|\exp\left(-\frac{(\mathbf{f}_u - \boldsymbol{\mu}'_{11})^2}{\sigma^2}\right)\exp\left(-\frac{(\mathbf{f}_u - \boldsymbol{\mu}'_{22})^2}{\sigma^2}\right) - \exp\left(-\frac{(\mathbf{f}_v - \boldsymbol{\mu}'_{12})^2}{\sigma^2}\right)\exp\left(-\frac{(\mathbf{f}_u - \boldsymbol{\mu}'_{21})^2}{\sigma^2}\right)\right|}{\left[\exp\left(-\frac{||\mathbf{f}_u - \boldsymbol{\mu}'_{11}||^2}{\sigma^2}\right) + \exp\left(-\frac{||\mathbf{f}_u - \boldsymbol{\mu}'_{21}||^2}{\sigma^2}\right)\right]\left[\exp\left(-\frac{||\mathbf{f}_v - \boldsymbol{\mu}'_{12}||^2}{\sigma^2}\right) + \exp\left(-\frac{||\mathbf{f}_v - \boldsymbol{\mu}'_{22}||^2}{\sigma^2}\right)\right]}
\end{aligned}
$$
(23)

Notably, the denominator can be denoted as

$$\mathbf{m} = \left[\exp\left(-\frac{||\mathbf{f}_u - \boldsymbol{\mu}'_{11}||^2}{\sigma^2}\right) + \exp\left(-\frac{||\mathbf{f}_u - \boldsymbol{\mu}'_{21}||^2}{\sigma^2}\right)\right]\left[\exp\left(-\frac{||\mathbf{f}_v - \boldsymbol{\mu}'_{12}||^2}{\sigma^2}\right) + \exp\left(-\frac{||\mathbf{f}_v - \boldsymbol{\mu}'_{22}||^2}{\sigma^2}\right)\right]$$
(24)

where each exponential component is in $\exp\left(-\frac{||\mathbf{f}_i - \boldsymbol{\mu}'_{ij}||^2}{\sigma^2}\right) \in [0, 1]$. Therefore, the denominator $\mathbf{m}$ is in the range $[0, 4]$. We then denote $\mathbf{m} = \exp(-A)$, where $A$ is a constant, correspondingly. Hence,

we can have:

$$|\mathbf{P}_1(\mathbf{y}_u = c_1|\mathbf{f}_u) - \mathbf{P}_2(\mathbf{y}_v = c_1|\mathbf{f}_v)|$$

$$= \frac{\left| \exp\left(-\frac{(\mathbf{f}_u - \boldsymbol{\mu}'_{11})^2}{\sigma^2}\right) \exp\left(-\frac{(\mathbf{f}_v - \boldsymbol{\mu}'_{22})^2}{\sigma^2}\right) - \exp\left(-\frac{(\mathbf{f}_u - \boldsymbol{\mu}'_{12})^2}{\sigma^2}\right) \exp\left(-\frac{(\mathbf{f}_v - \boldsymbol{\mu}'_{21})^2}{\sigma_2^2}\right) \right|}{\exp(-A)}$$

$$= \left| \exp\left(-\frac{(\mathbf{f}_u - \boldsymbol{\mu}'_{11})^2 + (\mathbf{f}_v - \boldsymbol{\mu}'_{22})^2}{\sigma^2} - A\right) - \exp\left(-\frac{(\mathbf{f}_u - \boldsymbol{\mu}'_{12})^2 + (\mathbf{f}_2 - \boldsymbol{\mu}'_{21})^2}{\sigma^2} - A\right) \right|$$

$$\underset{(a)}{\leq} \frac{1}{\sigma^2} \left| \left[(\mathbf{f}_u - \boldsymbol{\mu}'_{11})^2 - (\mathbf{f}_v - \boldsymbol{\mu}'_{12})^2\right] + \left[(\mathbf{f}_u - \boldsymbol{\mu}'_{21})^2 - (\mathbf{f}_v - \boldsymbol{\mu}_{22})^2\right] \right|$$

$$\underset{(b)}{=} \frac{1}{\sigma^2} \left| \left[ \left(\mathbf{f}_u - \frac{p^{(1)}\boldsymbol{\mu}_1 + q^{(1)}\boldsymbol{\mu}_2}{p^{(1)} + q^{(1)}}\right)^2 - \left(\mathbf{f}_v - \frac{p^{(2)}\boldsymbol{\mu}_1 + q^{(2)}\boldsymbol{\mu}_2}{p^{(2)} + q^{(2)}}\right)^2 \right] \right.$$

$$\left. + \left[ \left(\mathbf{f}_u - \frac{p^{(1)}\boldsymbol{\mu}_2 + q^{(1)}\boldsymbol{\mu}_1}{p^{(1)} + q^{(1)}}\right)^2 - \left(\mathbf{f}_v - \frac{p^{(2)}\boldsymbol{\mu}_2 + q^{(2)}\boldsymbol{\mu}_1}{p^{(2)} + q^{(2)}}\right)^2 \right] \right|$$

$$\leq \frac{\|\boldsymbol{\mu}_1 - \boldsymbol{\mu}_2\|}{\sqrt{2\pi}\sigma}\left(\|\mathbf{f}_u - \mathbf{f}_v\| + \left|\frac{p^{(1)}}{p^{(1)} + q^{(1)}} - \frac{p^{(2)}}{p^{(2)} + q^{(2)}}\right| \cdot \|\boldsymbol{\mu}_1 - \boldsymbol{\mu}_2\|\right)$$

$$\underset{(c)}{=} \frac{1}{\sqrt{2\pi}\sigma}|h_1 - h_2| \cdot \rho^2$$

$$(25)$$

$\rho = \|\boldsymbol{\mu}_1 - \boldsymbol{\mu}_2\|$ is the original feature separability, independent with structure. $|h_1 - h_2|$ is the homophily ratio difference between node 1 and 2. We explain the key steps as follows: **(a)** is derived from Lagrange's mean value theorem. From the Lagrange mean value theorem we have,

$$\exp(-x) - \exp(-y) = -\exp(-\xi)(y - x) \tag{26}$$

where $\xi$ is between $x$ and $y$. Therefore,

$$|\exp(-x) - \exp(-y)| = \exp(-\xi)|y - x| \leq |x - y| \tag{27}$$

since $\exp(-\xi) \leq 1$.

Knowing that $|\mathbf{P}_1(\mathbf{y}_u = c_1|\mathbf{f}_u) - \mathbf{P}_2(\mathbf{y}_v = c_1|\mathbf{f}_v)| < 1$, we can easily get

$$\frac{\exp\left(-\frac{(\mathbf{f}_u - \boldsymbol{\mu}'_{11})^2}{\sigma^2}\right) \exp\left(-\frac{(\mathbf{f}_v - \boldsymbol{\mu}'_{22})^2}{\sigma^2}\right) - \exp(B)}{\exp(-A)} < 1 \tag{28}$$

where $\exp(B) > 0$, we can get $\frac{(\mathbf{f}_u - \boldsymbol{\mu}'_{11})^2 + (\mathbf{f}_u - \boldsymbol{\mu}'_{22})^2}{\sigma^2} > A$. Similarly, with $|\mathbf{P}_1(\mathbf{y}_u = c_1|\mathbf{f}_u) - \mathbf{P}_2(\mathbf{y}_v = c_1|\mathbf{f}_v)| > 0$, we can get $\frac{(\mathbf{f}_2 - \boldsymbol{\mu}'_{12})^2 + (\mathbf{f}_2 - \boldsymbol{\mu}'_{21})^2}{\sigma^2} > A$.

In conclusion, we have:

$$\begin{aligned} A - \frac{(\mathbf{f}_2 - \boldsymbol{\mu}'_{12})^2 + (\mathbf{f}_2 - \boldsymbol{\mu}'_{21})^2}{\sigma^2} &< 0 \\ A - \frac{(\mathbf{f}_1 - \boldsymbol{\mu}'_{11})^2 + (\mathbf{f}_1 - \boldsymbol{\mu}'_{22})^2}{\sigma^2} &< 0 \end{aligned} \tag{29}$$

Let $x = A - \frac{(\mathbf{f}_2 - \boldsymbol{\mu}'_{12})^2 + (\mathbf{f}_2 - \boldsymbol{\mu}'_{21})^2}{\sigma^2}$ and $y = A - \frac{(\mathbf{f}_1 - \boldsymbol{\mu}'_{11})^2 + (\mathbf{f}_1 - \boldsymbol{\mu}'_{22})^2}{\sigma^2}$ in equation 27. The proof is complete.

**(b)** we utilize $\boldsymbol{\mu}'_{i1} = \frac{p^{(i)}\boldsymbol{\mu}_1 + q^{(i)}\boldsymbol{\mu}_2}{p^{(i)} + q^{(i)}}$ and $\boldsymbol{\mu}'_{i2} = \frac{p^{(i)}\boldsymbol{\mu}_2 + q^{(i)}\boldsymbol{\mu}_1}{p^{(i)} + q^{(i)}}$.

**(c)** we utilize $\mathbf{f}_u = \mathbf{f}_v$ and $h_i = \frac{p^{(i)}}{p^{(i)} + q^{(i)}}$. Notice that, (c) step can be easily generally to the case that $\mathbf{f}_u \neq \mathbf{f}_v$ with $\|\mathbf{f}_u - \mathbf{f}_v\| \leq \epsilon$. We can easily proof with the same logis. The lemma is shown as follows:

**Lemma 2.** *With assumptions (1) A balance class distribution with $\mathbf{P}(\mathbf{Y} = 1) = \mathbf{P}(\mathbf{Y} = 0)$ and (2) aggregated feature distribution shares the same variance $\sigma I$. When nodes have the same aggregated features $\|\mathbf{f}_u - \mathbf{f}_v\| \leq \epsilon$ but different structural patterns, $(p^{(1)}, q^{(1)})$ and $(p^{(2)}, q^{(2)})$, we can have:*

$$|\mathbf{P}_1(y_u = c_1|\mathbf{f}_u) - \mathbf{P}_2(y_v = c_1|\mathbf{f}_v)| \leq \frac{\rho}{\sqrt{2\pi}\sigma}(\epsilon_m + |h_1 - h_2| \cdot \rho) \tag{30}$$

$\rho = \|\boldsymbol{\mu}_1 - \boldsymbol{\mu}_2\|$ is the original feature separability, independent with structure. Lemma 2 implies that nodes with a small homophily ratio difference $|h_1 - h_2|$ are likely to share the same class, and vice versa. $\qquad\square$

# F  Proof details of PAC-Bayes Bound

In this section, we provide background knowledge, assumptions, and proof details on the PAC-Bayes analysis. Our PAC-Bayes bound is derived based on [46], which is the first PAC-Bayesian analysis for GNN on the semi-supervised node classification task. A details comparison between our bound and [46] can be found in Appendix A.

## F.1  Background Knowledge on PAC-Bayes Analysis

**PAC-Bayes Analysis** The Probably Approximately Correct-Bayesian (PAC-Bayes) analysis [48] is a powerful theoretical framework utilizing Bayesian learning principles for analyzing the generalization ability of machine learning models. Typically, PAC-Bayesian Analysis is to connect and bound the difference between a training error of a machine learning model with its expected generalization error. Various PAC-Bayes bounds [49, 95, 50, 51] are proposed for advanced Deep Learning models in recent years. Such theoretical guidance shows practical success in many real-world applications including pretraining model [96], medical image [97], and robustness [98].

**PAC-Bayes analysis on Graph Neural Networks** [90] is the first PAC-Bayesian generalization bound for GNNs. Nonetheless, it focuses on the i.i.d. graph classification task, which is different from the non-i.i.d. node classification task. [46] provides the first PAC-Bayesian generalization bound for GNN on semi-supervised node classification task. Our following theorem is developed based on it and generalized into more generalized scenarios. A more detailed discussion on the difference of our work and [46] can be found in the Appendix A.

## F.2  PAC-Bayesian Analysis on Subgroup Generalization bound of Deterministic classifier

In this section, we provide a more detailed discussion on the PAC-Bayesian analysis with the semi-supervised subgroup Generalization bound of Deterministic classifiers proposed in [46], which serves as the basis of proof. Notably, in the main content, we denote the training node subgroup as $V_{\text{tr}}$. As we focus on the relationship between the train nodes and test subgroups $V_m$ with $0 < m \leq M$, we re-denote the train nodes as $V_0$ . The PAC-Bayesian analysis on subgroup Generalization bound of Deterministic classifiers, focusing on the deterministic classifiers $\tilde{h}$, a classifier drawn in the predictor family $\mathcal{H}$, learned on training nodes. The primary objective of the PAC-Bayesian analysis is to derive bounds on the generalization gap between the empirical margin loss of $\tilde{h}$ denoting as $\widehat{\mathcal{L}}_0^\gamma(\tilde{h})$ on the train node subgroup $V_0$ and the corresponding expected margin loss $\mathcal{L}_m^0(\tilde{h})$ on the test node subgroup $V_m$. The empirical marginal loss $\widehat{\mathcal{L}}_0^\gamma(\tilde{h})$ on the train node subgroup $V_0$ is defined as:

$$\widehat{\mathcal{L}}_0^\gamma(\tilde{h}) := \frac{1}{N_0} \sum_{i \in V_0} \mathbb{1}\left[ \tilde{h}_i(X,G)[y_i] \leq \gamma + \max_{k \neq y_i} \tilde{h}_i(X,G)[k] \right], \tag{31}$$

where $\mathbb{1}[\cdot]$ is the indicator function and $\gamma$ is the margin. $V_0$ is the training node set. $N_0 = |V_0|$ is the number of the training node. $y_i$ is the label corresponding to node $i \in V_0$. $\tilde{h}_i(X,G) \in \mathbb{R}^K$ is the predictor output, where $\tilde{h}_i(X,G)[k]$ refers to the prediction probability of class $k$ on sample $i \in V_0$.

The expected margin loss is the expectation of $\mathcal{L}_m^0(\tilde{h})$ on the test node subgroup $V_m$ of the corresponding empirical loss $\widehat{\mathcal{L}}_m^0(\tilde{h})$.

$$\mathcal{L}_m^0(\tilde{h}) := \mathbb{E}_{y_i \sim \Pr(y|g_i(X,G)), i \in V_m} \widehat{\mathcal{L}}_m^0(\tilde{h}). \tag{32}$$

where $\Pr(y \mid g_i(X,G))$ is the conditional distribution. $g(X,G) : \mathbb{R}^{N \times D} \times \mathcal{G}_N \to \mathbb{R}^{N \times D'}$ is the aggregation function, typically, $g_i(X,G) = \frac{1}{|\mathcal{N}(i)|} \sum_{j \in \mathcal{N}(i)} X_j$ as default. $\mathcal{G}_n$ is the space for all undirected graphs with $n$ nodes.

To bound the generalization gap between the expected margin loss $\mathcal{L}_m^0(\tilde{h})$ on test subgroup $V_m$ and the empirical margin loss $\widehat{\mathcal{L}}_0^\gamma$ on train subgroup $V_0$. The theorem is shown as follows:

**Theorem 2** (Subgroup Generalization of Deterministic Classifiers [46]). *Let $\tilde{h}$ be any classifier in $\mathcal{H}$. For any $0 < m \leq M$, for any $\lambda > 0$ and $\gamma \geq 0$, for any "prior" distribution $P$ on $\mathcal{H}$ that is independent of the training data on $V_0$, with probability at least $1 - \delta$ over the sample of $y^0$, for any $Q$ on $\mathcal{H}$ such that $\mathrm{Pr}_{h \sim Q}\left(\max_{i \in V_0 \cup V_m} \|h_i(X, G) - \tilde{h}_i(X, G)\|_\infty < \frac{\gamma}{8}\right) > \frac{1}{2}$, we have*

$$\mathcal{L}_m^0(\tilde{h}) \leq \widehat{\mathcal{L}}_0^\gamma(\tilde{h}) + \frac{1}{\lambda}\left(\underbrace{2(D_{\mathrm{KL}}(Q\|P) + 1)}_{(a)} + \ln\frac{1}{\delta} + \frac{\lambda^2}{4N_{tr}} + D_{m,0}^{\gamma/2}(P; \lambda)\right) \tag{33}$$

The generalization bound is typically related to the following four terms: **(a)** $D_{\mathrm{KL}}(Q\|P) := \int \ln\frac{dQ}{dP}dQ$ is the KL-divergence between the learned (posterior) predictor distribution $Q$ and the predefined (prior) distribution $P$, independent of the training data. The generalization gap is bounded by the discrepancy between $P$ and $Q$. It is a general term in PAC-Bayes analysis considering as the model complexity measurement. **(2)** $\frac{\lambda^2}{4N_0}$ will vanish with larger number of training samples $N_0$ Notably, the above two common terms in PAC-Bayes analysis are not our focus. **(3)** $\ln\frac{1}{\delta}$ is the term related with the probability $1 - \delta$. **(4)** The expected loss Discrepancy between the training nodes $V_0$ and targeted test node subgroup $V_m$ is essential in our analysis, denoted as:

$$D_{m,0}^{\gamma/2}(P; \lambda) = \ln\mathbb{E}_{h \sim P}e^{\lambda\left(\mathcal{L}_m^{\gamma/4}(h) - \mathcal{L}_0^{\gamma/2}(h)\right)} \tag{34}$$

where a prior distribution $P$ over $\mathcal{H}$, $\gamma$ is the corresponding loss margin, $\lambda > 0$ is a parameter, reflects the concentration of the learning distribution $Q$. Intuitively speaking, the term $D_{m,0}^{\gamma/2}(P; \lambda)$ signifies the difference in expected loss between $V_m$ and $V_0$, evaluated in an expectation with respect to the prior distribution $P$.

The expected loss Discrepancy is typically for the non-i.i.d. semi-supervised setting, which trivially comes true in i.i.d. setting. In the i.i.d. case, all samples in $V_m$ and $V_0$ are i.i.d., where $\mathcal{L}_m^{\gamma/4}(h) = \mathcal{L}_0^{\gamma/4}(h) < \mathcal{L}_0^{\gamma/2}(h) \leq 0$ for any classifier $h$, leading to a trivial upper bound $D_{m,0}^{\gamma/2}(P; \lambda) \leq 0$. To provide a meaningful upper bound, we are required to essentially assume there exists the relationship between train node subgroup $V_0$ and test node subgroup $V_m$. Typically, we expect the expected margin loss $\mathcal{L}_m^{\gamma/4}(h)$ on $V_m$ is not much larger than $\mathcal{L}_0^{\gamma/2}(h)$ on $V_0$ when the number of samples becomes large. Taking a further step, to bound the difference $\mathcal{L}_m^{\gamma/4}(h) - \mathcal{L}_0^{\gamma/2}(h)$, we should find suitable assumptions and derive proof to bound on (1) the difference on $P(g(X, G))$ controlling with the feature distance between $V_0$ and $V_m$. (2) The differencce on $P(Y|g(X, G))$ controlling with the homophily ratio as shown in Lemma 1.

### F.3 Proof details of Expected Loss Discrepancy

To establish the generalization guarantee, it becomes necessary to provide an upper bound for the expected loss discrepancy, $D_{m,0}^\gamma(P; \lambda)$, which is the main focus in the proof. It emerges that certain assumptions about the data are required to derive a meaningful and useful upper bound. The assumptions are shown as follows. Notably, the assumptions 2 and 3 are adapted from [46]. **Remark: Notably, the following proof is derived based on [46]. For completeness, we show the previous proof as follows, the difference is that we consider a more complicated conditional probability $\mathbf{P}(y_i = k|g_i(X, G))$, correlated with the homophily ratio difference, rather than consider the conditional probability as $c$-Lipschitz continuous functions. In other words, our proof further strengthens the effect of the structure disparity.**

**Definition 1** (Generalized CSBM-S model ). *Each node subgroup $V_m$ follows the CSBM distribution $V_m \sim CSBM(\boldsymbol{\mu}_1, \boldsymbol{\mu}_2, p^{(i)}, q^{(i)})$, where different subgroups share the same class mean but different intra-class and inter-class probabilities $p^{(i)}$ and $q^{(i)}$. Moreover, node subgroups also share the same degree distribution as $p^{(i)} + q^{(i)} = p^{(j)} + q^{(j)}$.*

Instead of only considering one homophilic and one heterophilic pattern in CSBM-S model, the generalized CSBM-S model allows more diverse structural patterns with different levels of homophily.

**Assumption 1** (Data follows Generalized CSBM-S model assumption). *The graph data is generated from the Generalized CSBM-S model.*

**Assumption 2** (Equal-Sized and Disjoint Near Sets ). *For any $0 < m \leq M$, assume the near sets of each $i \in V_0$ with respect to $V_m$ are disjoint and have the same size $s_m \in \mathbb{N}^+$.*

Assumption 2 assumes that $V_m$ can be divided into equally sized partitions, each can be identified by the corresponding training samples. It assumes that test nodes are closely aligned with the respective training sample, while distant to the other training samples.

**Definition 2** (Distance To Training Set and Near Set). *For each $0 < m \leq M$, define the distance from the subgroup $V_m$ to the training set $V_0$ as*

$$\epsilon_m := \max_{j \in V_m} \min_{i \in V_0} \|g_i(X, G) - g_j(X, G)\|_2.$$

*Further, for each $i \in V_0$, define the near set of $i$ with respect to $V_m$ as*

$$V_m^{(i)} := \{j \in V_m \mid \|g_i(X, G) - g_j(X, G)\|_2 \leq \epsilon_m\}.$$

*Clearly,*

$$V_m = \cup_{i \in V_0} V_m^{(i)}.$$

**Assumption 3** (Concentrated Expected Loss Difference). *Let $P$ be a distribution on $\mathcal{H}$, defined by sampling the vectorized MLP parameters from $\mathcal{N}(0, \sigma^2 I)$ for some $\sigma^2 \leq \frac{(\gamma/8\epsilon_m)^{2/L}}{2b(\lambda N_0^{-\alpha} + \ln 2bL)}$. For any $L$-layer GNN classifier $h \in \mathcal{H}$ with model parameters $W_1^h, \ldots, W_L^h$, define $T_h := \max_{l=1,\ldots,L} \|W_l^h\|_2$. Assume that there exists some $0 < \alpha < \frac{1}{4}$ satisfying*

$$\Pr_{h \sim P} \left( \mathcal{L}_m^{\gamma/4}(h) - \mathcal{L}_0^{\gamma/2}(h) > N_0^{-\alpha} + cK\epsilon_m \mid T_h^L \epsilon_m > \frac{\gamma}{8} \right) \leq e^{-N_0^{2\alpha}}.$$

Assumption 3 postulates that the expected margin loss on the test node subgroup, $V_m$, is not significantly larger that on the train node subgroup, $V_0$, as the number of training samples $N_0 = |V_0|$ becomes larger. More discussions about this assumption can be found in Appendix A.5 of [46].

**Lemma 4** (Bound for $D_{m,0}^\gamma(P; \lambda)$ (Adaption of Lemma 6 in [46])). *Under Assumption 1, 2 and 3, for any $0 < m \leq M$, any $0 < \lambda \leq N_0^{2\alpha}$ and $\gamma \geq 0$, assume the "prior" $P$ on $\mathcal{H}$ is defined by sampling the vectorized MLP parameters from $\mathcal{N}(0, \sigma^2 I)$ for some $\sigma^2 \leq \frac{(\gamma/8\epsilon_m)^{2/L}}{2b(\lambda N_0^{-\alpha} + \ln 2bL)}$. We have*

$$D_{m,0}^{\gamma/2}(P; \lambda) \leq \ln 3 + \frac{\lambda K \rho}{\sqrt{2\pi}\sigma}(\epsilon + |h_0 - h_m| \cdot \rho). \tag{35}$$

We first present the following Lemma 5 that bounds the difference between the margin loss on $V_m$ and that on $V_0$.

**Lemma 5** (Adaption of Lemma 5 in [46]). *Suppose an $L$-layer GNN classifier $h$ is associated with model parameters $W_1, \ldots, W_L$. Define $T_h := \max_{l=1,\ldots,L} \|W_l\|_2$. Under Assumption 1 and 2, for any $0 < m \leq M$ and $\gamma \geq 0$, if $\epsilon_m T_h^L \leq \frac{\gamma}{4}$, then*

$$\mathcal{L}_m^{\gamma/2}(h) - \mathcal{L}_0^\gamma(h) \leq \frac{K\rho}{\sqrt{2\pi}\sigma}(\epsilon + |h_0 - h_m| \cdot \rho) \tag{36}$$

*Proof.* For simplicity in this proof, for any $i \in V_0 \cup V_m$ and $k = 1, \ldots, K$, we use $h_i$ to denote $h_i(X, G)$ and use $\eta_k(i)$ to denote $\Pr(y_i = k \mid g_i(X, G))$. And define $\mathcal{L}^\gamma(h_i, y_i) := \mathbb{1}[h_i[y_i] \leq \gamma + \max_{k \neq y_i} h_i[k]]$. Then we can write

$$\mathcal{L}_m^{\gamma/2}(h) - \mathcal{L}_0^\gamma(h)$$

$$= \mathbb{E}_{y^m}\left[ \frac{1}{N_m} \sum_{j \in V_m} \mathcal{L}^{\gamma/2}(h_j, y_j) \right] - \mathbb{E}_{y^0}\left[ \frac{1}{N_0} \sum_{i \in V_0} \mathcal{L}^\gamma(h_i, y_i) \right]$$

$$= \frac{1}{N_0}\mathbb{E}_{y^0, y^m} \sum_{i \in V_0} \frac{1}{s_m}\left( \sum_{j \in V_m^{(i)}} \mathcal{L}^{\gamma/2}(h_j, y_j) \right) - \mathcal{L}^\gamma(h_i, y_i)$$

where in the last step we have used Assumption 2. Therefore,

$$\mathcal{L}_m^{\gamma/2}(h) - \mathcal{L}_0^\gamma(h)$$

$$=\frac{1}{N_0}\sum_{i\in V_0}\frac{1}{s_m}\left(\sum_{j\in V_m^{(i)}}\mathbb{E}_{y_j}\mathcal{L}^{\gamma/2}(h_j,y_j)\right) - \mathbb{E}_{y_i}\mathcal{L}^\gamma(h_i,y_i)$$

$$=\frac{1}{N_0}\sum_{i\in V_0}\frac{1}{s_m}\left(\sum_{j\in V_m^{(i)}}\sum_{k=1}^K\eta_k(j)\mathcal{L}^{\gamma/2}(h_j,k)\right) - \sum_{k=1}^K\Pr(y_i=k)\mathcal{L}^\gamma(h_i,k)$$

$$=\frac{1}{N_0}\sum_{i\in V_0}\frac{1}{s_m}\sum_{j\in V_m^{(i)}}\sum_{k=1}^K\left(\eta_k(j)\mathcal{L}^{\gamma/2}(h_j,k) - \eta_k(i)\mathcal{L}^\gamma(h_i,k)\right)$$

$$=\frac{1}{N_0}\sum_{i\in V_0}\frac{1}{s_m}\sum_{j\in V_m^{(i)}}\sum_{k=1}^K\left(\eta_k(j)\left(\mathcal{L}^{\gamma/2}(h_j,k) - \mathcal{L}^\gamma(h_i,k)\right) + \left(\eta_k(j) - \eta_k(i)\right)\mathcal{L}^\gamma(h_i,k)\right) \quad (37)$$

$$\leq\frac{1}{N_0}\sum_{i\in V_0}\frac{1}{s_m}\sum_{j\in V_m^{(i)}}\sum_{k=1}^K\left(1\cdot\left(\mathcal{L}^{\gamma/2}(h_j,k) - \mathcal{L}^\gamma(h_i,k)\right) + \left(\eta_k(j) - \eta_k(i)\right)\cdot 1\right), \quad (38)$$

where the last inequality utilizes the facts that both $\eta_k(j)$ and $\mathcal{L}^\gamma(h_i,k)$ are upper-bounded by 1. According to the lemma 2, we can get:

$$\eta_k(j) - \eta_k(i) \leq \frac{\rho}{\sqrt{2\pi}\sigma}(\epsilon + |h_0 - h_m|\cdot\rho) \quad (39)$$

$\rho = \|\boldsymbol{\mu}_0 - \boldsymbol{\mu}_1\|$ denotes the original feature separability, independent with structure.

Further, as $h_i = f(g_i(X,G); W_1,\ldots,W_L)$ where $f$ is a ReLU-activated MLP, so

$$\|h_i - h_j\|_\infty \leq \|g_i(X,G) - g_j(X,G)\|_2\prod_{l=1}^L\|W_l\|_2 \leq \epsilon_m T_h^L \leq \frac{\gamma}{4}.$$

This implies that, for any $k = 1,\ldots,K$,

$$\mathcal{L}^{\gamma/2}(h_j,k) \leq \mathcal{L}^\gamma(h_i,k).$$

Detailed proof can be found in Lemma 5.

So we have

$$\mathcal{L}_m^{\gamma/2}(h) - \mathcal{L}_0^\gamma(h)$$

$$\leq\frac{1}{N_0}\sum_{i\in V_0}\frac{1}{s_m}\sum_{j\in V_m^{(i)}}\sum_{k=1}^K 0 + \frac{\rho}{\sqrt{2\pi}\sigma}(\epsilon + |h_0 - h_m|\cdot\rho)$$

$$=\frac{K\rho}{\sqrt{2\pi}\sigma}(\epsilon + |h_0 - h_m|\cdot\rho)$$

$\square$

Then we can prove the Bound for $D_{m,0}^\gamma(P;\lambda)$.

**Lemma 3** (Bound for $D_{m,0}^\gamma(P;\lambda)$). *Under Assumption 1, 2 and 3, for any $0 < m \leq M$, any $0 < \lambda \leq N_0^{2\alpha}$ and $\gamma \geq 0$, assume the "prior" $P$ on $\mathcal{H}$ is defined by sampling the vectorized MLP parameters from $\mathcal{N}(0,\sigma^2 I)$ for some $\sigma^2 \leq \frac{(\gamma/8\epsilon_m)^{2/L}}{2b(\lambda N_0^{-\alpha}+\ln 2bL)}$. We have*

$$D_{m,0}^{\gamma/2}(P;\lambda) \leq \ln 3 + \frac{\lambda K\rho}{\sqrt{2\pi}\sigma}(\epsilon + |h_0 - h_m|\cdot\rho). \quad (40)$$

*Proof.* Recall that $D_{m,0}^{\gamma/2}(P;\lambda) = \ln \mathbb{E}_{h \sim P} e^{\lambda \left( \mathcal{L}_m^{\gamma/4}(h) - \mathcal{L}_0^{\gamma/2}(h) \right)}$. We prove the upper bound of $D_{m,0}^{\gamma/2}(P;\lambda)$ by decomposing the space $\mathcal{H}$ into the two regimes: a regime with bounded spectral norms of the model parameters required by Lemma 5, and its complement. Following Lemma 5, for any classifier $h$ with parameters $W_1, \ldots, W_L$, we define $T_h := \max_{l=1,\ldots,L} \|W_l\|_2$.

We first prove an upper bound on the probability $\Pr\left( T_h^L \epsilon_m > \frac{\gamma}{8} \right)$ over the drawing of $h \sim P$. For any $h$, as its vectorized MLP parameters $\mathrm{vec}(W_l)$, for each $l = 1, \ldots, L$, is sampled from $\mathcal{N}(0, \sigma^2 I)$, we have the following spectral norm bound [99], for any $t > 0$,

$$\Pr(\|W_l\|_2 > t) \le 2be^{-\frac{t^2}{2b\sigma^2}},$$

where $b$ is the maximum width of all hidden layers of the MLP. Setting $t = \left( \frac{\gamma}{8\epsilon_m} \right)^{1/L}$ and applying a union bound, we have that

$$\Pr\left( T_h^L \epsilon_m > \frac{\gamma}{8} \right) = \Pr\left( T_h > \left( \frac{\gamma}{8\epsilon_m} \right)^{1/L} \right) \le 2bLe^{-\frac{(\gamma/8\epsilon_m)^{2/L}}{2b\sigma^2}} \le e^{-\lambda N_0^{-\alpha}},$$

where the last inequality utilizes the condition $\sigma^2 \le \frac{(\gamma/8\epsilon_m)^{2/L}}{2b(\lambda N_0^{-\alpha} + \ln 2bL)}$.

For any $h$ satisfying $T_h^L \epsilon_m \le \frac{\gamma}{8}$, by Lemma 5, we know that $e^{\lambda \left( \mathcal{L}_m^{\gamma/4}(h) - \mathcal{L}_0^{\gamma/2}(h) \right)} \le e^{\frac{K\lambda\rho}{\sqrt{2\pi}\sigma}(\epsilon + |h_0 - h_m| \cdot \rho)}$. For all $h$ such that $T_h^L \epsilon_m > \frac{\gamma}{8}$, by Assumption 3, with probability at least $1 - e^{-N_0^{2\alpha}}$,

$$e^{\lambda \left( \mathcal{L}_m^{\gamma/4}(h) - \mathcal{L}_0^{\gamma/2}(h) \right)} \le e^{\lambda N_0^{-\alpha} + \frac{K\lambda\rho}{\sqrt{2\pi}\sigma}(\epsilon + |h_0 - h_m| \cdot \rho)}.$$

Also note that $\mathcal{L}_m^{\gamma/4}(h) - \mathcal{L}_0^{\gamma/2}(h) \le 1$ trivially holds for any $h$. Therefore we have

$$D_{m,0}^{\gamma/2}(P;\lambda)$$
$$= \ln \mathbb{E}_{h \sim P} e^{\lambda \left( \mathcal{L}_m^{\gamma/4}(h) - \mathcal{L}_0^{\gamma/2}(h) \right)}$$
$$\le \ln \left( \Pr\left( T_h^L \epsilon_m > \frac{\gamma}{8} \right) \left( e^{-N_0^{2\alpha}} \cdot e^{\lambda} + (1 - e^{-N_0^{2\alpha}}) \cdot e^{\lambda N_0^{-\alpha} + \frac{K\lambda\rho}{\sqrt{2\pi}\sigma}(\epsilon + |h_0 - h_m| \cdot \rho)} \right) \right.$$
$$\left. + \Pr\left( T_h^L \epsilon_m \le \frac{\gamma}{8} \right) e^{\frac{K\lambda\rho}{\sqrt{2\pi}\sigma}(\epsilon + |h_0 - h_m| \cdot \rho)} \right)$$
$$\le \ln 3 + \frac{K\lambda\rho}{\sqrt{2\pi}\sigma}(\epsilon + |h_0 - h_m| \cdot \rho)$$

$\square$

**Lemma 5.** *Define* $\mathcal{L}^{\gamma}(h_i, y_i) = \mathbb{1}\left[ h_i[y_i] \le \gamma + \max_{k \ne y_i} h_i[k] \right]$, *if* $\|h_i - h_j\|_\infty \le \frac{\gamma}{4}$, *for any* $k = 1, 2, \cdots, K$, $\mathcal{L}^{\gamma/2}(h_j, k) \le \mathcal{L}^{\gamma}(h_i, k)$

We need to proof from $h_j[k] \le \frac{\gamma}{2} + \max_{l \ne k} h_j[l]$ to $h_i[k] \le \gamma + \max_{i \ne k} h_i[l]$

$$h_j[k] \le \frac{\gamma}{2} + \max_{l \ne k} h_j[l]$$
$$h_j[k] + (h_i[k] - h_j[k]) \le \frac{\gamma}{2} + \max_{l \ne k} h_j[l] + (h_i[k] - h_j[k])$$
$$h_i[k] \le \frac{\gamma}{2} + \max_{l \ne k} h_j[l] + (h_i[k] - h_j[k])$$
$$h_i[k] \le \frac{3}{4}\gamma + \max_{l \ne k} h_j[l]$$

Then we need to proof $\frac{3}{4}\gamma + \max_{l \neq k} h_j[l] \leq \gamma + \max_{l \neq k} h_i[l]$

$$\max_{l \neq k} h_j[l] - \max_{m \neq k} h_i[m] = \max_{l \neq k} \left[ h_j[l] - \max_{m \neq k} h_i[m] \right]$$

$$\leq \max_{l \neq k} \left[ h_j[l] - h_j[l] \right]$$

$$\leq \frac{\gamma}{4}$$

The proof complete.

## F.4 Proof details of subgroup generalization Bound for GNNs

With the bound for $D_{m,0}^{\gamma}(P; \lambda)$, we can then derive the subgroup generalization Bound for GNNs. **Remark: Notably, the following proof is derived based on [46]. For completeness, we show the previous proof as follows, the difference is that we consider a more complicated conditional probability $\mathbf{P}(y_i = k | g_i(X, G))$, correlated with the homophily ratio difference, rather than consider the conditional probability as $c$-Lipschitz continuous functions. In other words, our proof further strengthens the effect of the structure disparity.**

**Assumption 1** (GNN model). *We focus on SGC [16] with the following components: (1) a one-hop mean aggregation function $g$ with $g(X, G)$ denoting the output. (2) MLP feature transformation $f(g_i(X, G); W_1, W_2, \cdots, W_L)$, where $f$ is a ReLU-activated $L$-layer MLP with $W_1, \cdots, W_L$ as parameters for each layer. The largest width of all the hidden layers is denoted as $b$.*

Despite analyzing simple GNN architecture theoretically, similar with [46, 13, 47], our theory analysis could be easily extended to the higher-order case with empirical success across different GNN architectures shown in Section 3.4. Notably, the following assumptions 2 and 3 are adapted from [46].

**Assumption 2.** *Define $B_m := \max_{i \in V_0 \cup V_m} \|g_i(X, G)\|_2$. For any classifier $\tilde{h} \in \mathcal{H}$ with parameters $\{\widetilde{W}_l\}_{l=1}^L$, assume $\|\widetilde{W}_l\|_F \leq C$ for $l = 1, \ldots, L$. Assume $B_m, C$ are constants with respect to $N_0$.*

**Theorem 1** (Subgroup Generalization Bound for GNNs). *Let $\tilde{h}$ be any classifier in $\mathcal{H}$ with parameters $\{\widetilde{W}_l\}_{l=1}^L$. for any $0 < m \leq M$, $\gamma \geq 0$, and large enough $N_0$, with probability at least $1 - \delta$ over the sample of $y^0$, we have*

$$\mathcal{L}_m^0(\tilde{h}) \leq \widehat{\mathcal{L}}_0^{\gamma}(\tilde{h}) + O\left( \frac{2e^{\frac{1}{2}}}{\sqrt{2\pi}\sigma}(\epsilon_m + |h_0 - h_m| \cdot |\mu_0 - \mu_1|) + \frac{b \sum_{l=1}^L \|\widetilde{W}_l\|_F^2}{(\gamma/8)^{2/L} N_0^{\alpha}}(\epsilon_m)^{2/L} \right.$$
$$\left. + \frac{1}{N_0^{1-2\alpha}} + \frac{1}{N_0^{2\alpha}} \ln \frac{LC(2B_m)^{1/L}}{\gamma^{1/L}\delta} \right). \tag{41}$$

The proof of Theorem 1 relies on the combination of Theorem 2, Lemma 4, and an intermediate result of the Theorem 1 in [100] (which we state as Lemma 6 below).

**Lemma 6** ([100]). *Let $\tilde{h}$ be any classifier in $\mathcal{H}$ with parameters $\{\widetilde{W}_l\}_{l=1}^L$. Define $\tilde{\beta} = \left( \prod_{l=1}^L \|\widetilde{W}_l\|_2 \right)^{1/L}$. Let $\{U_l\}_{l=1}^L$ be the random perturbation to be added to $\{\widetilde{W}_l\}_{l=1}^L$ and $vec(\{U_l\}_{l=1}^L) \sim \mathcal{N}(0, \sigma^2 I)$. Define $B_m := \max_{i \in V_0 \cup V_m} \|g_i(X, G)\|_2$. If*

$$\sigma \leq \frac{\gamma}{84 L B_m \beta^{L-1} \sqrt{b \ln(4bL)}},$$

*and $\beta$ is any constant satisfying $|\tilde{\beta} - \beta| \leq \frac{\tilde{\beta}}{L}$, then with respect to the random draw of $\{U_l\}_{l=1}^L$,*

$$\Pr\left( \max_{i \in V_0 \cup V_m} \|f(g_i(X, G); \{\widetilde{W}_l\}_{l=1}^L) - f(g_i(X, G); \{\widetilde{W}_l + U_l\}_{l=1}^L)\|_\infty < \frac{\gamma}{8} \right) > \frac{1}{2}.$$

Then we prove Theorem 1, a re-stata as Theorem 1 with replacing the notation of training node subgroup $V_{\text{train}}$ to $V_0$.

**Theorem 1** (Subgroup Generalization Bound for GNNs (Adaption of Theorem 3 in [46])). *Let $\tilde{h}$ be any classifier in $\mathcal{H}$ with parameters $\{\widetilde{W}_l\}_{l=1}^L$. Under Assumptions 1, 1, 2, and 3, for any $0 < m \leq M$, $\gamma \geq 0$, and large enough $N_0$, with probability at least $1 - \delta$ over the sample of $y^0$, we have*

$$\mathcal{L}_m^0(\tilde{h}) \leq \widehat{\mathcal{L}}_{tr}^\gamma(\tilde{h}) + O\left(\frac{K\rho}{\sqrt{2\pi}\sigma}(\epsilon_m + |h_{tr} - h_m| \cdot \rho) + \frac{b\sum_{l=1}^L \|\widetilde{W}_l\|_F^2}{(\gamma/8)^{2/L} N_{tr}^\alpha}(\epsilon_m)^{2/L} + \frac{1}{N_0^{1-2\alpha}} + \frac{1}{N_0^{2\alpha}}\ln\frac{1}{\delta}\right)$$

$$(42)$$

*Proof.* Note that, the following proof is adopted from [46]. There are two main steps in the proof. In the first step, for a given constant $\beta > 0$, we first define the "prior" $P$ and the "posterior" $Q$ on $\mathcal{H}$ in a way complying the conditions in Lemma 4 and Lemma 6. Then for all classifiers with parameters satisfying $|\tilde{\beta} - \beta| \leq \frac{\tilde{\beta}}{L}$, where $\tilde{\beta} = \left(\prod_{l=1}^L \|\widetilde{W}_l\|_2\right)^{1/L}$, we can derive a generalization bound by applying Theorem 2 and Lemma 4. In the second step, we investigate the number of $\beta$ we need to cover all possible relevant classifier parameters and apply a union bound to get the final bound. The second step is essentially the same as [100] while the first step differs by the need of incorporating Lemma 4.

We first show the first step. Given a choice of $\beta$ independent of the training data, let

$$\sigma = \min\left(\frac{(\gamma/8\epsilon_m)^{1/L}}{\sqrt{2b(\lambda N_0^{-\alpha} + \ln 2bL)}}, \frac{\gamma}{84LB_m\beta^{L-1}\sqrt{b\ln(4bL)}}\right).$$

Assume the "prior" $P$ on $\mathcal{H}$ is defined by sampling the vectorized MLP parameters from $\mathcal{N}(0, \sigma^2 I)$; and the "posterior" $Q$ on $\mathcal{H}$ is defined by first sampling a set of random perturbations $\{U_l\}_{l=1}^L$ with $\text{vec}(\{U_l\}_{l=1}^L) \sim \mathcal{N}(0, \sigma^2 I)$ and then adding them to $\{\widetilde{W}_l\}_{l=1}^L$, the parameters of $\tilde{h}$. Then for any $\tilde{h}$ with $\{\widetilde{W}_l\}_{l=1}^L$ satisfying $|\tilde{\beta} - \beta| \leq \frac{\tilde{\beta}}{L}$, by Lemma 6, we have

$$\Pr_{h \sim Q}\left(\max_{i \in V_0 \cup V_m} |h_i(X, G) - \tilde{h}_i(X, G)|_\infty < \frac{\gamma}{8}\right) > \frac{1}{2}.$$

Therefore, by applying Theorem 2, we know the bound (33) holds for $\tilde{h}$, i.e., with probability at least $1 - \delta$,

$$\mathcal{L}_m^0(\tilde{h}) - \widehat{\mathcal{L}}_0^\gamma(\tilde{h})$$

$$\leq \frac{1}{\lambda}\left(2(D_{\text{KL}}(Q\|P) + 1) + \ln\frac{1}{\delta} + \frac{\lambda^2}{4N_0} + D_{m,0}^{\gamma/2}(P; \lambda)\right)$$

$$\leq \frac{1}{\lambda}\left(2(D_{\text{KL}}(Q\|P) + 1) + \ln\frac{1}{\delta} + \frac{\lambda^2}{4N_0} + \ln 3 + \frac{\lambda K\rho}{\sqrt{2\pi}\sigma}(\epsilon_m + |h_{\text{tr}} - h_m| \cdot \rho)\right) \quad (43)$$

$$\leq \frac{2}{N_0^{2\alpha}}D_{\text{KL}}(Q\|P) + \frac{1}{N_0^{2\alpha}}\left(\ln\frac{3}{\delta} + 2\right) + \frac{1}{4N_0^{1-2\alpha}} + \frac{K\rho}{\sqrt{2\pi}\sigma}(\epsilon_m + |h_{\text{tr}} - h_m| \cdot \rho), \quad (44)$$

where in (43) we have applied Lemma 4 to bound $D_{m,0}^{\gamma/2}(P; \lambda)$ under Assumptions 1, 2, and 3; and in (44) we have set $\lambda = N_0^{2\alpha}$.

Moreover, since both $P$ and $Q$ are normal distributions, we know that

$$D_{\text{KL}}(Q\|P) \leq \frac{\sum_{l=1}^L \|\widetilde{W}_l\|_F^2}{2\sigma^2}.$$

By Assumption 2, both $B_m$ and $C$ are constant with respect to $N_0$. Later we will show that we only need $\beta \leq C$. Therefore, for large enough $N_0$, we can have

$$\frac{(\gamma/8\epsilon_m)^{1/L}}{\sqrt{2b(N_0^\alpha + \ln 2bL)}} < \frac{\gamma}{84LB_m\beta^{L-1}\sqrt{b\ln(4bL)}},$$

which implies,

$$\sigma = \frac{(\gamma/8\epsilon_m)^{1/L}}{\sqrt{2b(N_0^\alpha + \ln 2bL)}},$$

and hence,

$$D_{\mathrm{KL}}(Q\|P) \leq \frac{b(N_0^\alpha + \ln 2bL)\sum_{l=1}^{L}\|\widetilde{W}_l\|_F^2}{(\gamma/8)^{2/L}}(\epsilon_m)^{2/L}. \tag{45}$$

Therefore, with probability at least $1 - \delta$,

$$\mathcal{L}_m^0(\tilde{h}) - \widehat{\mathcal{L}}_0^\gamma(\tilde{h})$$
$$\leq \frac{K\rho}{\sqrt{2\pi}\sigma}(\epsilon_m + |h_0 - h_m| \cdot \rho) + \frac{2}{N_0^{2\alpha}}D_{\mathrm{KL}}(Q\|P) + \frac{1}{N_0^{2\alpha}}\left(\ln\frac{3}{\delta} + 2\right) + \frac{1}{4N_0^{1-2\alpha}}$$
$$\leq O\left(\frac{K\rho}{\sqrt{2\pi}\sigma}(\epsilon_m + |h_0 - h_m| \cdot \rho) + \frac{b\sum_{l=1}^{L}\|\widetilde{W}_l\|_F^2}{(\gamma/8)^{2/L}N_0^\alpha}(\epsilon_m)^{2/L} + \frac{1}{N_0^{1-2\alpha}} + \frac{1}{N_0^{2\alpha}}\ln\frac{1}{\delta}\right). \tag{46}$$

Then we show the second step, i.e., finding out the number of $\beta$ we need to cover all possible relevant classifier parameters. Similarly as [100], we will show that we only need to consider $(\frac{\gamma}{2B_m})^{1/L} \leq \tilde{\beta} \leq C$ (recall that $\|\widetilde{W}_l\|_F \leq C, l = 1, \ldots, L$). For any $\tilde{\beta}$ outside this range, the bound (47) automatically holds. If $\tilde{\beta} < (\frac{\gamma}{2B_m})^{1/L}$, then for any node $i \in V_0$, $\|\tilde{h}_i(X, G)\|_\infty < \frac{\gamma}{2}$, which implies $\widehat{\mathcal{L}}_0^\gamma(\tilde{h}) = 1$ as the difference between any two output logits for any training node is smaller than $\gamma$. Also noticing that $\mathcal{L}_m^0(\tilde{h}) \leq 1$ by definition, so the bound (47) trivially holds. And for $\tilde{\beta}$ in this range, $|\beta - \tilde{\beta}| \leq \frac{1}{L}(\frac{\gamma}{2B_m})^{1/L}$ is a sufficient condition for $\beta$ to satisfy $|\tilde{\beta} - \beta| \leq \frac{\tilde{\beta}}{L}$, and we need at most $\frac{LC(2B_m)^{1/L}}{\gamma^{1/L}}$ of $\beta$ to cover all $\tilde{\beta}$ in the above range. Taking a union bound on all such $\beta$, which is equivalent to replace $\delta$ with $\frac{\delta}{\frac{LC(2B_m)^{1/L}}{\gamma^{1/L}}}$ in (46), it gives us the final result: with probability at least $1 - \delta$,

$$\mathcal{L}_m^0(\tilde{h}) \leq \widehat{\mathcal{L}}_0^\gamma(\tilde{h}) + O\left(\frac{K\rho}{\sqrt{2\pi}\sigma}(\epsilon_m + |h_0 - h_m| \cdot \rho) + \frac{b\sum_{l=1}^{L}\|\widetilde{W}_l\|_F^2}{(\gamma/8)^{2/L}N_0^\alpha}(\epsilon_m)^{2/L}\right.$$
$$\left. + \frac{1}{N_0^{1-2\alpha}} + \frac{1}{N_0^{2\alpha}}\ln\frac{LC(2B_m)^{1/L}}{\gamma^{1/L}\delta}\right). \tag{47}$$

$\square$

# G  Experiment details

## G.1  Hardware & Software Environment

The experiments are performed on two Linux servers (CPU: Intel(R) Xeon(R) CPU E5-2690 v4 @ 2.60GHz, Operation system: Ubuntu 16.04.6 LTS). For GPU resources, four NVIDIA Tesla V100 cards are utilized The python libraries we use to implement our experiments are PyTorch 1.12.1 and PyG 2.1.0.post1. The maximum time for training one epoch is no more than one minute.

## G.2  Model details & hyperparameter settings & results

In this section, we provide details for baseline methods, hyperparameter search space, and the accuracy results.

**Details for baseline methods** We provide details for baseline methods in this section. MLP, GCN [2], GAT [11], and SGC [16] are classical baseline models, we directly adopt the implementation of them from Pytorch Geometric. Notice that, SRGNN and EERM are two baseline method specifically for the Graph OOD scenario.

- GLNN, as proposed by [29], address the scalability challenges faced by Graph Neural Networks (GNNs) in practical industry deployments due to their data dependency. By leveraging the strengths of both GNNs and MLPs through knowledge distillation (KD), the authors

demonstrate that MLP performance can be significantly improved. The authors' code can be found at `https://github.com/snap-research/graphless-neural-networks`. In this work, we adopt GCN as the teacher model.

- APPNP, developed by [15], leverages the relationship between graph convolutional networks (GCN) and PageRank to develop an enhanced propagation scheme based on personalized PageRank. APPNP focuses on addressing the limitations of traditional neural message-passing algorithms in graph-based semi-supervised classification, which only considers a small, fixed neighborhood of nodes. The author's code is available at `https://github.com/gasteigerjo/ppnp`.

- GPRGNN, introduced by [53], aggregates features across multiple steps and subsequently combines them linearly, with the weights of the linear combination being learned during model training. The original features prior to aggregation are also incorporated into the combination. The authors' code is available at `https://github.com/jianhao2016/GPRGNN`.

- GCNII, developed by [52], is a deep learning method for graph-structured data that extends the traditional GCN model by incorporating two innovative yet straightforward techniques: "Initial residual" and "Identity mapping." These techniques effectively address the over-smoothing issue commonly encountered in shallow GCN models. The author's code is available at `https://github.com/chennnM/GCNII`.

- SRGNN, introduced by [63], is a novel framework incorporating a shift-robust regularizer that encourages the learned representation to enhance the ood generalization capabilities of GNNs and overcome the limitations posed by localized graph training data. The author's code is available at `https://github.com/GentleZhu/Shift-Robust-GNNs`.

- EERM, introduced by [62], proposes a principled methodology to identify and quantify these factors in graph data and incorporate a novel invariance-inducing regularization term into the GNN training objective. The author's code is available at `https://github.com/qitianwu/GraphOOD-EERM`.

**hyperparameter searching range** For model-specific parameters, we use the same notations for the following models considering brevity. Details are shown as follows.

- $\lambda$: For GLNN, $\lambda$ refers to the weight of knowledge distillation loss. For GCNII, $\lambda$ refers to the strength of identity mapping.

- $\alpha$: For APPNP and GPRGNN, $\alpha$ refers to the teleport probability. For GCNII, $\alpha$ refers to the strength of residual connection.

- $K$: For SGC, APPNP, and GPRGNN, $K$ refers to the number of transformation layers.

In hyperparameter searching, we utilize the common range of hyperparameters for most models. Notice that, there are particular models where the reported optimal parameters in their papers are quite specific. In order to achieve better implementation results, we expand the search range on top of the original reported set for these models.

Table 7: Hyperparameter searching range for baseline models on Cora, CiteSeer, PubMed

| Hyperparameters for Baseline GNNs | | | | | | | | | |
|---|---|---|---|---|---|---|---|---|---|
| Models\Hyperparameters | lr | weight_decay | dropout | hidden | # layers | Gat heads | $\lambda$ | $\alpha$ | $K$ |
| GCN | {1e-2, 5e-2, 5e-3} | {0, 5e-4, 5e-5} | {0., 0.5, 0.8} | {16, 32, 64} | 2 | - | - | - | - |
| SGC | {1e-2, 5e-2, 5e-3} | {0, 5e-4, 5e-5} | {0., 0.5, 0.8} | {16, 32, 64} | {1,2} | - | - | - | {1,2,3} |
| GAT | {1e-2, 5e-2, 5e-3} | {0, 5e-4, 5e-5} | {0., 0.5, 0.6, 0.8} | {8, 16, 32, 64} | 2 | {1, 4, 8} | - | - | - |
| GLNN | {1e-2, 5e-2, 5e-3} | {0, 5e-4, 5e-5} | {0., 0.5, 0.8} | {16, 32, 64} | {1, 2} | - | {0.0-1.0} | - | - |
| APPNP | {1e-2, 5e-2, 5e-3} | {0, 5e-4, 5e-5} | {0., 0.5, 0.8} | {8, 16, 32, 64} | 2 | - | - | {0.0-1.0} | {5,6,7,8,9,10} |
| MLP | {1e-2, 5e-2, 5e-3} | {0, 5e-4, 5e-5} | {0., 0.5, 0.8} | {16, 32, 64} | 2 | - | - | - | - |
| GPRGNN | {1e-2, 5e-2, 5e-3} | {0, 1e-3, 1e-5, 1e-4, 5e-4, 5e-5} | {0., 0.2, 0.5, 0.8} | {16, 32, 64, 128} | {2,3} | - | - | {0.0-1.0} | {5,8,9,10} |
| GCNII | {1e-2, 5e-3, 1e-3} | {0, 5e-5, 5e-4} | {0., 0.1, 0.2, 0.4, 0.5, 0.8} | { 16, 32, 64, 128, 256} | {2, 4, 8, 16, 32, 64} | - | {0.0-1.0} | {0.0-1.0} | - |

**Reimplementation experimental results** We illustrate the experimental results with the above hyperparameter search in Table 10 and 11 on homophilic and heterophilic datasets, respectively.

## G.3 Dataset details

In this paper, we majorly focus on two homophilic datasets, PubMed and Ogbn-Arxiv and two heterophilic datasets, Squirrel and Chameleon. Moreover, we consider more datasets for comprehensive

Table 8: Hyperparameter searching range for baseline models on Chameleon, Squirrel, Actor, Amazon-ratings

| | | | Hyperparameters for Baseline GNNs | | | | | | |
|---|---|---|---|---|---|---|---|---|---|
| Models\Hyperparameters | lr | weight_decay | dropout | hidden | # layers | Gat heads | $\lambda$ | $\alpha$ | $K$ |
| GCN | {1e-2, 5e-2, 5e-3, } | {0, 5e-4, 5e-5 } | {0., 0.2, 0.5, 0.8} | {16, 32, 64, 128, 256} | {2,3,4} | - | - | - | - |
| SGC | {1e-2, 5e-2, 5e-3, 3e-5} | {0, 5e-4, 5e-5} | {0, 0.2, 0.5, 0.8} | {16, 32, 64, 128, 256} | {2, 3} | - | - | - | {1,2,3} |
| GAT | {1e-2, 5e-2, 1e-3, 2e-3, 3e-5} | {0, 5e-3, 5e-2} | {0., 0.2, 0.5, 0.6, 0.8} | {8, 16, 32, 64, 128, 256} | {2, 3} | {1, 4, 8} | - | - | - |
| GLNN | {1e-2, 5e-2, 5e-3} | {0, 5e-4, 5e-3} | {0., 0.2, 0.5, 0.8} | {16, 32, 64, 128, 256} | {1, 2} | - | {0.0-1.0} | - | - |
| APPNP | {1e-2, 5e-2, 5e-3} | {0, 5e-4, 5e-3} | {0., 0.2,0.5, 0.8} | {8, 16, 32, 64, 128} | 2 | - | - | {0.0-1.0} | {5,6,7,8,9,10} |
| MLP | {1e-2, 5e-2, 5e-3} | {0, 5e-4, 5e-3, 5e-5, 5e-8} | {0., 0.2, 0.5, 0.8} | {16, 32, 64, 128, 256} | 2 | - | - | - | - |
| GPRGNN | {1e-2, 5e-2, 5e-3} | {0, 1e-3, 1e-5, 1e-4, 5e-5} | {0., 0.2, 0.5, 0.8} | {16, 32, 64, 128} | {2,3} | - | - | {0.0-1.0} | {5,8,9,10} |
| GCNII | {1e-2, 5e-3, 1e-3} | {0, 5e-5, 5e-4} | {0., 0.1, 0.2, 0.4, 0.5, 0.8} | { 16, 32, 64, 128, 256} | {2, 4, 8, 16, 32, 64} | - | {0.0-1.0} | {0.0-1.0} | - |

Table 9: Hyperparameter searching range for baseline models on OGB-Arxiv, Twitch-gamer, IGB-tiny

| | | | Hyperparameters for Baseline GNNs | | | | | | |
|---|---|---|---|---|---|---|---|---|---|
| Models\Hyperparameters | lr | weight_decay | dropout | hidden | # layers | Gat heads | $\lambda$ | $\alpha$ | $K$ |
| GCN | {1e-2, 5e-2, 5e-3} | {0, 5e-4, 5e-5} | {0., 0.5, 0.8} | {128, 256, 512} | {2, 3} | - | - | - | - |
| SGC | {1e-2, 5e-2, 5e-3} | {0, 5e-4, 5e-5} | {0., 0.5, 0.8} | {64, 128, 256, 512} | {2, 3} | - | - | - | {1,2,3} |
| GAT | {1e-2, 5e-2, 5e-3} | {0, 5e-4, 5e-5} | {0., 0.5, 0.6, 0.8} | {64, 128, 256, 512} | {2, 3} | {1, 4, 8} | - | - | - |
| GLNN | {1e-2, 5e-2, 5e-3} | {0, 5e-4, 5e-3} | {0., 0.5, 0.8} | {1024, 2048} | {2, 3} | - | {0.0-1.0} | - | - |
| APPNP | {1e-2, 5e-2, 5e-3} | {0, 5e-3, 1e-2, 5e-4, 5e-5} | {0., 0.2, 0.3, 0.5, 0.8} | {128, 256, 512} | {2, 3} | - | - | {0.0-1.0} | {5,6,7,8,9,10} |
| MLP | {1e-2, 5e-2, 5e-3} | {0, 5e-4, 5e-5} | {0., 0.5, 0.8} | {256, 512} | {2, 3} | - | - | - | - |
| GPRGNN | {1e-2, 5e-2, 5e-3} | {0, 1e-3, 1e-5, 1e-4, 5e-5} | {0., 0.2, 0.5, 0.8} | {128, 256, 512} | {2,3} | - | - | {0.0-1.0} | {5,8,9,10} |
| GCNII | {1e-2, 5e-3, 1e-3} | {0, 5e-5, 5e-4} | {0., 0.1, 0.2, 0.4, 0.5, 0.8} | { 64, 128, 256, 512} | {2, 4, 8, 16, 32, 64} | - | {0.0-1.0} | {0.0-1.0} | - |

Table 10: The accuracy of GNN and MLP models on homophilic graphs

| Dataset | Cora | Citeseer | Pubmed | Arxiv | IGB-tiny |
|---|---|---|---|---|---|
| MLP | 61.1±1.2 | 60.0±1.4 | 69.0±2.3 | 54.0±0.1 | 73.2±0.1 |
| GLNN | 81.3±1.5 | 73.0±2.7 | 78.2±2.6 | 71.7±0.1 | 73.2±0.1 |
| GCN | 81.5±1.4 | 73.7±1.6 | 77.9±2.0 | 71.4±0.1 | 70.7±0.1 |
| SGC | 81.7±1.4 | 72.7±2.2 | 77.0±2.7 | 68.0±0.1 | 71.0±0.1 |
| GAT | 82.2±1.1 | 73.6±1.6 | 77.3±1.5 | 71.0±0.1 | 70.8±0.2 |
| APPNP | 83.1±1.3 | 75.0±1.1 | 79.6±1.3 | 70.3±0.5 | 71.2±0.1 |
| GCNII | 82.8±1.1 | 73.8±1.7 | 79.0±2.5 | 71.7±0.5 | 73.5±0.1 |
| GPRGNN | 82.9±1.4 | 72.4±1.8 | 78.3±2.1 | 72.3±0.3 | 73.9±0.1 |

Table 11: The accuracy of GNN and MLP models on heterophilic graphs

| Dataset | Chameleon | Squirrel | Twitch-gamers | Actor | Amazon-ratings |
|---|---|---|---|---|---|
| MLP | 49.0±2.4 | 30.1±1.7 | 60.7±0.2 | 37.0±0.7 | 45.9±0.8 |
| GLNN | 39.2±2.7 | 52.3±1.4 | 61.1±0.1 | 37.3±1.0 | 54.0±0.7 |
| GCN | 68.0±2.0 | 54.7±1.4 | 62.2±0.2 | 30.7±0.9 | 49.0±0.6 |
| SGC | 69.1±1.8 | 53.0±1.1 | 62.0±2.0 | 30.0±1.5 | 46.5±0.6 |
| GAT | 67.0±1.9 | 53.2±1.7 | 59.9±0.3 | 30.7±1.0 | 48.0±0.5 |
| APPNP | 56.7±2.5 | 42.4±1.9 | 59.7±0.1 | 37.0±1.3 | 44.9±0.8 |
| GCNII | 64.7±1.8 | 44.0±1.5 | 64.5±0.3 | 36.0±1.2 | 50.0±0.5 |
| GPRGNN | 68.5±1.4 | 53.8±1.4 | 61.9±0.2 | 36.5±1.4 | 49.8±0.5 |

experimental results including Cora, CiteSeer, IGB-tiny, Actor, Amazon-rating, and Twitch-gamers. Dataset statistics details can be found in Tab. 13. The license of datasets can be found in Table 12.

Table 12: The license of datasets.

| Dataset | license |
|---|---|
| Cora | NLM license |
| CiteSeer | NLM license |
| PubMed | NLM license |
| Ogbn-arxiv | ODC-BY |
| IGB-tiny | ODC-By-1.0 |
| Chameleon | MIT license |
| Squirrel | MIT license |
| Actor | MIT license |
| Amazon-ratings | MIT license |
| Twitch-gamers | MIT license |

We then provide a detailed description on those datasets and the corresponding data split as follows:

- Planetoid datasets include Cora, CiteSeer, and PubMed. They consist of scientific publications as nodes. The citation links between the papers create a graph structure, with nodes representing papers and edges representing citations. We utilize the 10 fixed splits in [101]. For each split, we have 20 nodes per class for training, 30 nodes per class for validation, and all other nodes for testing. More details can be found on `https://github.com/BUPT-GAMMA/CPF/tree/master/data/npz`

- Ogbn-Arxiv dataset is based on a large-scale scholarly paper citation network, where nodes represent papers and edges represent citation links between them. We utilize the default fix split provided by [24]. More details can be found on `https://ogb.stanford.edu/docs/nodeprop/#ogbn-arxiv`.

- Wikipedia datasets include Chameleon and Squirrel, which are a collection of web pages from Wikipedia. The node features correspond to several informative nouns in the respective Wikipedia pages. The edge represents the link between node. We utilize the 10 fixed splits in [25]. For each split, we have 48% nodes for training, 32% nodes for validation, and 20% for test. More details can be found on `https://github.com/graphdml-uiuc-jlu/geom-gcn`

- The Actor dataset [102] is a graph that represents the co-occurrence relationships between actors. We utilize the 10 fixed splits in [25]. For each split, we have 48% nodes for training, 32% nodes for validation, and 20% for test. More details can be found on `https://github.com/graphdml-uiuc-jlu/geom-gcn`

- The Twitch-gamers dataset represents the relationships between accounts on the streaming platform Twitch, where each node corresponds to a Twitch account, and edges are present between accounts that are mutual followers. We utilize the 5 fixed splits in [21]. For each split, we have 50% nodes for training, 25% nodes for validation, and 25% for test. More details can be found on `https://github.com/CUAI/Non-Homophily-Large-Scale`

- Amazon-ratings a new dataset derived from the Amazon product co-purchasing network metadata dataset, which is part of the SNAP Datasets [103]. We utilize the 10 fixed splits in [104]. For each split, we have 50% nodes for training, 25% nodes for validation, and 25% for test. More details can be found on `https://github.com/yandex-research/heterophilous-graphs/tree/main/data`

- IGB-tiny is a recent new citation graph dataset. It utilizes two publicly available datasets: Microsoft Academic Graph (MAG) and SemanticScholar Corpus. We utilize the homogeneous setting with default fix split provide by [105]. More details can be found on `https://github.com/IllinoisGraphBenchmark/IGB-Datasets`.

**Details on Data distribution** The node homophily ratio distribution for different datasets are presented in Figure 13. The following observations can be made: (1) Cora, CiteSeer, PubMed, and Ogbn-Arxiv are four homophilic datasets where most nodes have a homophily ratio $h > 0.5$. (2) Chamaleon, Squirrel, and Actor are three heterophilic datasets where most nodes have a homophily ratio $h \leq 0.5$. Nonetheless, the Actor dataset exhibits entirely different properties from Chamaleon and Squirrel where graph structure information is barely useful, leading to performance degradation. Empirical evidence can be found in Table 11 where the vanilla MLP outperforms all the GNN models. (3) IGB-tiny [105] and Twitch-gamers [21] are two representative datasets with no apparent majority structural pattern, exhibiting graph homophily ratios of 0.567 and 0.556, respectively. Specifically, IGB-tiny contains both large numbers of homophilic and heterophilic nodes, respectively, whereas, most nodes in the Twitch-gamers dataset do not show a clear homophilic nor heterophilic pattern with a homophily ratio close to 0.5. Notably, the IGB-tiny is still recognized as a homophilic graph [105] while Twitch-gamers is generally recognized as a homophilic graph [20, 21]. (4) Nodes in Amazon-ratings [104] exhibit diverse structural patterns with multiple peaks observed for homophily ratios 0.0, 0.2, 0.4, 0.6, and 0.8.

### G.4 OOD dataset statistics and details

We provide more analysis and statistics details about datasets with our proposed out-of-distribution (OOD) data split for a more comprehensive analysis. The data split statistics can be found in Table 14. 20% of the majority nodes are selected as the validation set. Notably, we generally use nodes with $h_i < 0.5$ and $h_i > 0.5$ as the majority and minority on heterophilic graph.

Table 13: Detailed dataset statistics

| Dataset | Cora | CiteSeer | PubMed | Arxiv | IGB-tiny | Squirrel | Chameleon | Amazon-ratings | Actor | Twitch-gamers |
|---|---|---|---|---|---|---|---|---|---|---|
| Homo Ratio | 0.814 | 0.714 | 0.792 | 0.631 | 0.567 | 0.216 | 0.247 | 0.376 | 0.221 | 0.556 |
| Nodes | 2485 | 2110 | 19717 | 169343 | 100000 | 5201 | 2277 | 24492 | 7600 | 249992 |
| Edges | 5069 | 3668 | 44324 | 583121 | 223538 | 198493 | 31421 | 93050 | 15009 | 93050 |
| Features | 1433 | 3703 | 500 | 128 | 1024 | 2089 | 2235 | 300 | 932 | 300 |
| Classes | 7 | 6 | 3 | 40 | 19 | 5 | 5 | 5 | 5 | 5 |

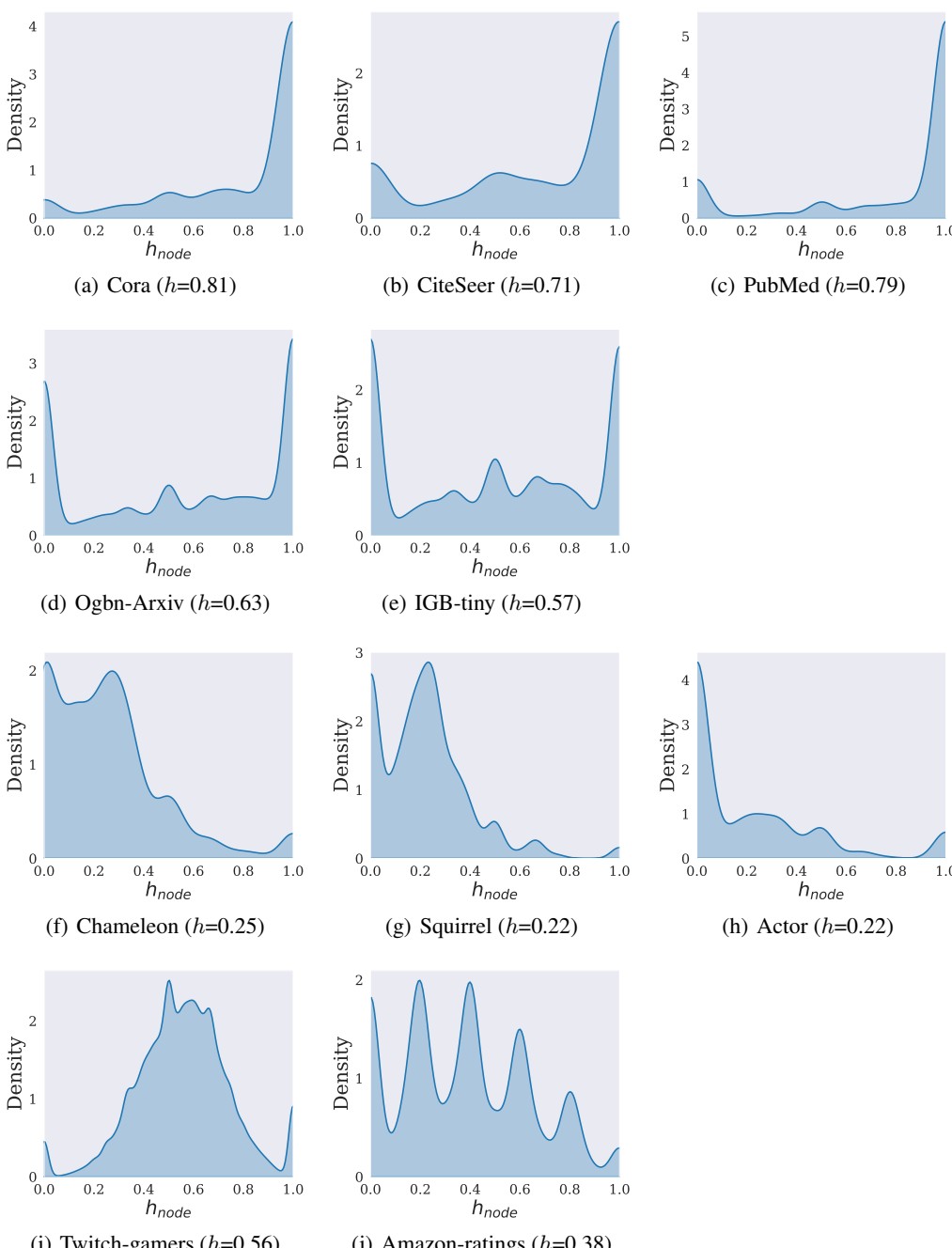

Figure 13: Node homophily ratio distributions of different datasets. Both homophilic and heterophilic nodes exist across all graphs

Table 14: the numbers of train, validation, test nodes on OOD data split

| Dataset | Cora | CiteSeer | PubMed | Arxiv | Squirrel | Chameleon |
|---------|------|----------|--------|-------|----------|-----------|
| #train  | 1599 | 1160     | 12466  | 85788 | 3709     | 1642      |
| #valid  | 400  | 290      | 3117   | 21447 | 928      | 441       |
| #test   | 486  | 660      | 4134   | 62108 | 564      | 564       |

Nonetheless, the Squirrel dataset only has 71 nodes with $h_i > 0.5$. To ensure enough test nodes, we adapt the nodes with $h_i > 0.4$ as the minority nodes for test. We conduct experiments to measure the distributional shift between train and test nodes. To further analysis on the distribution shift, we employ Maximum Mean Discrepancy (MMD) [106] as a discrepancy metric. A large MMD distance indicates the covariate shift with $P^{\text{train}}(X) \neq P^{\text{test}}(X)$. MMD measures the distance between distributions as follows:

$$\text{MMD}^2(X, Y) = \left\| \frac{1}{n} \sum_{i=1}^{n} \phi(x_i) - \frac{1}{m} \sum_{j=1}^{m} \phi(y_j) \right\|^2 \tag{48}$$

where $\phi$ is a kernel function that maps the input samples into a higher-dimensional feature space, and $\| \cdot \|_2$ denotes the $L_2$ norm. Using the Gaussian kernel, we can define the kernel matrices as:

$$\begin{aligned} K_{xx,ij} = k(x_i, x_j), & \quad K_{yy,ij} = k(y_i, y_j), \\ K_{xy,ij} = k(x_i, y_j), & \quad K_{yx,ij} = k(y_i, x_j), \end{aligned} \tag{49}$$

where the kernel function is given by:

$$k(x, y) = \exp\left( -\frac{\|x - y\|^2}{2\sigma^2} \right) \tag{50}$$

and $\sigma$ is a parameter that controls the width of the kernel. In our experiment, we choose multiple $\sigma$ including [0.01, 0.1, 1, 10, 100]. We average the MMD distance with different $\sigma$ for a more robust distance measurement. The MMD distance can be computed using these kernel matrices:

$$\text{MMD}^2(X, Y) = \frac{1}{n(n-1)} \sum_{i \neq j} K_{xx,ij} - \frac{2}{nm} \sum_{i,j} K_{xy,ij} + \frac{1}{m(m-1)} \sum_{i \neq j} K_{yy,ij}, \tag{51}$$

where the first and third terms represent the within-set variations, and the second term represents the between-set variations. We then conduct experiments to determine where the source of distribution discrepancy from the covariate shift or concept shift. We measure the MMD distance between the train set and validation set as the i.i.d. distribution distance, and the MMD distance between the training set and test set as the OOD distribution distance. However, the in-distribution and out-distribution distances may not be directly comparable due to potential estimation errors introduced by different numbers of nodes in valid and test sets.

To enable a fair comparison, we also measure in-distribution and out-distribution distances on the i.i.d. data split as a control group. The MMD distances are measured on the last hidden representation of a well-trained GCN. The experiment results are provided in Table 15. Focusing on the i.i.d setting, we observe an MMD distance difference on the validation set and test set to the training set, induced by the difference in node counts between the validation and test set. Comparing the OOD setting with the i.i.d one no significant MMD differences are found, except for PubMed. This suggests that the primary factor driving the distribution shift is not the covariate shift with $\mathbf{P}^{\text{train}}(\mathbf{X}) \neq \mathbf{P}^{\text{test}}(\mathbf{X})$, but the concept shift with $\mathbf{P}^{\text{train}}(\mathbf{Y}|\mathbf{X}) \neq \mathbf{P}^{\text{test}}(\mathbf{Y}|\mathbf{X})$. The above observations further indicate the existence of the concept shift in our proposed OOD scenario.

**Additional OOD results**: We show additional experimental results on Cora and CiteSeer datasets in Table 16 with a consistent observation. Notice that, all experimental results are average with 10 random seeds including (1, 3, 5, 7, 11, 13, 17, 19, 23, 29).

Table 15: MMD distance between train and validation, test sets on both i.i.d. and ood settings.

| Dataset | Cora | CiteSeer | PubMed | Arxiv | Chameleon | Squirrel |
|---|---|---|---|---|---|---|
| IID valid | 0.565 | 0.345 | 0.082 | 0.149 | 0.951 | 1.04 |
| IID test | 0.610 | 0.600 | 0.050 | 0.276 | 0.882 | 0.92 |
| OOD valid | 0.564 | 0.233 | 0.127 | 0.211 | 0.977 | 1.192 |
| OOD test | 0.597 | 0.598 | 0.442 | 0.420 | 0.854 | 0.92 |

Table 16: Additional performance with OOD data split on Cora and CiteSeer.

| Dataset | Cora | CiteSeer |
|---|---|---|
| GCN (i.i.d) | 87.53±0.35 | 77.30±1.26 |
| GCN | 51.05±0.58 | 43.74±0.88 |
| MLP | 51.09±0.74 | 45.26±0.62 |
| GLNN | 54.14±0.89 | 47.21±0.33 |
| GCNII | 55.33±0.41 | 46.47±0.59 |
| GPRGNN | 55.16±0.51 | 45.76±0.50 |
| SRGNN | 62.79±0.87 | 49.62±0.96 |
| EERM | 54.34±0.78 | 47.72±0.50 |
| EERM(II) | 70.91±1.89 | 51.12 ± 0.82 |

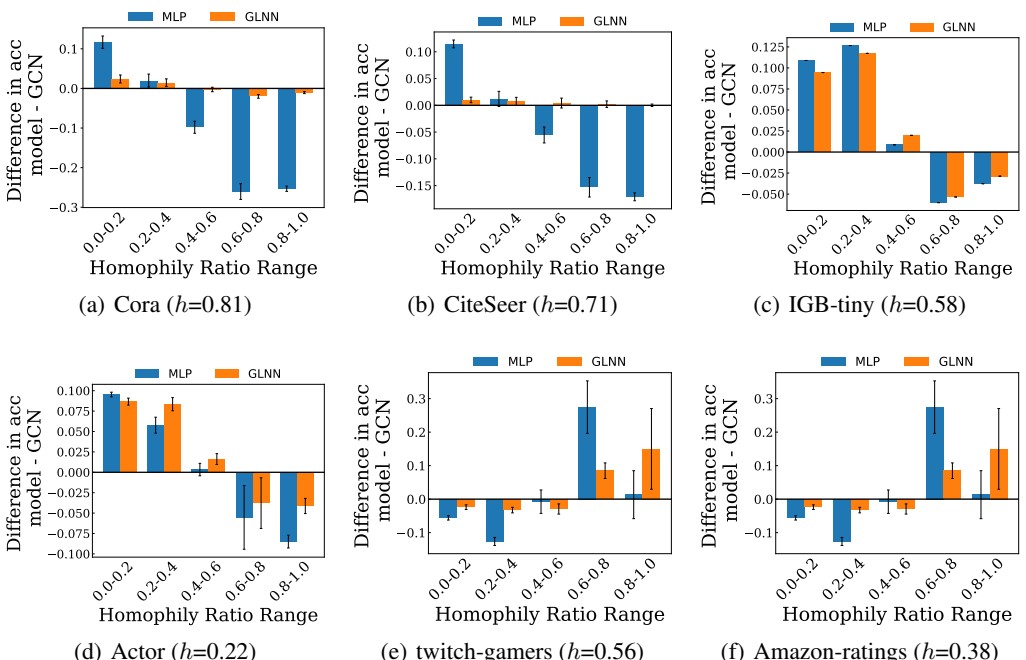

Figure 14: Performance comparison between GCN and MLP-based models on more datasets. Each bar represents the accuracy gap on a specific test node subgroup exhibiting a local homophily ratio within the range specified on the x-axis.

## H   Additional results on performance comparison between GCN and MLP-based models

In Section 3, we conduct a performance comparison between GCN with two MLP-based models, i.e., MLP and GLNN, on test nodes with different structural patterns. In this section, we provide more elaboration on why using GLNN and conduct additional experiments on more datasets.

We first provide more elaborations on why using GLNN. The experiment is to examine the effectiveness of GCN on utilizing different structural patterns. Therefore, we compare GCN and MLP

architectures as GCN utilizes graph structure during inference while MLP cannot, serving as a structure-agnostic baseline. When GCN surpasses MLP,it indicates GNN benefits from structural patterns effectively, and vice visa. Notably, GLNN can be viewed as a better-trained MLP model. GLNN also utilizes the same MLP model architecture as vanilla MLP, the only difference between vanilla MLP and GLNN is that GLNN is trained in an advanced distillation manner while vanilla MLP is trained with cross-entropy loss The reason why we utilize GLNN rather than only comparing GCN with vanilla MLP is that MLP meets optimization issue on training. Such an obstacle leads to a large performance gap (more than 20%) between under-trained vanilla MLP and well-trained GCN. Consequently, a large performance gap induced by training difficulty hinders the potential for MLP architecture. Contrastly, GLNN enjoys a better training process leads to a more clear comparison between well-trained GNN and well-trained MLP(GLNN) architecture with a convincing conclusion

Additional experiments are illustrated in Fig 14. Cora and CiteSeer are two Planetoid datasets, showing similar properties as PubMed. Observations on Cora and CiteSeer are consistent with those in Section 3, further substantiating our conclusions. It should be noted that our paper primarily focuses on homophilic graphs and heterophilic graphs with a clear major structural pattern. Nonetheless, there exists more complicated real-world graphs with more diverse structural properties, as demonstrated in Figure 13. We conduct further investigations on these datasets, including IGB-tiny, Twitch-gamers, Amazon-ratings, and Actor. A comprehensive discussion on those datasets can be found in Appendix G.3. Experimental results are depicted in Figure 14. We make initial observations as follows: (1) IGB-tiny is a homophilic dataset exhibiting consistent observations in line with the main analysis in Section 3, where the MLP-based models outperform GCN on the heterophilic nodes and underperform on homophilic nodes. (2) Twitch-gamers and Amazon-ratings are two heterophilic datasets exhibiting consistent observations in line with the main analysis in Section 3, where the MLP-based models outperform GCN on the homophilic nodes while falling short on the heterophily nodes. (3) Actor is a heterophilic dataset where graph information is of limited utility. Empirical evidence shows that MLP-based models show an overall better performance than GCN. From a local perspective, the MLP-based models outperform GCN on the heterophilic nodes while underperforming on the homophily nodes. Those observations indicate our conclusion can be extended on most datasets with diverse structural patterns Nonetheless, it still remains a mystery on the Actor dataset. We leave a more comprehensive investigation especially on the Actor dataset as the future work.

## I  Additional results on performance comparison between GCN and deeper GNN models

In Section 4.1, we conduct a performance comparison between GCN with deeper GNN models, on test nodes with different structural patterns. In this section, we conduct additional experiments on more datasets, correspondingly. Cora and CiteSeer are two Planetoid datasets, showing similar properties as PubMed. Notably, deeper GNNs generally show better performance across different structural patterns, where the improvement still primarily lies in the minority groups, Observations on Cora and CiteSeer are consistent with those in Section 3, further substantiating our conclusions.

It should be noted that our paper primarily focuses on homophilic graphs and heterophilic graphs with a clear major structural pattern. Nonetheless, there exist more complicated real-world graphs with more diverse structural properties, as demonstrated in Figure 13. We conduct further investigations on these datasets, including IGB-tiny, Twitch-gamers, Amazon-ratings, and Actor. A comprehensive discussion on those datasets can be found in Appendix G.3. New experimental results are depicted in Figure 15. We observe that deeper GNNs generally outperform GCN on the heterophilic node while underperforming on the homophilic nodes on those datasets with diverse structural properties. We can conclude that the effectiveness of GNN on those datasets with no clear majority structural patterns primarily from the improvement from the heterophilic nodes. We leave a more comprehensive investigation as the future work.

## J  Additional Experiment on performance disparity across subgroups

In Section 3.4, we conduct experiments to empirically examine the effects of two-hop aggregated-feature distance and the homophily ratio difference on the test performance of different GNN models. Despite the empirical success on the second hop, we conduct additional experiment results focusing

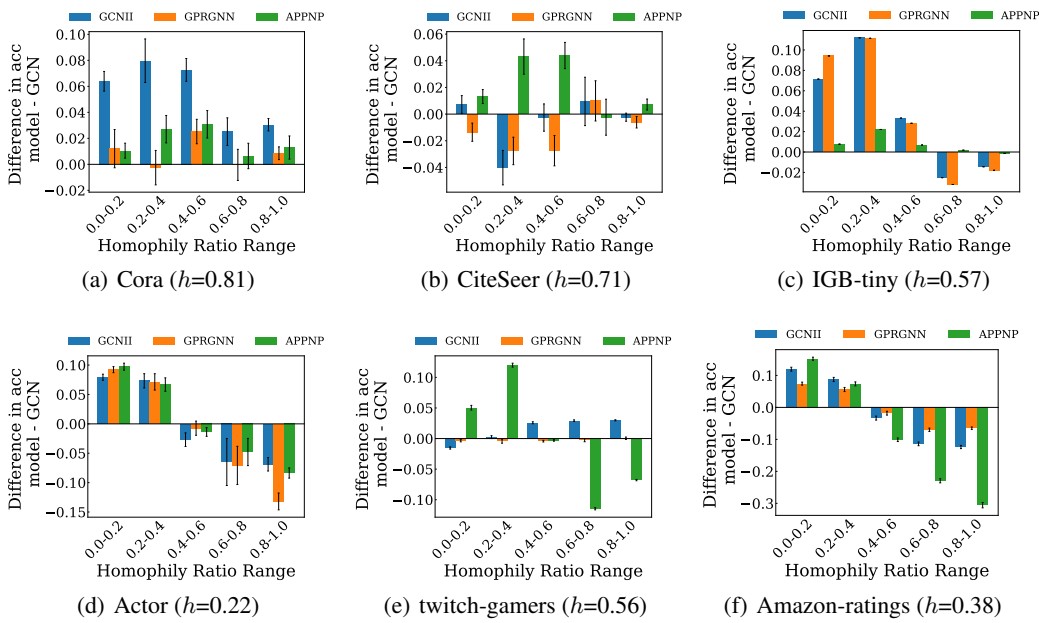

Figure 15: Performance comparison between GCN and deeper GNN models on more datasets. Each bar represents the accuracy gap on a specific test node subgroup exhibiting a local homophily ratio within the range specified on the x-axis.

on observations on higher-order homophily ratios, higher-order aggregated feature distance, and more datasets in this section. Notably, despite more complicated datasets, our theoretical analysis can still achieve empirical success.

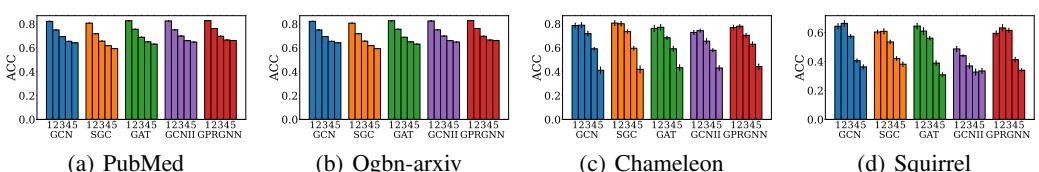

Figure 16: Test accuracy disparity across node subgroups by aggregated-feature distance and **higher-order homophily ratio differences** to training nodes. Each figure corresponds to a dataset, and each bar cluster corresponds to a GNN model. Bars labeled 1 to 5 represent subgroups with increasing differences to training set.

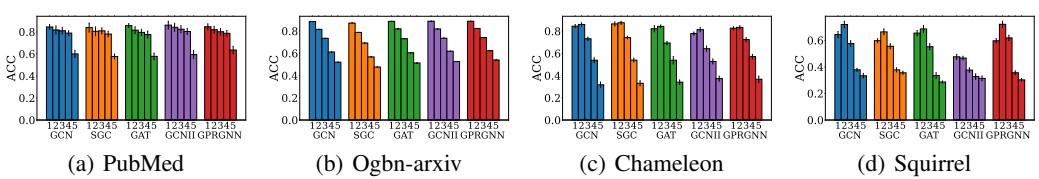

Figure 17: Test accuracy disparity across node subgroups by **higher-order aggregated-feature distance** and homophily ratio differences to training nodes. Each figure corresponds to a dataset, and each bar cluster corresponds to a GNN model. Bars labeled 1 to 5 represent subgroups with increasing difference to training set.

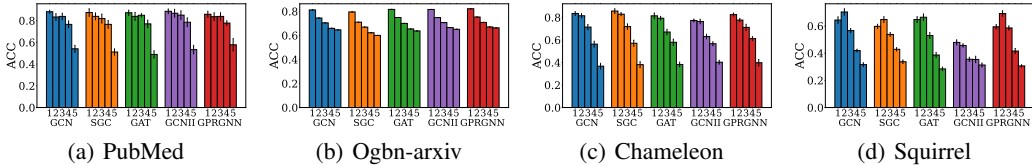

|  (a) PubMed | (b) Ogbn-arxiv | (c) Chameleon | (d) Squirrel |

Figure 18: Test accuracy disparity across node subgroups by **higher-order aggregated feature distance and higher-order homophily ratio differences** to training nodes. Each figure corresponds to a dataset, and each bar cluster corresponds to a GNN model. Bars labeled 1 to 5 represent subgroups with increasing difference to training set.

## J.1  Additional investigation on higher-order neighborhood

Our theory 1 in Section 3.3 focuses on the one-hop aggregation while it could be easily extended to the higher-order aggregation case in the real-world scenario. More experiments revolving on higher-order homophily ratio and aggregated features are shown in this subsection. In particular, the $k$-hop node homophily ratio can be calculated as:

$$ h_i^{(k)} = \frac{|\{u \in \mathcal{N}_k(v_i) : y_u = y_v\}|}{|\mathcal{N}_k(v_i)|} \tag{52}$$

where $\mathcal{N}_k(v_i)$ denotes the $k$-hop neighbor node set of $v_i$. We then sort test nodes in terms of the disparity score and split them into 5 equal-size. Figures 16 and 17 illustrate results on higher-order (third hop) homophily ratio difference and higher-order aggregated feature distance (third hop), respectively while Figure 18 illustrates on the result on higher-order homophily ratio difference and higher-order aggregated feature distance.

A similar phenomenon can be found with the expected trend of test accuracy in terms of increasing differences on the aggregated feature distance and homophily ratio differences in different scenarios. Exceptional observation can be found in the following scenarios: (1) Squirrel dataset with higher-order aggregated feature. It may be because such higher-order feature aggregation leads to more distribution overlapping between different categories. (2) PubMed with both higher-order aggregated features and homophily ratio. The potential reason is that the PubMed dataset does not rely on higher-order information. Overall speaking, those results further indicate that for the generalization of our theory in the real-world scenario.

## J.2  Additional results on more datasets

In this subsection, we conduct more experimental results on the datasets with more complicated with more diverse structural patterns, including Cora, CiteSeer, IGB-tiny, Twitch-gamers, Amazon-ratings, and Actor. A comprehensive discussion can be found in Appendix G.3. The performance on different node subgroups is presented in Figure 19. A consistent observation can be found with the expected test accuracy degradation trend in terms of larger feature distance and homophily ratio. the difference in most datasets, except for the Actor. The potential reason is that its structure could hardly provide useful information. The key evidence is that the vanilla MLP model outperforms all the GNN models shown in Table 11. The actor dataset shows entirely different properties with other datasets. Furthermore, we investigate on the individual effect of aggregated feature distance and homophily ratio difference in Figure 20 and 21, respectively. An overall trend of performance decline with increasing disparity score is evident though some exceptions are present. When only considering the aggregated feature distance to training nodes, the accuracy in Twitch-gamers exhibits an opposite increasing tendency along with the increasing difference to train nodes. Notably, when considering only the homophily ratio difference, the decreasing tendency is clear across datasets exception Actor. We can observe even better phenomenon on the Amazon-ratings dataset than combining both homophily ratio difference and the aggregated feature distance. It indicates that the homophily ratio difference w.r.t structural disparity significantly contributes to those datasets with more diverse structural patterns. We make the above initial observations showing the practicability of our theoretical analysis on more datasets with more diverse structural patterns. Moreover, exception can still be found on the Actor dataset. We leave a more comprehensive discussion on the Actor, as the future work.

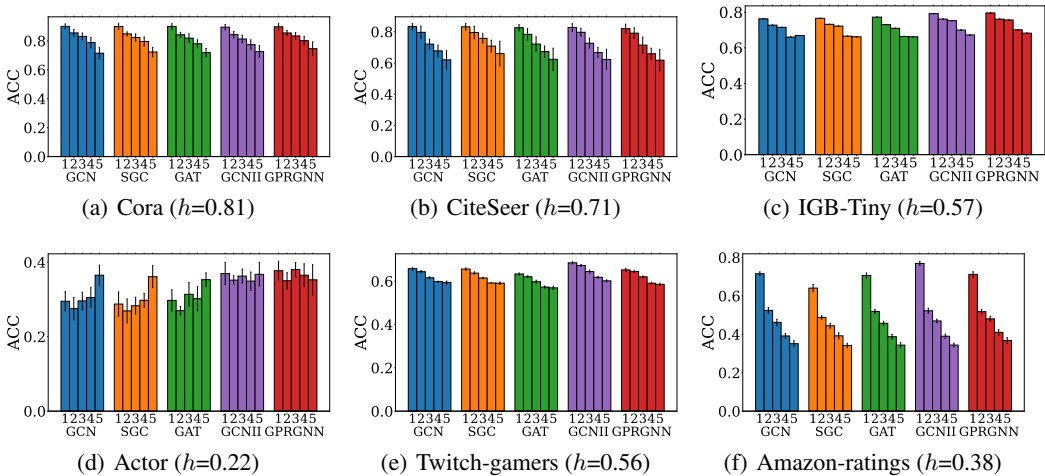

(a) Cora ($h$=0.81)  (b) CiteSeer ($h$=0.71)  (c) IGB-Tiny ($h$=0.57)

(d) Actor ($h$=0.22)  (e) Twitch-gamers ($h$=0.56)  (f) Amazon-ratings ($h$=0.38)

Figure 19: Test accuracy disparity across node subgroups by **aggregated-feature distance and homophily ratio differences** to training nodes on more datasets. Each figure corresponds to a dataset, and each bar cluster corresponds to a GNN model. Bars labeled 1 to 5 represent subgroups with increasing difference to training set. A clear performance decrease tendency can be found from subgroups 1 to 5 with increasing differences to the training set except the Actor dataset.

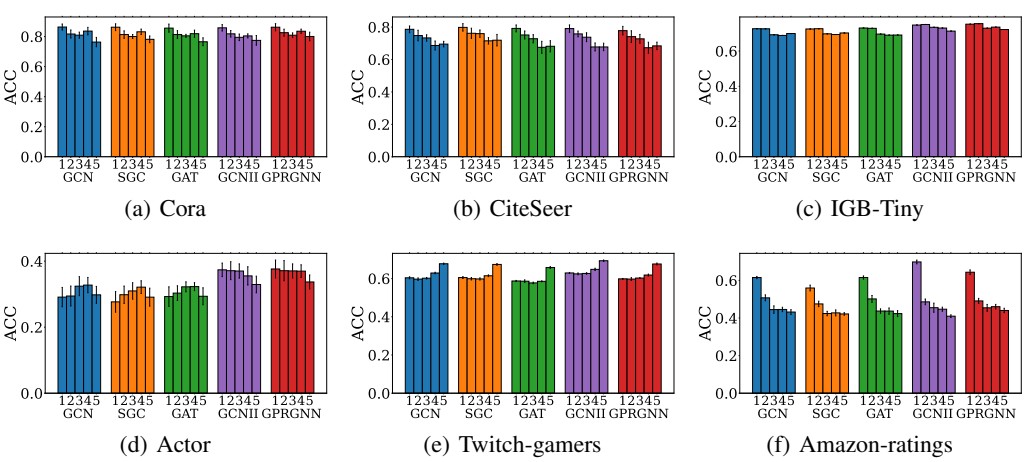

(a) Cora  (b) CiteSeer  (c) IGB-Tiny

(d) Actor  (e) Twitch-gamers  (f) Amazon-ratings

Figure 20: Test accuracy disparity across node subgroups by **aggregated-feature distance** to training nodes on more datasets. Each figure corresponds to a dataset, and each bar cluster corresponds to a GNN model. Bars labeled 1 to 5 represent subgroups with increasing differences to the training set.

## K  Additional investigation on the discriminative ability of GCN

In Section 3.2, we examine discriminative ratios on input features and multiple hops aggregated features for homophilic and heterophilic test nodes, respectively. Nonetheless, those experiments only focus on feature aggregation while ignoring the feature transformation and the training procedure. In this section, we further explore the discriminative ratios on well-trained GCNs with both feature aggregation and transformation. Typically, we utilize the hidden representation trained with the best hyperparameter as shown in Appendix G. Experimental results can be found in Figure 22. Observations on GCN are consistent with those in Section 3.2 focusing on aggregation, where majority nodes generally show a larger discriminative improvement than the minority nodes along with more aggregation layers, indicating a better discriminative ability on majority nodes than the minority ones. Those observations further substantiate our conclusions in a more general setting.

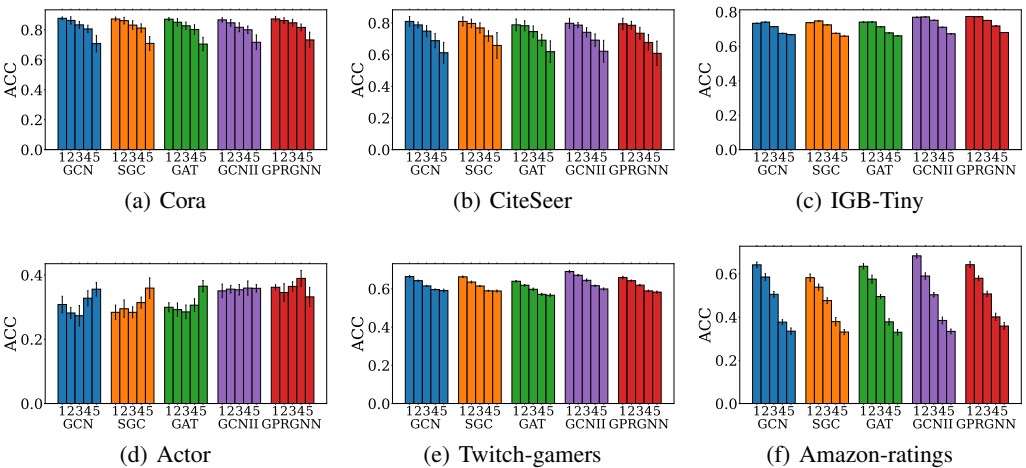

(a) Cora      (b) CiteSeer      (c) IGB-Tiny

(d) Actor      (e) Twitch-gamers      (f) Amazon-ratings

Figure 21: Test accuracy disparity across node subgroups by **homophily ratio differences** to training nodes on more datasets. Each figure corresponds to a dataset, and each bar cluster corresponds to a GNN model. Bars labeled 1 to 5 represent subgroups with increasing differences to the training set.

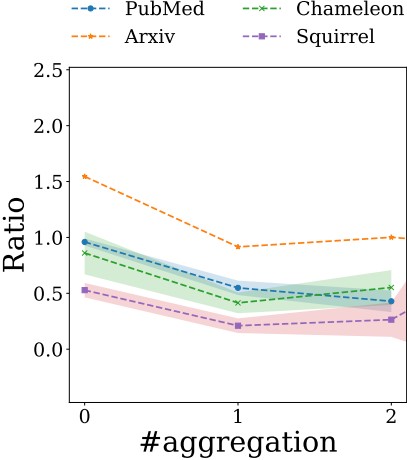

Figure 22: Illustration of the change on relative discriminative ratio along with the aggregation. The x-axis represents the number of aggregations and the y-axis represents the relative discriminative ratio.

## L   Significant test between GCN and MLP-based models

We conduct the performance comparison between GCN with MLP-based models, on test nodes with different homophily ratios, as shown in Fig 3. In the corresponding, we include the p-values of a paired t-test and confidence intervals between GCN and MLP-base models in Table 23, to determine whether the performance difference is statistically significant. The $x$-axis corresponds to the node homophily ratio range. The $y$-axis corresponds to p-values of whether the performance of GCN is significantly better or worse than the corresponding MLP model. Typically, $p < 0.05$ indicates statistical significance.

## M   Significant test between GCN and deeper GNN models

We conduct the performance comparison between GCN with deeper GNNs, on test nodes with different homophily ratios, as shown in Fig 8. In the corresponding, we include the p-values of a paired t-test and confidence intervals between GCN and deeper GNN models in Table 24, to determine

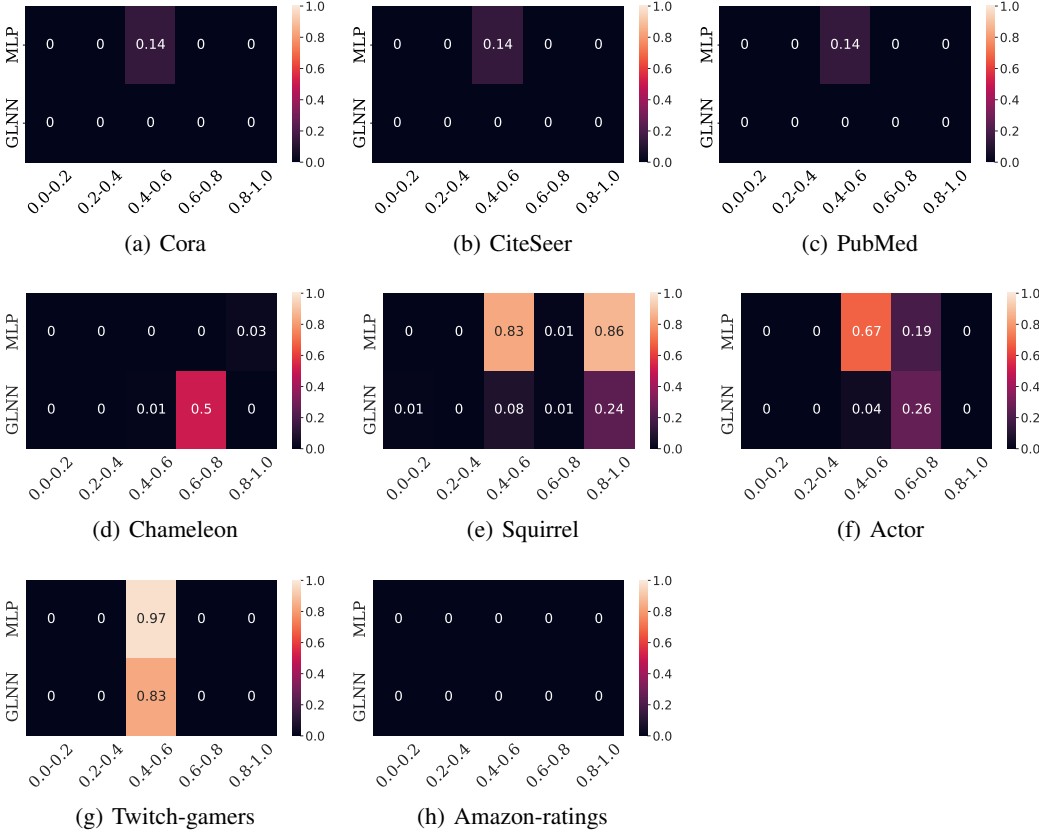

Figure 23: p-value of the paired t-test between GCN and MLP-based models with respect to accuracy on nodes with different homophily ratios.

whether the performance difference is statistically significant. The $x$-axis corresponds to the node homophily ratio range. The $y$-axis corresponds to p-values of whether the performance of GCN is significantly better or worse than the corresponding deeper model. Typically, $p < 0.05$ indicates statistical significance.

# N   Limitation

It is important to acknowledge certain limitations in our work despite our research providing valuable insights into the effectiveness of GNN on structural disparity. In order to facilitate a more feasible theoretical analysis, we have made several assumptions, the most significant of which is the use of the generalized CSBM-S model assumption in our theoretical analysis. This CSBM assumption, although prevalent in graph analysis, is subject to a notable drawback as the CSBM assumption postulates that feature dimensions are independent and identically distributed. It restricts the generality of our analysis.

Our current findings lay a strong foundation for further exploration. It is worth mentioning that our theoretical analysis predominantly focuses on simple aggregation techniques. We hope that these efforts will contribute significantly to the Graph Neural Network research community. Moving forward, we aspire to broaden our findings to encompass more advanced message-passing neural networks, with particular emphasis on deeper GNN models. Currently, there is only empirical evidence showing the effectiveness of the deeper GNN models. Besides, the node homophily ratio is calculated on the undirected graph while the one on the directed graph is still under exploration. More comprehensive analyses on the higher-order aggregated feature and higher-order homophily ratio are still under-explored. Moreover, We aim to utilize our understanding to mitigate the performance difference induced by the structural disparity in the future.

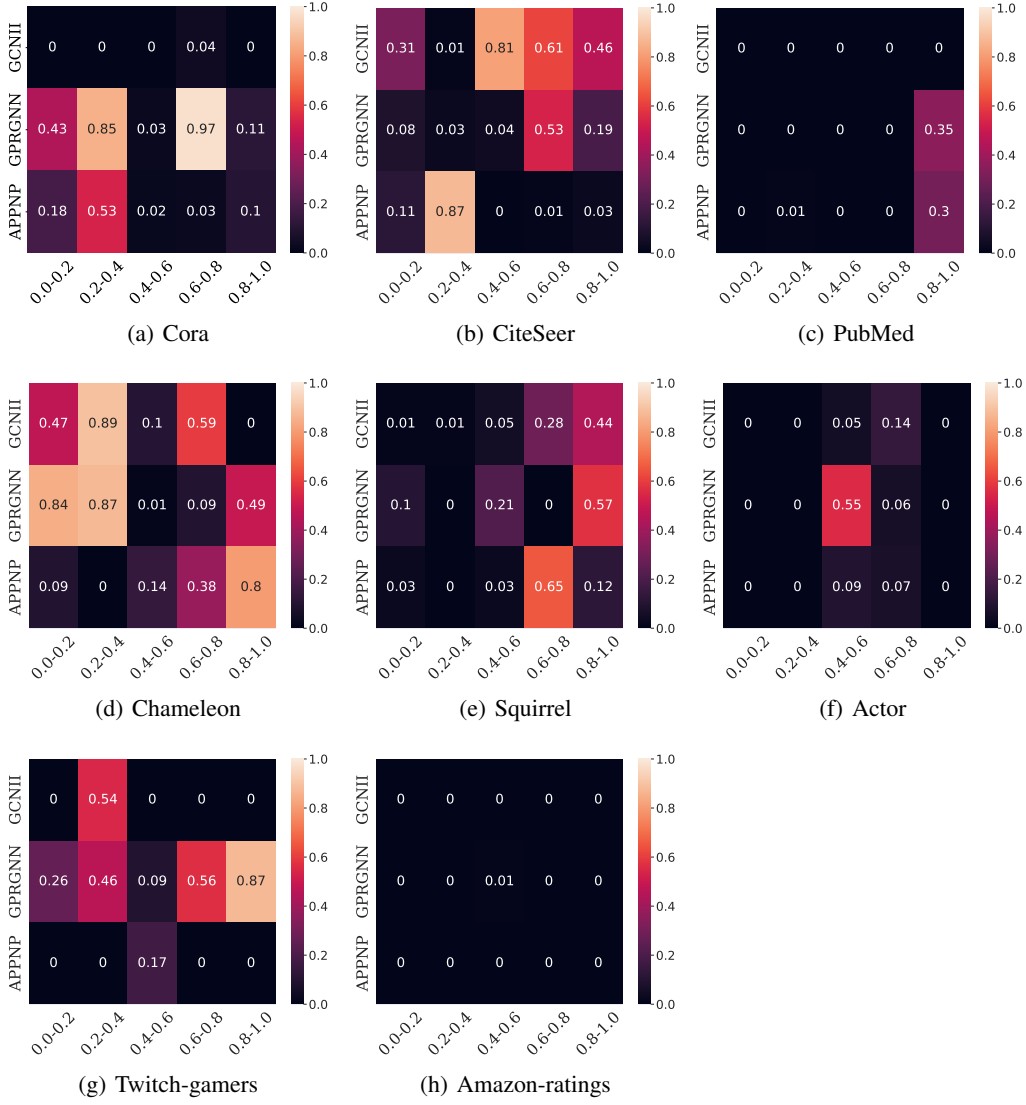

Figure 24: p-value of the paired t-test between GCN and deep GNN models with respect to accuracy on nodes with different homophily ratios.

## O   Broader Impact

Graph Neural Networks (GNNs) have emerged as a powerful architecture for modeling graph-structured data across a wide range of practical applications [2, 5, 3, 6–8, 73, 107]. A key aspect of GNNs is the inductive bias on leveraging neighborhood information. However, such characteristics may lead to a biased test prediction, particularly when majority neighborhood patterns are prevalent. For example, a scenario in which we are required to predict the gender of individuals with a dating network. GNN may excessively rely on the pattern that when all neighboring nodes are female, the center node should be male. This approach disregards the minority group of individuals who are more likely to date others of the same gender. This overlooked drawback in GNN may lead to ethical concerns.

In this study, we reveal the potential for performance disparity inherent in Graph Neural Networks and propose a novel OOD scenario for exploring methods to mitigate such issues. It is important to emphasize that our research offers an understanding of these limitations rather than introducing a new methodology or approach. We only identify the problem and show the potential solution to

address this issue. Consequently, we do not foresee any negative broader impacts stemming from our findings. We expect our work to contribute significantly to the ongoing research efforts aimed at enhancing the versatility and fairness of GNN models when applied to diverse data settings.

Our findings can also help to boost different graph domains. In section 4, we show how our understanding can help to elucidate the effectiveness of deeper GNN, reveal an overlooked OOD factor, and propose a new Graph out-of-distribution scenario. It is worth noting that our findings can derive new understandings of more graph applications and inspire us to identify new problems in graph domains. We further illustrate more understanding on the robustness of GNN and more initial understanding on Graph out-of-distribution problem as follows.

**Graph Robustness.** Recent studies have exposed the susceptibility of GNNs to graph adversarial attacks [108–112], where small perturbations can mislead GNNs into making incorrect predictions. several literature [113–115] reveals that increasing the heterophily of the homophily graph could be a key factor contributing to a successful attack. More recently, [116] observes that (1) learning-based attacks [114] tend to increase the heterophily in the smaller of train and test sets. (2) Directed attacks on particular nodes can outperform the undirected attack on the whole graph. Those observations are aligned with the understanding in our paper that (1) Increasing the heterophily of the homophilic graph enlarges the homophily ratio difference, leading to performance degradation. (2) Perturbations lying in the smaller set can efficiently enlarge the homophily ratio difference between training and test nodes with minimal perturbation budgets. (3) Undirected attack on the whole graph may not lead to a larger homophily ratio difference between the train and test nodes with permutation on both train and test nodes, leading to performance degradation.

**Understanding on Graph out-of-distribution problem.** Despite the new proposed Graph Out-Of-Distribution (OOD) scenario, our findings can further inspire a new understanding of existing graph OOD problems. We provide an initial discussion as follows. The uniqueness challenge of the OOD problem in the graph domain is that the discrepancy in graph structure can also lead to test performance degradation, in addition to node feature differences. There exists various graph OOD scenarios[62–64, 24, 66, 46], e.g., graph density, biased training labels, time shift, and popularity shift. Nonetheless, despite various graph OOD scenarios with different qualitative concepts, it still lacks of quantitative graph metrics to measure how the distribution shift happens, causing performance degradation. Inspired by our finding, We can roughly recognize the main reasons as graph covariate shift and concept shift [68–70, 117] in a graph context: (1) Graph covariate shift [118] is defined as $P^{\text{train}}(X) \neq P^{\text{test}}(X)$ where $P^{\text{train}}(\cdot)$ and $P^{\text{test}}(\cdot)$ are training and test distribution, respectively. Such graph covariate shift corresponds to the large feature distance $\|\mathbf{f}_u - \mathbf{f}_v\|$ between train and test nodes. (2) Graph concept shift is defined as $P^{\text{train}}(Y|X) \neq P^{\text{test}}(Y|X)$. Such graph concept shift corresponds to the homophily ratio difference $|h_u - h_v|$ between train and test nodes. We leave a more detailed discussion and a careful design on the graph OOD quantative metrics as a further work.

# P  Future work

**Further investigation on Deeper GNN**   In section 4, we conduct experiments on the effectiveness of deeper GNNs, indicating the performance gain is majorly from the improved discriminative ability on nodes in minor patterns. It could be a good direction to further explain and understand why such phenomenon happens. A potential reference for conducting discriminative analysis of deeper GNN is [39], aims to theoretically quantify the discriminative ability on deeper GNNs with more aggregations. Nonetheless, their analysis uses the vanilla CSBM model as the data assumption, denoted as $\text{CSBM}(\mu_1, \mu_2, p, q)$. The CSBM model presumes that all nodes follow either homophilic with $p > q$ or heterophilic patterns with $p < q$ exclusively. However, this assumption conflicts with real-world scenarios as homophilic and heterophilic patterns coexist across different graphs, as shown in Figure 2. As far as we can see, conducting a similar discriminative analysis as [1] on our proposed CSBM-M model considering both homophily and heterophily patterns could be a good solution.

**Other factors on performance disparity**   Existing literature [63, 46, 86, 87] shows that some other structural information, e.g., degree, geodesic distance to the training node, and Personal Pagerank score, could lead to a performance disparity. Nonetheless, all those analyses and conclusions are conducted on homophilic graphs, e.g., PubMed and ogbn-arxiv, while ignoring the heterophilic graphs e.g., Chameleon and Squirrel, which also broadly exist in the graph domain. It could be a good new research direction to see how the above factors, focus on the homophilic pattern in the context

of both homophilic and heterophilic properties. Another one could be how to find other important factors on both homophilic and heterophilic graphs.

**Solutions on performance disparity**    In this paper, we propose and analyze the importance of the performance disparity problem without no explicit solution. We provide two potential solution as follows:

- Combining MLP and GNN in an adaptive approach since MLPs can achieve better performance on minority nodes and GNNs better on majority nodes. Ideally, we can adaptively select MLP for minority nodes and GNN for majority nodes, using an adaptively gated function to control the proportion of MLP and GNN. Our findings suggest the homophilic/heterophilic pattern selection can serve as a guide for learning the gate function.
- Utilizing global structural information, as the global pattern is more robust, showing less disparity than the local structural pattern. Empirical evidence can be found in Figure 9 that the higher-order homophily ratio differences are smaller than the local structure disparity.

