# OpenReview forum: "Demystifying Structural Disparity in Graph Neural Networks: Can One Size Fit All?"
_NeurIPS.cc/2023/Conference — NeurIPS 2023 poster_

### Official Review · Reviewer_itvA · 2023-06-28

**Soundness:** 2 fair
**Presentation:** 2 fair
**Contribution:** 2 fair
**Rating:** 5
**Confidence:** 4

**Summary:**

This paper investigates the effectiveness of GNNs on nodes with different structural patterns in real-world graphs and proposes a new method to identify the reasons for performance disparities. The authors found that GNNs tend to perform well on homophilic nodes within homophilic graphs, but struggle with the opposite node set, and they provide insights into the performance disparities by analyzing aggregated feature distance and homophily ratio difference between training and testing nodes.

**Strengths:**

This paper provides both empirical and theoretical evidence to support its findings. It conducts rigorous analyses and derives a non-i.i.d. PAC-Bayesian generalization bound for GNNs, which adds credibility to the research. The authors substantiate their findings with controllable synthetic experiments and real-world datasets, further strengthening the quality of the paper.

**Weaknesses:**

1.	Writing needs to improve.
2.	Some explanation needs more elaboration.

**Questions:**

1.	Line 49-50, “while heterophilic graphs exhibit an opposite phenomenon with homophilic nodes in the majority and heterophilic ones in the minority.” Check this definition of heterophilic graphs.
2.	Line 55-56, “ Such differences may lead to performance disparity between nodes in majority and minority patterns.” If you randomly sample training and test data, the distribution of the majority and minority patterns should be the same for training and test nodes. Thus, the disparity shown in this example is not valid in practice.
3.	“Consequently, the performance disparity can be overwhelmed by such gap which renders the effect from structural patterns.” I don’t understand this sentence. Please elaborate the reason why you use GLNN here.
4.	Are the observations in Figure 3 also hold on other benchmark datasets?
5.	Line 170, why both homophilic and heterophilic patterns have p>q? In your definition, nodes from the subgroup but different classes have the same (p,q)?
6.	Line 179-181, check the English.
7.	“This proposition suggests that aggregation results in a distance gap between different patterns within the same class.” I cannot see how this proposition suggests the distance gap. Please elaborate it.
8.	In Lemma 1, are nodes u,v from the same class? Why do you want to examine “discrepancy between nodes and with the same aggregated feature but different structural patterns.”? How does this relates to the proposed claim about majority and minority nodes?

**Limitations:**

Yes

---

> ### Author Rebuttal · Authors · 2023-08-09
>
> **Q2:** Such differences lead to performance disparity between nodes in majority and minority patterns.If randomly sample train and test data,the distribution of majority and minority patterns is same for train and test nodes.Thus,the disparity shown in example is not valid in practice
>
> **R:** The focus of this paper is structural disparity but not distribution shift, which are not the same concept.Disparity shown in example is valid in practice for **there are always both homophilic and heterophilic nodes in the same graph**. The only difference from toy example is homophilic and heterophilic patterns may not be so extreme, there are some heterophilic edges in homophilic nodes,and vice versa.
>
>
> Details explanations:Structural disparity is that homophilic and heterophilic patterns exist simultaneously in a single node set. Such disparity consistently exists in all graphs, as shown in Figure 2, 13.If randomly sample train and test data, **both train and test set will have homophilic and heterophilic patterns, indicating structural disparity** happens on both train and test set.**Example is to show two patterns together in the same graph, not train is in homophilic pattern, but test is in heterophilic pattern.**
>
> Empirical evidence in Figure 2 shows structural disparity leads to performance disparity, where **experiments follow random split setting without distribution shift.**The reason for performance disparity is GNNs tend to learn better on the train majority pattern with more supervised signals while ignoring the minority ones. Then GNNs will perform well on the test majority node but not minority ones.
>
> **Q8:** In Lemma 1,are nodes u,v from the same class?Why examine discrepancy between nodes with the same aggregated feature but different structural patterns?How this relates to claim about majority and minority nodes?
>
> **R:** In lemma 1,we do not necessarily require nodes u, v from the same class, instead, we attempt to measure how likely nodes u and v are in the same class regarding to the structure disparity.
>
> We clarify section 3.1 aims to answer:How does aggregation affect nodes with different structural patterns? We examine how nodes with structure disparity(homophily ratio difference $|h_i-h_j|$) show different behaviors along with aggregation, the key operation in GNN.Thus, our analysis focuses on the node aggregated features in different structural patterns
>
> Following explanations are four-folds:1.what and how to measure behavior differences 2.what is structural disparity 3.Lemma 1 explanation:structural disparity leads to behavior difference 4.how lemma is correlated with claim about majority and minority nodes
>
> 1. In the case that behavior difference **does not exist**,nodes u and v the same aggregated feature $f_u = f_v$ **should be in the same class**. In contrast, behavior difference corresponds to the case that the same feature but not in the same class.The probability gap of two nodes sharing the same class $|P(y=c_1|f_u)-P(y=c_1|f_v)|$ is utilized to measure what extent behavior difference happens. Large $|P(y=c_1|f_u)-P(y=c_1|f_v)|$ indicates more likely that two nodes are in different class
> 2. Structural disparity is nodes in the same set but different structural patterns.We measure it with $|h_u-h_v|$
> 3. Lemma 1 shows when structure disparity $|h_u-h_v|$ is large, $|P(y=c_1|f_u)-P(y=c_1|f_v)|$ could be large, indicating node u and v are likely to have different classes,**violating the consistency behavior (with no structural disparity) expecting the same aggregated feature should map to the same class**
> 4. Lemma 1 does not directly correlate to majority and minority nodes, but serve as a necessary preliminary step for the claim about majority and minority nodes.Notably,in section 3.1,we only mention the existence of both homophilic and heterophilic patterns,without identifying which pattern is the majority.Lemma 1 only shows the existence of behavior disparity,serving as a preliminary step for analysis on majority and minority nodes.Once only behavior disparity exists on different structural patterns,it is possible for performance difference exists in majority and minority patterns
>
> **Q3:** Elaborate reason why use GLNN in Figure 1
>
> **R:** We rephrase with more evidence: Consequently, an under-trained vanilla MLP comparing with a well-trained GNN leads to an unfair comparison without rigorous conclusion
>
> The experiment is to examine the effectiveness of GCN on utilizing different structural patterns. Therefore, we compare GCN and MLP architectures as GCN utilizes graph structure during inference while MLP cannot, serving as a structure-agnostic baseline. When GCN surpasses MLP,it indicates GNN benefits from structural patterns effectively, and vice visa.Notably, GLNN can be viewed as a better-trained MLP model. GLNN also utilizes the same MLP model architecture as vanilla MLP, the only difference between vanilla MLP and GLNN is that GLNN is trained in an advanced distillation manner while vanilla MLP is trained with cross-entropy loss
>
> The reason why we utilize GLNN rather than only comparing GCN with vanilla MLP is that MLP meets optimization issue on training. Experimental results are shown in Figure 1(rebuttal pdf). Such an obstacle leads to a large performance gap (more than 20%) between under-trained vanilla MLP and well-trained GCN.**Consequently, large performance gap induced by training difficulty hinders the potential for MLP architecture.** Contrastly, GLNN enjoys better training process leads to a more clear comparison between well-trained GNN and well-trained MLP(GLNN) architecture with a convincing conclusion
>
> **Q4:** Are observations in Figure 3 hold on other datasets?
>
> **R:** Yes.See additional results in Figure 14, Appendix H. Conclusions also hold for Cora, CiteSeer, IGB-tiny, twitch-gamers, and Amazon-ratings datasets
>
> **Response on other questions on explanation and typos are in global rebuttal**
>
> - Q6,7->problem1
> - Q5->problem2
> - Q1->problem3

---

> > ### Author Response · Authors · 2023-08-18
> > **A gentle remind to reviewer itvA**
> >
> > Dear reviewer itvA
> >
> > Thanks for your review. Your time and efforts in evaluating our work are appreciated greatly. Could we kindly know if the responses have addressed your concerns and if further explanations or clarifications are needed? Thanks.

---

> > ### Comment · Reviewer_itvA · 2023-08-19
> > **Thanks for your rebuttal**
> >
> > The authors addressed most of my concerns. In general, I find this paper interesting. I will raise my score to 5.

---

> > > ### Author Response · Authors · 2023-08-19
> > > **Thanks for your response**
> > >
> > > Thank you for your feedback and support. We’re pleased to hear that our rebuttal has addressed your concerns. If there are any further issues or questions, please inform us, and we'll be glad to address them.

---

### Official Review · Reviewer_GxBV · 2023-07-04

**Soundness:** 3 good
**Presentation:** 3 good
**Contribution:** 2 fair
**Rating:** 5
**Confidence:** 4

**Summary:**

This paper focuses on the performance disparity on homophily and heterophily nodes in node-level classification tasks. It claims that although GNNs have good performance on both pure homophily and pure heterophily graphs, GNNs cannot perform well when dealing with graphs with both these two types of nodes. The paper then analyses the reason for different impact on majority and minority nodes and illustrates the aggregation function's contribution on this. It delves into the theoretical analysis, deriving a non-i.i.d PAC-Bayesian generalization bound based on the Subgroup Generalization bound of Deterministic Classifier. The theoretical analysis indicates that test nodes with larger aggregated feature distances and homophily ratio differences from training nodes experience performance degradation. The practical implications of the findings are demonstrated on real-world datasets.

**Strengths:**

- The paper provides sufficient real-world experiments to show how the homophily ratio and the feature distance influence the GNN’s aggregation and the actual performance on both homophilic and heterophilic datasets. This could sever as a foundation for future GNN architecture design.
-  The paper performs the theoretical analysis that reveals potential reasons for the performance disparity. The findings are supported by experimental results.
- The paper is well structured with good clarity in presentation.


**Weaknesses:**

- The paper identifies the problem of performance disparity but does not provide a solution to address it.
- The classification of nodes into homophily and heterophily classes using a hard threshold is a simple and ideal setting. This setting may fail to capture the complexity and nuances that may exist in real-world scenarios. Do the conclusions/findings still hold if we directly use the continuous homophily ratio rather than thresholding it?


**Questions:**

See the weaknesses section.

---

> ### Author Rebuttal · Authors · 2023-08-09
>
> **W1:** The paper identifies the problem of performance disparity but does not provide a solution to address it
>
> **R:** Thanks for the great question revolving the contribution of this paper. We would like to first provide a more comprehensive understanding on the motivation and contribution of our paper and then discuss the potential solution
>
> We first clarify the contribution of our paper as follows:
>
> 1. points out the missing fact that homophilic and heterophilic patterns coexist across graphs
> 2. offers a new local perspective to evaluate GNN performance and indeed
> 3. gains insights into which nodes and why GNN can work well or not.
>
> Overall speaking, our paper does not focus on specific model design but **provides a new landscape for understanding existing GNNs and paving the way for future GNN design.**
>
> More specifically, our work can
>
> 1. bring insights for previous research progress. We figure out how recent-developed deeper GNNs are better than vanilla GNNs
> 2. provide an outlook for future research to solve the performance disparity across different GNN architectures. It clearly points out the remaining improvement space for GNN design
> 3. give insights into different graph applications including Graph OOD, robustness,and fairness
>
> We believe our work is technique-sound with both empirical and theoretical insights, helping to understand and push the graph domain further.
>
> Accordingly, we will **add future work section with potential solutions and preliminary solution to mitigate performance disparity** as follows, leaving as inspiration for future work. Potential solutions are as follows:
>
> 1. combine MLP and GNN in an adaptive approach since MLPs can achieve better performance on minority nodes and GNNs better on majority nodes. Ideally, we can adaptively select MLP for minority nodes and GNN for majority nodes, using an adaptively gated function to control proportion of MLP and GNN. Our findings suggest the homophilic/heterophilic pattern selection can serve as a guide for learning the gate function.
> 2. utilize global structural information, as the global pattern is more robust, showing less disparity than the local structural pattern. Empirical evidence can be found in Figure 9 that the higher-order homophily ratio differences are smaller than the local structure disparity
>
> We then propose a simple solution to solve **performance disparity problem** inspired by the first potential solution. Instead of learning a gate function with homophilic/heterophilic pattern selection as guidance, we use a simple heuristic threshold to identify whether a node is in majority pattern or minority pattern, then select GCN and GLNN (an MLP-based model) for inference on test nodes in majority pattern and minority pattern, respectively. A simple smoothness-based metric $r=\frac{1}{|\mathcal{N}_i|}\sum_{j\in \mathcal{N}_i}\|\mathbf{x}_i-\mathbf{x}_j\|_F^2$ is then applied.A small r indicates center node i is similar to neighborhood nodes, reflecting the homophilic pattern
>
> Our proposed **minor selection** has the following steps
>
> 1. calculate $R=\{r_1,\cdots,r_n\}$ for all test nodes
> 2. sort the smoothness list R
> 3. If graph is a homophilic one, select largest $\alpha$% test nodes, utilizing MLP for inference, while others utilize GCN
> 4. If graph is a homophilic one, select smallest $\alpha$% test nodes, utilizing MLP for inference, while others utilize GCN
>
> Notably,**our preliminary study is based on the vanilla GCN model**, our minority selection can also combine with other GNNs.The focus of the proposed method is to mitigate the performance disparity issue while keeping the overall performance
>
> The overall accuracy, WDP, and WSD scores are shown in Table 1(rebuttal pdf). Overall accuracy is comparable with other models.  WDP and WSD are to evaluate performance disparity, defined as:
> $$
> WDP=\frac{\sum_{i=1}^D N_i\cdot|A_i-A_{avg}|}{N_{total}}
> $$
>
> $$
> WSD=\sqrt{\frac{1}{N_{total}}\sum_{i=1}^DN_i\cdot (A_i-A_{avg})^2}
> $$
>
> D is the number of groups,$N_i$ is the node number of group i,$A_i$ is the accuracy of group i, $A_{avg}$ is the weighted average accuracy of all groups.**A smaller WSD and WDP indicate the performance disparity is small**. Our proposed method shows much better fairness than vanilla GCN, and even better than deeper GNN in most datasets. Although our study was conducted on a tight schedule and is preliminary, the impressive performance indicates a promising direction. Further refined designs could enhance it.
>
> **W2:** The classification of nodes into homophily and heterophily classes using a hard threshold is a simple and ideal setting. It may fail to capture nuances that may exist in real-world scenarios. Do conclusions still hold if we directly use the continuous homophily ratio rather than thresholding it?
>
> **R:** Thanks for the question on homophilic and heterophilic patterns. We initially utilize node homophily ratio $h>0.5$ as homophilic node and $h<0.5$ as heterophilic node. Nonetheless, we want to clarify that the purpose of **using a hard threshold 0.5 is to ease the problem statement** and better understanding at the beginning of the paper. Notably, **most of analyses and conclusions do not revolve around the hard threshold.** For theoretical analysis in Sections 3.1 and 3.3, we majorly consider the homophily ratio difference $|h_i-h_j|$ where $h_i$ is the continuous homophily ratio.
>
> For empirical analysis, disparity scores $s_u=\|F_u^{(2)}-F_v^{(2)}\|+|h_u^{(2)}-h_v^{(2)}|$ in Section 3.4 also **considers the continuous homophily ratio case.** Moreover, we also verify the conclusion on multiple datasets with complexity and nuances scenarios as shown in Figure 13. Additional experiments shown in Figure 15-21 of Appendix J indicate the validity of our conclusion on Cora, CiteSeer, Amazon-rating, IGB-tiny, and Twitch-gamers datasets. The above observations indicate that our conclusion could successfully extend to multiple datasets with complexity and nuances.

---

> > ### Comment · Reviewer_GxBV · 2023-08-16
> > **Thank you for the response**
> >
> > The rebuttal has addressed my concerns. I will keep the current rating.

---

> > > ### Author Response · Authors · 2023-08-17
> > > **Thanks for your response**
> > >
> > > Thanks for your response. We are glad to know that our rebuttal has addressed all your concerns. Please let us know in case there still remain outstanding concerns, and if so, we will be happy to respond. Moreover, we still kindly hope you could consider raising the score if feasible.

---

### Official Review · Reviewer_JCYy · 2023-07-04

**Soundness:** 3 good
**Presentation:** 3 good
**Contribution:** 4 excellent
**Rating:** 8
**Confidence:** 4

**Summary:**

This paper provides a rigorous analysis of the effect of structural disparity on the performance of GNNs. The proposed CSBM-S model and the application of PAC-Bayes analysis, among others, show the different effects of aggregation on the performance of nodes with different structural disparity. The analysis further indicate subgroup generalization bound for GNN and elucidate the effectiveness of deeper GNNs.

**Strengths:**

1.	The problem considered in this paper is extremely important. I agree with the authors that real-world graphs can’t be easily classified as homophilic or heterophilic. Therefore, analyzing how structural disparity influences the performance is essential to GNN design.
2.	It is well backed up by theoretical analysis and examples. The authors show several nice analysis methods, including CSBM-S model and subgroup generalization bound for GNNs.
3.	Extensive experiment enables the conclusions presented in this paper to be convincing


**Weaknesses:**

1.	The conclusions hold in this paper share the same promise that the aggregation operation is neighbourhood averaging, which may not generalize to a broad range of GCNs models
2.	The writing could be improved in terms of tone and phrases. There are also some grammar and spelling errors


**Questions:**

There are two assumptions that nodes from different subgroup share the same distribution and similar degree distribution. How does the conclusions are still valid without the above assumptions as stated in the supplementary material.

---

> ### Author Rebuttal · Authors · 2023-08-09
>
> **W1:** The conclusions hold in this paper share the same promise that the aggregation operation is neighborhood averaging, which may not generalize to a broad range of GCNs models.
>
> **R:** Thanks for your great question pointing out the gap between our theoretical analysis and empirical result. We want to first clarify that the reason for utilizing average aggregation is for better motivation with the toy model in the introduction section and the theoretical analysis in Section 3.1 and 3.3, but not our ultimate conclusion.  Such mean aggregation is also widely adopted for theoretical analysis in many existing GNN literature [1-5]. Despite most existing GNNs may not be exact neighborhood averaging, they are still generally based on weighted neighborhood averaging''.
>
> Moreover, inspired by the theoretical understanding, we verify the conclusion with comprehensive empirical experiments which could be found in Section 3.4 and Appendix J. We can see that our theoretical results are still valid qualitatively across different datasets and various architectures. Note that, the experimental results include both shallow GNN, e.g., GCN, GAT and deeper GNN, e.g., GCNII, GPRGNN.Therefore, our conclusion that GNN can perform well on nodes in the majority pattern but not the minority pattern is still valid across architectures empirically.
>
> **W2:** The writing could be improved in terms of tone and phrases. There are also some grammar and spelling errors.
>
> **R:** We sincerely thank the reviewer for your valuable feedback. We have carefully revised the paper to address these weaknesses to meet the required standards.
>
> **Q1:** There are two assumptions that nodes from different subgroup share the same distribution and similar degree distribution. How does the conclusions are still valid without the above assumptions as stated in the supplementary material.
>
> **R:** Thanks for your great questions revolving on the necessity of assumptions in. We first need to clarity that assumptions are not strictly necessary but employed for the sake of elegant expression. We want to clarify that we only claim that assumption 2 can be loose but not assumption 1. Assumption 1 tells that node features within the same class are sampled from the same Gaussian distribution, regardless of different structural patterns. The reason why we adopt assumption 1 is that our paper majorly focuses on the structure disparity, controlling by $p$ and $q$, but not the disparity on the original node feature. Therefore, we just keep the samples from the same distribution.
> Such an assumption is also aligning with the real-world graph as most original node feature shows a strong correlation with the class information. Empirical evidence can be found in Table 10 and 11. We can see that the MLP taking only node feature as input also shows certain discriminative ability.
>
> The assumption 2 is that nodes follow the same degree distribution with $p^{(1)} + q^{(1)} = p^{(2)} + q^{(2)}$. To get rid of this assumption, we can assume that $p^{(1)} + q^{(1)} = \alpha (p^{(2)} + q^{(2)})$, where $\alpha \in [0, +\infty)$ is a proportionality coefficient.
>
> Then the new Lemma 1 is:
> $$
>     |\mathbf{P}_1(y_u=c_1|\mathbf{f}_u)-\mathbf{P}_2(y_v=c_1|\mathbf{f}_v)| \le \frac{\alpha \rho^2}{\sqrt{2\pi}\sigma} |h_u - h_v|
> $$
> Original Lemma 1 is
> $$
> |\mathbf{P}_1(y_u=c_1|\mathbf{f}_u)-\mathbf{P}_2(y_v=c_1|\mathbf{f}_v)| \le \frac{\rho^2}{\sqrt{2\pi}\sigma} |h_u - h_v|
> $$
> $\rho=\left \|\mathbf{u}_1-\mathbf{u}_2\right \|$ is original feature separability, independent with structure.We can see that the only difference is an additional coefficient $\alpha$ depending on the degree distribution differences. Such minor difference does not affect our conclusion that nodes with a small homophily ratio difference $\|h_1 -h_2\|$ are likely to share the same class.
>
> [1] Yao Ma, Xiaorui Liu, Neil Shah, and Jiliang Tang. Is homophily a necessity for graph neural networks? The Eleventh International Conference on Learning Representations. 2022.
>
> [2] Aseem Baranwal, Kimon Fountoulakis, and Aukosh Jagannath. Graph convolution for semi-supervised classification: Improved linear separability and out-of-distribution generalization. International Conference on Machine Learning. PMLR, 2021.
>
> [3] Aseem Baranwal, Kimon Fountoulakis, and Aukosh Jagannath. Effects of graph convolutions in multi-layer networks. In The Eleventh International Conference on Learning Representtions, 2023.
>
> [4] Haonan Wang, Jieyu Zhang, Qi Zhu, and Wei Huang. Augmentation-free graph contrastive learning. arXiv preprint arXiv:2204.04874, 2022.
>
> [5] Wu, Xinyi, et al. "A Non-Asymptotic Analysis of Oversmoothing in Graph Neural Networks." The Eleventh International Conference on Learning Representations. 2022.

---

> > ### Comment · Reviewer_JCYy · 2023-08-16
> >
> > Thanks for authors' response. Rebuttal has adressed most my concerns and I have raised my score. This work contribute a lot to analysis the effectiveness of GNNs on different types of graphs from node-level perspective, I recommend  acceptance.

---

> > > ### Author Response · Authors · 2023-08-17
> > > **Thanks for your response**
> > >
> > > Thanks for your response and support. We are glad to know that our rebuttal has addressed your concerns. Please let us know in case there still remain outstanding concerns, and if so, we will be happy to respond.

---

### Official Review · Reviewer_pgwY · 2023-07-24

**Soundness:** 3 good
**Presentation:** 4 excellent
**Contribution:** 3 good
**Rating:** 7
**Confidence:** 4

**Summary:**

This work tries to understand the effectiveness of GNNs w.r.t different structural disparities within a graph. Previous studies have focussed on GNN's effectiveness on overall graphs, but here the authors try to understand GNN's effectiveness w.r.t structural patterns such as homophilic and heterophilic nodes to provide deeper insight into GNN's performance.

Contributions include understanding why the performance of GNNs is rather good on homophilic nodes in homophilic graphs and heterophilic nodes in heterophilic graphs, but not on the opposite set. Specifically, how aggregated feature distances and homophily ratio differences in mixed graphs impact the GNN performance. Authors also present experimental results on how deeper GNNs perform better on minority node subgroups while also proposing a new data split strategy to where majority nodes are selected for train/validation and minority nodes for test.

**Strengths:**

GNN performance studies on datasets across difference structural subgroups is a crucial metric to understand the overall performance of the model. Quantitive understanding of the effect of aggregation and homophily ratio difference w.r.t train nodes is a significant metric to compare different GNN architectures. This provides new evaluation strategies for future GNN studies. The effect of new data splitting strategy is again a significant contribution and is useful to evaluate the overall GNN performance.



**Weaknesses:**

The work by itself is strong and would have been even stronger if presented with more understanding of the effectiveness of Deeper GNNs on doing comparatively well on minority nodes.

**Questions:**

1. Is there a way to prove deeper GNNs improved discriminative ability on minority nodes?
2. Are there any additional factors other than aggregation and homophily ratio difference that you thought about?

---

> ### Author Rebuttal · Authors · 2023-08-09
>
> **W1:** The work by itself is strong and would have been even stronger if presented with more understanding of the effectiveness of Deeper GNNs on doing comparatively well on minority nodes.
>
> **Q1:** Is there a way to prove deeper GNNs improved discriminative ability on minority nodes?
>
> **R:** Thanks for your great question which inspires us in the future work on analyzing the effectiveness of Deeper GNN, especially on proving deeper GNNs improved discriminative ability on minority nodes. Note that, the following content, providing a detailed discussion of more understanding on deeper GNN, will be added in the revision. The potential approach to achieve analysis on deeper GNN can be found as follows:
>
> 1. To empirically verify the improved discriminative ability on minority nodes, we apply a discriminative analysis with initial results in Figure 1(rebuttal pdf).The y-axis indicates the discriminative value $r=\sum_{i=1}^K\|\mu_i^{tr}-\mu_i^{mi} \|$ where $\mu_i^{tr},\mu_i^{mi}$ is are the prototype of class i on train nodes and test minority nodes. x-axis indicates the hidden representation of different hops. With more hops of aggregations, the discriminative value decrease, indicates improved discriminative ability. The above discussion and experimental results will add to the revision.
>
> 2. To theoretically verify the improved discriminative ability on minority nodes, we provide a potential thought based on our CSBM-S assumption and discrminative analysis of [1] in ICLR2023. [1] aims to theoretically quantify the discriminative ability on deeper GNNs with more aggregations. However, their analysis uses the vanilla CSBM model as the data assumption, denoted as $CSBM(\mu_1, \mu_2, p, q)$, where $\mu_i$ is the feature mean of class $c_i$ with $i \in \{1, 2\}$. The CSBM model presumes that all nodes follow either homophilic with $p>q$ or heterophilic patterns $p<q$ exclusively. However, this assumption conflicts with real-world scenarios as homophilic and heterophilic patterns coexist across different graphs, as shown in Figure 2. In contrast, our proposed CSBM-S model (Definition 1, line 167), $\text{CSBM-S}(\mu_1,\mu_2,(p^{(1)}, q^{(1)}),(p^{(2)},q^{(2)}), \Pr(homo))$, is more practical than the CSBM model.As far as we can see,conducting a similar discriminative analysis as [1] on our proposed CSBM-M model could be a good solution.
>
> Notably, the above discussion and experimental results will add to the future work section in the revision.
>
> **Q2:** Are there any additional factors other than aggregation and homophily ratio difference that you thought about?
>
> **R:** Thanks for your great question which inspires us in the future work on other important factors for performance disparity. Note that, the following content, providing a more detailed discussion of the important factor for performance disparity,will be added in Appendix.
>
> We first want to clarify that,in our paper, we find that both the aggregated feature distance and homophily ratio difference could lead to the performance disparity along with aggregation, the key operation in GNNs. Detailed discussion can be found in Section 3.
>
> Moreover, a comprehensive discussion in the Related Work section (Appendix A). Existing literature [2-5] shows that some other structural information, e.g., degree, geodesic distance to the training node, and Personal Pagerank score, could lead to a performance disparity. Nonetheless, all those analyses and conclusions are conducted on homophilic graphs, e.g., PubMed and ogbn-arxiv, while ignoring the heterophilic graphs e.g., Chameleon and Squirrel, which also broadly exists in the graph domain. It could be an exciting new research direction to see how the above factors, which focus on the homophilic pattern in the context of both homophilic and heterophilic properties.
>
>
>
> [1] Wu, Xinyi, et al. "A Non-Asymptotic Analysis of Oversmoothing in Graph Neural Networks." The Eleventh International Conference on Learning Representations. 2022.
>
> [2] Qi Zhu, Natalia Ponomareva, Jiawei Han, and Bryan Perozzi. Shift-robust gnns: Overcoming the limitations of localized graph training data. Advances in Neural Information Processing Systems, 34:27965–27977, 2021.
>
> [3] Jiaqi Ma, Junwei Deng, and Qiaozhu Mei. Subgroup generalization and fairness of graph
> 521 neural networks. Advances in Neural Information Processing Systems, 34:1048–1061, 2021.
>
> [4] Xianfeng Tang, Huaxiu Yao, Yiwei Sun, Yiqi Wang, Jiliang Tang, Charu Aggarwal, Prasenjit Mitra, and Suhang Wang. Investigating and mitigating degree-related biases in graph convoltuional networks. In Proceedings of the 29th ACM International Conference on Information Knowledge Management, pages 1435–1444, 2020.
>
> [5] Yushun Dong, Ninghao Liu, Brian Jalaian, and Jundong Li. Edits: Modeling and mitigatin data bias for graph neural networks. In Proceedings of the ACM Web Conference 2022, pages 629 1259–1269, 2022.630

---

### Author Rebuttal · Authors · 2023-08-09

Thanks to all reviewers for their constructive reviews. The pdf file contains new experimental results for reviewers pgwY and GxBV.



With the help of reviewers, we find some typos and writing issues in our paper. We will correct those mistakes and add more explanations for better understanding in our revision. Detailed problems are shown as follows.

**Problem1:** Grammar problem in Lines 179-181. Elaborate proposition 1.

**R:** The updated version of Proposition 1 is as follows. The aggregated feature mean distance between homophilic and heterophilic node subgroups within class $c_1$ is$\left\|\frac{p^{(1)}\mu_{1}+q^{(1)}\mu_{2}}{p^{(1)}+q^{(1)}}-\frac{p^{(2)}\mu_{1}+q^{(2)}\mu_{2}}{p^{(2)}+q^{(2)}}\right\|>0$, indicating the aggregated feature of homophilic and heterophilic subgroups are from different feature distributions, with a mean distance larger than 0 distance before aggregation, since original node features draw from the same distribution, regardless of different structural patterns.

The Lemma aims to show how nodes with structural disparity $|h_i-h_j|$ behave differently along with aggregation. Typically, behavior difference is measured with the distance of feature means from different structural patterns within the same class. **Behavior disparity and feature mean distance does not exist before aggregation**, since nodes in the same class are samples from the same original feature distribution $\mathcal{N}(\mu_1,\sigma^2)$, regardless of different groups, **with the same feature mean**. The feature mean distance should be $\|\mu_1-\mu_1\|=0$.

After aggregation, the aggregated feature mean of the homophilic group will be $\frac{p^{(1)} \mu_{1}+q^{(1)}\mu_{2}}{p^{(1)}+q^{(1)}}$ while the heterophilic group will be $\frac{p^{(2)}\mu_{1}+q^{(2)}\mu_{2}}{p^{(2)}+q^{(2)}}$. Since $p^{(1)}\ne p^{(2)}$ and $q^{(1)}\ne q^{(2)}$, the aggregated mean feature distance $\left\|\frac{p^{(1)}\mu_{1}+q^{(1)}\mu_{2}}{p^{(1)}+q^{(1)}}-\frac{p^{(2)}\mu_{1}+q^{(2)}\mu_{2}}{p^{(2)}+q^{(2)}}\right\|>0$. It indicates that **there is a distance gap feature mean from different node subgroups in the same class**, indicating behavior differences.

**Problem2:** typo in Line 170, heterophilic patterns have $p>q$

**R:** Heterophilic pattern should have $p^{(2)}<q^{(2)}$ as shown in line 163. We will correct it in revision. Consequently, the CSBM-S model has different p and q values where $p^{(1)}>q^{(1)}$ and $p^{(2)}<q^{(2)}$.

**Problem3:** typo in line 58-60 on homophilic and heterophilic

**R:** The revision is: while heterophilic graphs exhibit an opposite phenomenon with **heterophilic** nodes in the majority and **homophilic** ones in the minority.

---

### Decision · Program_Chairs · 2023-09-21

**Decision:**

Accept (poster)

**Comment:**

This paper studies the problem of understanding why GNNs work well on certain types of graph settings and not on others (for node classification tasks). For this purpose the authors propose to study the effect of the presence of homophillic and heterophillic nodes and show that GNNs work well when only one types of patterns are present but not when the two patterns are mixed. This is supported by theoretical analysis in the form of generalization bounds and experiments on real world graphs.

Overall the reviewers liked the paper and were positive about the results presents and the problem studied. During the discussion phase a few presentation issues were noted by the reviewers and the authors are strongly encouraged to fix these in the next version of the paper.